# Linking big models to big data: efficient ecosystem model calibration through Bayesian model emulation

Istem Fer[1], Ryan Kelly[2], Paul R. Moorcroft[3], Andrew D. Richardson[4,5], Elizabeth M. Cowdery[1], and Michael C. Dietze[1]

[1]Department of Earth and Environment, Boston University, Boston, MA 02215, USA
[2]RK Analytics, Durham, NC 27712, USA
[3]Department Organismic and Evolutionary Biology, Harvard University, Cambridge, MA 02138, USA
[4]School of Informatics, Computing and Cyber Systems, Northern Arizona University Flagstaff, AZ 86011, USA
[5]Center for Ecosystem Science and Society, Northern Arizona University, Flagstaff, AZ 86011, USA

**Correspondence:** Istem Fer (fer.istem@gmail.com)

**Abstract.** Data-model integration plays a critical role in assessing and improving our capacity to predict ecosystem dynamics. Similarly, the ability to attach quantitative statements of uncertainty around model forecasts is crucial for model assessment and interpretation and for setting field research priorities. Bayesian methods provide a rigorous data assimilation framework for these applications, especially for problems with multiple data constraints. However, the Markov Chain Monte Carlo (MCMC) techniques underlying most Bayesian calibration can be prohibitive for computationally-demanding models and large data sets. We employ an alternative method, Bayesian model emulation of sufficient statistics, that can approximate the full joint posterior density, is more amenable to parallelization, and provides an estimate of parameter sensitivity. Analysis involved informative priors constructed from a meta-analysis of the primary literature, specification of both model and data uncertainties, and introduced novel approaches to autocorrelation corrections on multiple data streams and emulating the sufficient statistics surface. We report the integration of this method within an ecological workflow management software, Predictive Ecosystem Analyzer (PEcAn), and its application and validation with two process-based terrestrial ecosystem models: SIPNET and ED2. In a test against a synthetic dataset, the emulator was able to retrieve the true parameter values. A comparison of the emulator approach to standard "bruteforce" MCMC involving multiple data constraints showed that the emulator method was able to constrain the faster and simpler SIPNET model's parameters with comparable performance to the bruteforce approach, but reduced computation time by more than two orders of magnitude. The emulator was then applied to calibration of the ED2 model, whose complexity precludes standard (bruteforce) Bayesian data assimilation techniques. Both models are constrained after assimilation of the observational data with the emulator method, reducing the uncertainty around their predictions. Performance metrics showed increased agreement between model predictions and data. Our study furthers efforts toward reducing model uncertainties showing that the emulator method makes it possible to efficiently calibrate complex models.

# 1 Introduction

Terrestrial ecosystems continue to be a major source of uncertainty in future projections of global carbon cycle. Model predictions disagree on the size and nature of the ecosystem response to novel conditions expected under climate change (Friedlingstein et al., 2014). This is partly due to different assumptions and representations of ecosystem processes in models (Fisher et al., 2014; Medlyn et al., 2015), and partly due to lack of constraints on uncertainties associated with modeled processes and parameters (Dietze, 2017b). Key to improving both model structure and calibration is to ground models in data through parameter data assimilation (PDA) which refers to the calibration of model parameters through statistical comparisons between models and real-world observations to improve the match between them (Richardson et al., 2010). However, despite having more models and data than ever before, we still have not successfully reduced the uncertainties in our predictions because of the technical difficulties of linking models and data together (Hartig et al., 2012; Fisher et al., 2014). This is particularly true for regional- and global-scale models, which are computationally complex and need to be calibrated against large datasets. Three specific technical challenges that need to be addressed in PDA are multiple data constraints, partitioning of uncertainties, and model complexity.

In Bayesian calibration it is possible to use more than one type of data to simultaneously constrain multiple output variables in a model. Using multiple data constraints is particularly helpful because model errors can compensate for each other and single variables often do not provide robust constraints (Raupach et al., 2005; Williams et al., 2009). However, implementing multiple data constraints is challenging because data are available at different spatial and temporal scales, with large differences in observational uncertainties and data volume between measurement types (MacBean et al., 2017; Keenan et al., 2013). The calibration of model parameters is sensitive to which data are used, how different data sources are combined, and how uncertainties are accounted for (Richardson et al., 2010; Keenan et al., 2011). As opposed to piecewise evaluation of different parts of the model against different data sets, a Bayesian framework allows the evaluation of the whole model at once against all data sources, reflecting the connections between variables and the covariances among parameters (Dietze, 2017a).

The Bayesian approach also distinguishes between parametric, model structural and data uncertainties, which is critical for ecological forecasting. Parameter uncertainty refers to the uncertainty about the true values of the model parameters due to data deficiency and model simplification (McMahon et al., 2009; van Oijen, 2017). As models are simplified representations of reality, it is often not possible to measure the true value of an ecosystem model parameter precisely in the field, regardless of the measurement errors (van Oijen, 2017). However, measurements can still provide estimates for parameter values that makes the model represent the reality better (van Oijen, 2017). Hence, it is possible to reduce parameter uncertainty with more measurements, conditioned upon the model structure and the measurement error (van Oijen, 2017; Dietze, 2017a). Therefore, the parameter uncertainty should be reflected by probability distributions and propagated into model predictions.

By contrast, process or model structural uncertainty refers to the uncertainty about how to represent ecological processes in models. As every model is a simplification of reality, there will always be underrepresented processes or insufficiently modeled interactions in ecological models (van Oijen, 2017; McMahon et al., 2009; Clark, 2005). With more observations, we can advance our theoretical understanding and better characterize ecological processes, but process uncertainty does not

necessarily decrease with more data, the way parameter uncertainty does (Dietze, 2017a; Gupta et al., 2012; Clark, 2005). As process uncertainty is part of our imperfect models, it is part of the uncertainty associated with the model predictions.

Unlike process and parameter uncertainties, data (observation) uncertainty does not need to be propagated into model predictions. Observation error is a result of the limited precision and accuracy of the measurement instruments, hence, the uncertainty about it is not part of the process that we are trying to model (van Oijen, 2017; McMahon et al., 2009). In Bayesian PDA, observation uncertainty should be treated independent of the deviations of model predictions from data as part of the likelihood for observations to inform model predictions without biases (Dietze, 2017a). For a more in depth terminology for these concepts in the context of process-based models and Bayesian methods, see review by van Oijen (2017).

Despite the advantages to the Bayesian paradigm when it comes to estimating parameters for ecosystem models, most of this research remains focused on computationally inexpensive models (such as SIPNET, Sacks et al. (2006); DALEC, Keenan et al. (2011); Lu et al. (2017); FöBAAR, Keenan et al. (2013)). This is largely due to the relatively high computational costs of Markov Chain Monte Carlo (MCMC) techniques underlying most Bayesian computation. Such techniques can require models to be evaluated $10^4$ - $10^7$ times, which can be prohibitively expensive for even simple models, let alone complex simulation models that may take hours to days to complete a single evaluation. In this aspect, the Markovian nature of MCMC techniques, which requires that the computation be performed sequentially, proves to be a fundamental limitation. By contrast, high-performance computing environments are optimized for parallel computation and advances in computing power are increasingly coming in terms of number of processors rather than CPU speed. Thus, it is particularly advantageous to consider techniques that are both parallel in nature and which have substantial "memory" (i.e. they use the results from all previously evaluated parameter set in proposing new parameters rather than just the previous or last few points).

One possible solution to this challenge is through model emulation (Sacks et al., 1989). An emulator (also referred as 'surrogate' in the literature) is a statistical model that is used in place of the full model in cases where an exhaustive analysis of the full model would be computationally prohibitive. In the emulator approach, we first propose a set of parameter vectors according to a statistical design (each parameter vector defines a point in multivariate parameter space). Then, we run the full model with this set of parameter vectors, and compare the model outputs with data. Next, we fit a statistical approximation through the design points (a.k.a. knots, see black dots in Fig. 1) which we obtain by evaluating the model. Once built, emulators generally take far less time to evaluate than the model itself, therefore the emulator is then used in place of the full model in subsequent analyses, i.e. it could be passed to a MCMC algorithm. In comparison to the $10^4$ - $10^7$ sequential model runs required for MCMC, far fewer model runs are required to construct the emulator, and these runs can be parallelized, as the design points in parameter space are proposed at the beginning or iteratively in large batches.

Emulators are constructed by interpolating a response surface between the knots where the model has been run. Previous studies on emulation of biosphere models mostly focused on emulating the model outputs (Kennedy et al., 2008; Ray et al., 2015; Huang et al., 2016). However, comparing model outputs to 'big data' requires emulating a large, nonlinear multivariate output space. Furthermore, for the purpose of model calibration what we are actually interested in is not the output space itself but the mismatch between the model and the data, which can typically be summarized by much lower dimensional statistics (e.g. sum of squares).

Instead of constructing an emulator for the raw model output, we adopt the approach of constructing an emulator of the likelihood – the statistical assessment of the probability of the data given a vector of model parameters which forms the basis for both frequentist and Bayesian inference. Emulating the likelihood has the advantage that likelihood surfaces are generally smooth and univariate (Oakley and Youngman, 2017). A further novel generalization we introduce in this study is to emulate the sufficient statistics of the likelihood that contains all the information to calculate the desired likelihood, rather than the likelihood itself. This facilitates estimating the statistical parameters in the likelihood, such as the residual error.

Overall, the goal of this study is to validate the emulator's performance against bruteforce MCMC methods in terms of parameter estimation, and assess the trade-offs in clock-time and emulator approximation errors. We first tested the emulator performance with the simplified Photosynthesis and Evapotranspiration (SIPNET) model against a synthetic dataset where we know the true values. Next, we compare both bruteforce and the emulator for calibrating SIPNET against data from the Bartlett Experimental Forest Ameriflux site, a temperate deciduous forest in the northeastern US. Third, we use the emulator technique to calibrate the Ecosystem Demography model (version 2, hereinafter ED2), whose computational demands preclude MCMC calibration. Finally, we evaluate the scaling properties of the emulator method and discuss its potential limitations and future applications.

## 2   Methods

### 2.1   Emulator-based calibration

A primary methodological focus of this paper is on the technique of parameter data assimilation using a model emulator. The general workflow of the emulator method (Figure 1) is given in Algorithm 1.

As a first step (1), it is critical to decide carefully where in parameter space the full model will be evaluated. This step is nontrivial because the space encompassed increases rapidly with the number of parameters, making exhaustive searches of the parameter space impractical. Furthermore, the total number of model evaluations is usually limited due to the computational costs of running the full model. As the emulator is an approximation, adding more design points to explore the parameter space means less approximation error. However, due to the trade-off between the accuracy and the clock time, we also do not want to propose too many knots. Therefore, we need to choose a design that maximizes information from a limited number of runs. Proposing points at random is inefficient because some points will be close together and thus uninformative – in practice a sampling design that is over-dispersed in parameter space is preferable. Here, we use a Latin Hyper Cube (LHC) design whereby a sequence of values is specified for each parameter that has the same length as the total number of samples and then each sequence is randomly permuted independent of the others to construct the overall design matrix. In the current application, the sequences for each variable are constructed to be uniform quantiles of the prior distributions (see section, *Model information and priors*), which results in greater sampling in the regions of higher probability and less sampling in the tails for non-uniform prior distributions.

The second step (2) is to evaluate the full model using the proposed parameter vectors, and it is the only step where we run the full model. As these model runs are independent of each other, they can be performed in parallel. Next (step 3), a

**Algorithm 1** Emulator workflow

---

(1) Propose initial $N_{knots}$ parameter vectors

(2) Run full model with each parameter vector (parallelizable over $N_{knots}$)

(3) For each model run ($K$), compare each data set to the appropriate model output variable ($V$) and calculate a sufficient statistic ($T_{V,K}$) summarizing model error

(4) Fit a separate Gaussian Process ($GP_V$) model for each $T_V$ to construct a response surface describing how model error varies across parameter space (parallelizable over $V$)

(5) Perform MCMC using the emulators

**for** $i = 1$ to $N_{MCMC}$ **do**

    (5a) Propose a new vector of process-model parameter values

    (5b) Use $GP_V$ to draw both the current and proposed $T_V$ with interpolation uncertainty (parallelizable)

    (5c) Calculate likelihoods from $T$

    (5d) Calculate current and proposed posterior values, $P_i$ and $P_{i-1}$

    (5e) Accept/reject according to the Metropolis-Hastings rule, $P_i/P_{i-1}$

    (5f) Gibbs update statistical parameters conditional on process-model parameters

**end for**

(6) (optional) Refine emulator by proposing new design points, **goto** (2)

---

sufficient statistic ($T$) is calculated by comparing each model output to each data set (Fig. 1). Statistic $T$ is sufficient for the job of estimating the unknown parameters "when no other statistic calculated from the same sample provides any additional information" (Fisher, 1922). We treat the deviations of model predictions from data in terms of sufficient statistics ($T$), instead of the likelihood itself, because we want to estimate data-model parameters, such as the residual error, as part of the MCMC.

For example, assume the residuals are distributed Gaussian. In this case, $T$ for a Gaussian likelihood would be the sum of squared residuals, $\Sigma(y_i - \mu_i)^2$, where $y$ is the observation and $\mu$ is the model prediction:

$$L = \prod_{i=1}^{n} N(y_i \mid \mu, \tau) = \prod_{i=1}^{n} \frac{\sqrt{\tau}}{\sqrt{2\pi}} \exp\left(\frac{-\tau(y_i - \mu)^2}{2}\right) \tag{1}$$

$$lnL \propto \frac{n}{2} ln(\tau) - \frac{\tau}{2} \underbrace{\sum_{i=1}^{n}(y_i - \mu)^2}_{T} \tag{2}$$

From Eq. (2), if we know $T$, we can calculate the likelihood without needing the full data set and the model outputs. This

allows us to not only accept/reject a proposed parameter vector (5e) but also sample the precision parameter, $\tau$, conditional on that parameter vector (step 5f). Such $T$ can be found for other likelihood functions as well.

This approach requires constructing an emulator for each data set (Step 4), instead of building one emulator on the overall likelihood surface. For example, if carbon ($C$) and water ($H_2O$) fluxes are used for constraining the model parameters, we

need to build one emulator that estimates the $T_C$ and another one that estimates $T_{H2O}$. Then, at each iteration of the MCMC,

we can update the model errors ($\tau_C$ and $\tau_{H2O}$) for each response variable conditional upon the emulated $T$. However, both the construction and evaluation of the emulator for each $T$ can be done in parallel, therefore, building more than one emulator does not defy the purpose of reducing computational costs.

In this study, we fitted a Gaussian process (GP) model as our statistical emulator, using the "$mlegp$" (v3.1.4) package in R (Dancik, 2013). GP assumes that the covariance between any set of points in parameter space is multivariate Gaussian, and the

correlation between points decreases as the distance between them increases ($mlegp$ uses power exponential autocorrelation function). We chose a GP model as our emulator because of its desirable properties: First, because GP is an interpolator rather than a smoother it will always pass exactly through the design points. Second, GP allows for the estimation of uncertainties associated with interpolation – uncertainty for a GP model will converge smoothly to zero at the design points (knots, Fig. 1). Third, among non-parametric approaches, GP is shown to be the best emulator construction method (Wang et al., 2014). The

GP model is essentially the anisotropic multivariate generalization of the Kriging model commonly employed in geostatistics (Sacks et al., 1989). Because we are dealing with a deterministic model, we assume that the variance at a lag of distance zero, known as the nugget in geostatistics, is equal to zero, but this assumption could be relaxed for stochastic models. We do not go into further details of GP modeling, or its comparison to other emulator methods since both are well-documented elsewhere (e.g. Kennedy and O'Hagan (2001); Rasmussen and Williams (2006)).

Once constructed, we pass the emulator to an adaptive Metropolis-Hastings algorithm (Haario et al., 2001) with block sampling, i.e. proposing new values for all parameters at once (Step 5). In the MCMC, we use the GP to estimate $T$ for both the current and the proposed parameter vector at each iteration (5b). GP provides a mean and the variance for the estimated values (here $T$) given the parameters. To propagate this interpolation uncertainty, it is important to draw the $T$ stochastically from the GP, and draw new values for both the current and proposed parameter set at each iteration. Once the process-model parameters

are updated according to the Metropolis ratio of current and proposed posteriors, statistical parameters of the likelihood can be updated via Gibbs sampling conditional upon the updated process-model parameters (5f).

To build the emulator, the parameter vectors need not be dependent on one another in a Markovian sense. This is in contrast with traditional optimization and MCMC algorithms that only leverage the current vector of parameter values when proposing new parameters. The independence of runs here allows us to efficiently leverage all previous runs, in addition to the model

evaluations from this step, to iteratively refine the emulator (step 6). Iteratively proposing additional knots over multiple rounds can be more effective because each round refines our understanding of where the posterior is located in parameter space, allowing new knots to be proposed where they provide the most new information. In this study, new knots were added by proposing 10% of the new parameter vectors from the original prior distribution and 90% from the joint posterior of the previous emulator round (via re-sampling the MCMC samples in between the rounds). Unless otherwise noted, all emulator

calibrations in this study were run in 3 rounds, each with 100K iterations of 3 MCMC chains, using a total of $p^3$ knots for $p$ parameters.

We compared the emulator approach to the Differential Evolution Markov Chain with snooker update algorithm (DREAMzs) as it is one of the fastest converging algorithms known in the literature (Laloy and Vrugt, 2012). The implementation of

DREAMzs was provided by the BayesianTools package (Hartig et al., 2017) which is called within the bruteforce data as-
similation framework of PEcAn (v1.4.10), an ecosystem modeling informatics system (LeBauer et al., 2013). The emulator
framework has also been implemented in PEcAn. Both ecosystem models (see next section) used in this study were coupled
to PEcAn and the specific runs reported in this paper are given in the supplementary material, Table A7-8. All PEcAn code is
available on GitHub (https://github.com/PecanProject/pecan), and the parameter data assimilation (PDA) modules developed
here are accessible via modules/assim.batch and modules/emulator. In addition, a virtual machine version of PEcAn with model
inputs, and code required to reproduce the present study is available online (http://pecanproject.org).

## 2.2 Multi-objective parameterization

We focus on three joint data constraints from Bartlett Experimental Forest, NH (Lee et al. (2018); also see supplement, *Study
site*): Net Ecosystem Exchange (NEE) and latent heat flux (LE) as measured by the eddy-covariance tower, and soil respiration
(SoilResp) as sampled within the inventory plots.

NEE and LE data were filtered for the low friction velocity, u*, values to eliminate time periods of poor mixing. A con-
servative u* of 0.40 was selected, which results in an elimination of 76% of the night-time data. Flux data was not gap-filled
because this results in a model-model comparison rather than a model-data comparison. The error distribution of flux data is
known to be both heteroskedastic, with variance increasing with the magnitude of the flux, and to have a double exponential
distribution (Richardson et al., 2006; Lasslop et al., 2008). In previous studies, the error distributions of high flux magnitudes
and fluxes averaged over time were also argued to be approximately Gaussian (Lasslop et al., 2008; Richardson et al., 2010).
However, as we assimilate all flux magnitudes at half-hourly time-step and as the errors of flux data have heavy tails like a
Laplacian distribution (i.e. big errors are more common than they would be under a Gaussian distribution), we modeled the
error distributions of NEE and LE fluxes as asymmetric heteroskedastic Laplacian distribution:

$$Flux_{data} \sim Laplace(Flux_{model}, \alpha_0 + \alpha_1 * Flux_{model}) \tag{3}$$

$$\alpha_1 = \begin{cases} \alpha_p, & \text{if } Flux_{model} \geq 0 \\ \alpha_n, & \text{otherwise} \end{cases}$$

where Laplace($\mu$, $\alpha$) refers to the Laplace distribution that models the distribution of absolute differences between model
prediction and data. Here we accounted for the fact that flux errors scale differently for positive and negative fluxes by using
different scale parameters $\alpha_p$ and $\alpha_n$, respectively.

Because NEE and LE data are time-series, we cannot treat each residual as independent. To reduce the influence of error
autocorrelation on parameter estimation, we correct the likelihoods by inflating the variance terms by $N/N_{eff}$ where $N$ is
the sample size and $N_{eff}$ is an estimate of the effective sample size based on the autocorrelation of the residuals. However,
estimating $N_{eff}$ is not straightforward to do within the MCMC because, paradoxically, a poor model prediction would end

up with higher autocorrelation on the residuals, making the $N_{eff}$ smaller and the values producing those model outputs more likely. We also cannot calculate the autocorrelation on the data itself, because flux data contain considerable observation error, making the $N_{eff}$ larger than it should be (i.e. also paradoxically indicating that the data provide more information the larger the observation error). To address these apparent paradoxes we propose a two-step approach to estimating effective sample size. First, the latent unobserved "true" fluxes were estimated via a state-space time series model fitted to the flux data, which allows separation of observation error from process variability (Dietze, 2017b). So as to not impose external structure on this filtering, we use a random walk process model. Second, the AR(1) autocorrelation coefficient, $\rho$, was estimated on the latent state time series and $N_{eff}$ was estimated as:

$$N_{eff} = N \frac{(1 - \rho)}{(1 + \rho)} \tag{4}$$

For soil respiration ($R_d$: data, $R_m$: model), we assume a Gaussian likelihood with a multiplicative bias, $k$, and a variance $\sigma_R^2$ which takes the form $R_d \sim N(k \cdot R_m, \sigma_R^2)$. The bias term is included to account for the scaling from the discrete soil collars to the stand as a whole (van Oijen et al., 2011). This term was also introduced because observed soil chamber fluxes were typically over twice the ecosystem respiration estimated from the eddy-covariance tower (Phillips et al., 2017). As in previous studies, this parameter is also estimated in the calibration (van Oijen et al., 2011), using a standard log-normal distribution as its prior. While the introduction of the bias term makes it impossible for this data to constrain the magnitude of soil carbon fluxes, it does provide information on the shape of the functional response (e.g. temperature dependencies). Due to the coarser time-step, small sample size (n=39), and the introduction of the bias term, no additional autocorrelation corrections were applied to the soil respiration data.

## 2.3 Model information and priors

The two models used in this study are SIPNET (Braswell et al., 2005) and ED2 (Medvigy et al., 2009). In the main text we will only describe the aspects of the models related to their calibration, further details of the models and their settings are given in the Supplement. Forest inventory data collected in the tower footprint were used to set initial conditions for the models (Table A1). We calibrate the models using data from 2005 and 2006. Both models provide outputs at the same half-hourly time steps as the assimilated flux data. SIPNET is a fast model ($\sim$ 5.5 sec per execution, in this study), which makes it suitable for application of traditional bruteforce MCMC methods. In constrast, it takes approximately 6.5 hours for ED2 to complete a single run for this 2-year period, which precludes its bruteforce calibration.

We targeted both the plant physiological and soil biogeochemistry parameters of the models. Unlike SIPNET, it is possible to run ED2 simulations with more than one competing PFT. To reduce the dimensionality of the calibration for ED2, differences among PFTs were assumed to vary proportionally to the differences among their priors and a parameter scaling correction factor (SF) was targeted by the parameter data assimilation algorithm instead of targeting each parameter per PFT. The SF operates on the prior CDF probability space [0,1]. For instance, when the SF for a certain parameter is 0.3, it would correspond to the 30% percentile of the parameter prior for each PFT.

We generated the priors and estimates for model parameters based on a Hierarchical Bayesian trait meta-analysis using PEcAn's workflow. Meta-analysis priors were specified by fitting distributions to raw data collected from literature searches, unpublished data sets, or from expert knowledge (LeBauer et al., 2013). Direct mapping of previous information to model parameters allows us to account for the uncertainties in measurements derived from the collective weight of a large range of studies rather than arbitrarily choosing values from any one study (LeBauer et al., 2017). The use of literature constraints ensures that the posterior parameter estimates fall within a biologically plausible range, and reduces the problem of equifinality, as parameters that are already well constrained cannot vary much, and thus cannot trade-off with poorly constrained parameters. The parametric prior and posterior distributions of the targeted parameters are given in Table A2 and A5-6 for SIPNET and ED2, respectively. The scaling factors used for common ED2 PFT parameters all have $Beta(1,1)$ prior distributions.

## 2.4 Emulator experiments

To test and validate the emulator approach we conducted the following experiments: 1) a test against synthetic data using the emulator with SIPNET, 2) comparison of emulator and bruteforce performances against real-world data using SIPNET, 3) calibrating ED2 with emulator using real-world data, and 4) a scaling test with the emulator to evaluate how the actual clock time varies as a number of design points (full model runs) using SIPNET.

Before these experiments, we conducted a predictive uncertainty analysis (for more details on the uncertainty analysis workflow in PEcAn please see LeBauer et al. (2013); Dietze et al. (2014)) to choose the model parameters for calibration. The parameters that can be constrained by data are those that contribute to the model predictive uncertainty for that corresponding variable. Figure 2 shows the plant physiology and soil biogeochemistry parameters of the models that are targeted by the calibration according to this uncertainty analysis. We chose a cut-off value of 5% for SIPNET, meaning we only targeted parameters that contribute more than 5% of the model predictive uncertainty for each variable of interest. In order to facilitate comparisons among the contributions of parameters to predictive uncertainty of output variables with different units, partial variances were used. Partial variances are the variances of each parameter divided by the sum of variances across all parameters per output variable. For ED2, we lowered this threshold to 1% because there are more than one PFT that shares the uncertainty. In the end, 9 and 10 parameters were targeted in SIPNET and ED2, respectively. To be more specific, the 8 (9) parameters for SIPNET (ED2) that are shown in Fig. 2, plus the multiplicative bias parameter were targeted in the PDA. Therefore, in total $9^3$ ($10^3$) knots were proposed in three iterative emulator rounds (also see Table 1 caption). For ED2, 6 out of the 9 model parameters were plant physiological parameters that are common to all its PFTs, for which we used the scaling factors (Fig. A5).

We first tested the emulator performance on retrieving true values using a synthetic dataset. We generated a random parameter set for the SIPNET parameters shown in Fig. 2, and ran the model forward with these values (Table A3). In order to give the synthetic data real characteristics, model outputs were reformatted to have the same gaps, time-steps and sample sizes as the data used in this study. Then, the likelihood parameters were calculated from the synthetic dataset, and next, further noise was added by drawing values from their respective likelihood functions to obtain the final synthetic dataset. In addition, the SoilResp data was multiplied by a constant ($k = 1.5$) to mimic the real world situation. Then, treating the model outputs as

a synthetic dataset, we tested whether emulator method posteriors converge on the true values. As this dataset was generated by the model itself, this approach allows us to assume that we have the perfect model (Trudinger et al., 2007; Fox et al., 2009). We compared the emulator run in three rounds to an emulator fit to the same number of knots in a single run to test whether increasing the number of knots iteratively is more effective than proposing the same number of knots in the beginning all-at-once.

We then tested the emulator with real-world data. As true parameter values are unknown, we assessed the emulator per-5 formance by comparing it to the bruteforce MCMC. In the bruteforce, the full model is run at every iteration, whereas in the emulator, the posteriors are approximated. Therefore, this experiment evaluates the influence of the numerical approximation error introduced by the emulator. As the larger computation time for ED2 does not permit the use of bruteforce, we only compared the pre- and post-calibration performance of ED2. The before and after calibration performances of both models were determined by comparing a 500 run model ensemble to data. Ensemble runs are forward model runs, with parameter 10 values randomly sampled from their distributions (which is the prior distribution for the pre-PDA comparison and the posterior distribution for the post-PDA).

In our scaling experiment, we evaluate the trade-off between the number of model runs and the approximation error by comparing the 8-parameter SIPNET bruteforce calibration to emulator calibrations with varying numbers of $k$ knots ($k = \{120, 240, 480, 960\}$). To do this, we compared the post-emulator PDA ensemble confidence interval errors relative (RCI) 15 to the post-bruteforce PDA ensemble CI in terms of mean Euclidean distance between their 2.5% - 97.5% CIs. For each experiment with $k$ different knots and variable $(CI_{E,L,k} - CI_{B,L,k})^2$ values were calculated where $E$ stands for emulator, $B$ stands for bruteforce ensemble, and $L$ stands for the lower CI limit. The same is calculated for the upper CI limit ($U$) and sum of their mean is used as a score for relative confidence interval (RCI) coverage per variable:

$$RCI_{VAR,k} = mean((CI_{E,L,k} - CI_{B,L,k})^2) + mean((CI_{E,U,k} - CI_{B,U,k})^2) \tag{5}$$

20 Next, each RCI vectors ($RCI_{VAR} = \{RCI_{VAR,960}, RCI_{VAR,480}, RCI_{VAR,240}, RCI_{VAR,120}\}$) are normalized by dividing by their mean to obtain values independent of the units. Then, the sum over the variables (in our case, $RCI_{FINAL} = RCI_{NEE}$ + $RCI_{LE}$ + $RCI_{SoilResp}$) gives is the final RCI score.

In an additional scaling experiment, we evaluated the capacity to calibrate the model with emulator vs. actual clock time. For this experiment, we chose $m$ parameters ($m = \{4, 6, 8, 10\}$) of SIPNET considering the order of their contribution to the 25 overall model predictive uncertainty (Fig. 2, Table A8). For each calibration, we again built an emulator with k knots. After calibration, we used overall deviance of 500-run ensemble mean as a metric to evaluate calibrated model performances.

**Table 1.** Time elapsed (in seconds) for each step of the emulator calibrations. "Model run time" refers to the computation time for running the LHC model ensemble needed to construct the emulator. Sub-columns refer to the rounds of the emulator ($1^{st}$: 243, $2^{nd}$: 486, $3^{rd}$: 729 = $9^3$ knots cumulatively for SIPNET; $1^{st}$: 334, $2^{nd}$: 667, $3^{rd}$: 1000 = $10^3$ knots cumulatively for ED2).

| | Model run time | | | GP model fitting | | | 100K MCMC | | | |
| --- | --- | --- | --- | --- | --- | --- | --- | --- | --- | --- |
| | $1^{st}$ | $2^{nd}$ | $3^{rd}$ | $1^{st}$ | $2^{nd}$ | $3^{rd}$ | $1^{st}$ | $2^{nd}$ | $3^{rd}$ | Total |
| SIPNET | 1278 | 1335 | 1307 | 105 | 843 | 4940 | 2265 | 3898 | 5794 | 21765 |
| ED2 | 26018 | 22380 | 22927 | 249 | 2171 | 7838 | 2207 | 4996 | 7773 | 96559 |

## 3 Results

### 3.1 Test against synthetic data

The test against synthetic data showed that the emulator was able to successfully retrieve the true parameter values that were used in creating the synthetic dataset (Fig. 3). Diagnostics showed that the chains mixed well and converged (all visual and Gelman-Rubin MCMC diagnostics can be accessed via the links provided in the Workflow ID Table A7). As expected, after each round of emulation, posteriors were resolved finer around the true values. Especially the multiplicative bias parameter was only able to resolve in the last round (R3). The posteriors of our "all-at-once" test, where we ran a single emulator proposing all 729 knots at once, compared less well to the true values than the iterative approach. This shows that adaptive refinement of the parameter space exploration is more effective than screening the parameter space with the same (cumulative) number of knots.

### 3.2 Bruteforce vs emulator

Even with the fast SIPNET model, the gain in wall-clock time with emulator was substantial. The three emulator rounds, cumulatively took ∼6 hrs (≈21765 sec, Table 1) while the bruteforce approach took 112 hours. Both metrics (RMSE and deviance) were improved for NEE and LE after calibration with both methods (Table 2). RMSE for SoilResp got worse after calibration with both methods, however this was expected as we informed the model for the shape of the SoilResp flux instead of the absolute magnitude. Indeed, both the deviance metric (which includes the multiplicative bias parameter) and the soil respiration-temperature curve (Fig. 4, bottom panel) improved after calibration with the emulator. However, neither the deviance nor the curve improved after calibration with the bruteforce approach. Overall, the post-PDA ensemble spread was reduced with both methods, while it was narrower after bruteforce-PDA (Fig. 4, A2). This was expected because the emulator includes additional numerical approximation uncertainty in parameter estimates, which propagates into wider confidence intervals in predictions. This can also be seen in the posterior distributions where bruteforce has tighter posterior distributions than the emulator (Fig. 5). The strongest correlations between leaf growth and leaf turnover rate, andf growth and half saturation PAR, soil respiration rate and soil respiration Q10 parameters were also detectable in emulator posteriors (emulator Fig. A3, bruteforce Fig. A4).

**Table 2.** Performance statistics of ensemble means before and after the PDA for both models and output variables. While root-mean-square-error (RMSE) scores evaluate the deviations of model predictions from data, deviance (-2 x log-likelihood) scores evaluate the goodness-of-fit under the assumed data model. For both metrics lower scores are better.

| | | NEE | | LE | | SoilResp | |
|---|---|---|---|---|---|---|---|
| | | pre-PDA | post-PDA | pre-PDA | post-PDA | pre-PDA | post-PDA |
| | $SIPNET_E$ | **140** | **43** | 89 | 79 | **18** | **26** |
| RMSE | $SIPNET_B$ | | **43** | | 77 | | **32** |
| | ED2 | **122** | **68** | 124 | 89 | **29** | **18** |
| | $SIPNET_E$ | 2745 | 976 | 9879 | 8424 | -1333 | -1353 |
| Deviance | $SIPNET_B$ | | 944 | | 8331 | | -1315 |
| | ED2 | 3152 | 1523 | 9914 | 9103 | -1380 | -1390 |

$SIPNET_E$: Emulator PDA. $SIPNET_B$: Bruteforce PDA. RMSE values for LE are in W m-2. RMSE values for NEE and SoilResp are in kgC m-2 s-1 and (bold values) were rescaled by $10^9$ for easier comparison. Deviance values are in log-likelihood units.

20  The effective information content of each data type in the calibration was balanced with autocorrelation correction and effective sample size calculation. The weights of each data after correction can be seen from the deviance values (Table 2). LE and NEE still contribute more to the overall calibration than the SoilResp. After autocorrelation correction, the effective sample sizes for these two data sets were approximately 280 and 51, respectively. For comparison, with uncorrected sample sizes of 7945 and 9426, the deviance values would have been 85357 and -278065 for pre-PDA SIPNET LE and NEE.

### 3.3 ED2 calibration

The emulator calibration for ED2 took ∼27 hrs (≈96559 sec, Table 1). In contrast, a 100K iteration of Metropolis-Hastings MCMC with ED2 would have taken approximately 74 years. Both metrics for all variables showed improvement post-PDA
(Table 2) and their ensemble spread got narrower (Fig. 6). Fitted parametric posterior distributions of ED2 are given in the supplement (Fig. A5, Table A6). In addition, all raw MCMC samples and posterior density distribution plots are available in the respective workflow directories (see Table A7). While all the chains are mixed and converged, the growth respiration factor and fine root allocation scaling factors were less well resolved, indicating that a fourth round might improve their calibration; however, these model outputs were not too sensitive to these parameters (Fig. 2).

Post-PDA ensemble mean of ED2 shows a worse agreement with the NEE and LE data than SIPNET, and a better agreement with the SoilResp (Table 2). However, the time-series plot of the LE for SIPNET (Fig. 4, middle panel) shows that SIPNET largely overestimates the winter moisture fluxes whereas ED2 does not (Fig. 6, middle panel). SIPNET still has an early onset of C fluxes post-PDA whereas ED2 is late to turn off carbon fluxes (top panels). Both pre- and post-PDA ED2 performance for SoilResp were better than SIPNET (bottom panels). ED2 also captures summer diurnal cycle better than SIPNET and both
models were improved after emulator-PDA (Fig. A6)

### 3.4 Emulator scaling

Fig. 7 shows how the emulator method scales with more knots using the 'mlegp' R-package and the trade-off between wall-clock time vs. the approximation error. As expected, the post-PDA ensemble CI approaches to the bruteforce post-PDA CI. In other words, the RCI asymptotically converges to zero, while the clock time to increases with the number of knots (Fig. 7a). The tradeoff between improved model-data agreement (lower deviance values) vs. wall-clock time suggests the more we explore the parameter space (more knots), the lower the deviance gets in general (Fig. 7b). Deviance also lowers with number of parameters targeted in general. However, the best fit was not always to the model with most parameters, and the number of parameters of the best fit varied with the number of knots. With lower number of knots, fewer parameters were well-constrained, but with too few parameters we traded-off the ability to get a good fit. The clock time is largely determined by the number of knots, with much lower sensitivity to the number of parameters as number of knots was much greater than ($\gg$) the number of parameters in this study.

## 4 Discussion

### 4.1 Adaptive sampling design

Our experiment against synthetic data showed that the Gaussian Process model emulator method was able to recapture the true values successfully. While the posteriors of the emulators with few knots (initial round) could be wide, additional rounds of emulator refinement were able to constrain the posteriors better. Our test where we proposed the cumulative number of design points all-at-once showed that, even though we proposed the same number of knots in the end, where you propose those points in the parameter space is important, and iteratively refining the search is a more efficient way of exploring the parameter space. This is because the initial proposal of parameters with LHC had no way of knowing which parts of parameter space are most important to explore, and thus the tails of the distributions end up over sampled and the core undersampled. Furthermore, without multiple iterations the covariances among parameters are also underconstrained, unless informative prior distributions are chosen or previously known covariances are provided. Sampling new knots from the posteriors of the previous iteration informs the algorithm about the posterior means and covariances and allows the GP be refined adaptively. The efficiency of this workflow could potentially be increased further by other adaptive sampling designs, and this remains an important area for further research. For example, Oakley and Youngman (2017) used an initial set of simulator runs to screen-out low likelihood regions to reduce the parameter space before the calibration. For a review of adaptive sampling methods, and emulator design methodologies in general, see Forrester and Keane (2009).

### 4.2 Emulator construction

In this study, we focused on calibrating process-based mechanistic simulators (ecosystem models) using computationally cheaper emulators. Variations of emulator approach are many, and can be found in Jandarov et al. (2014), Aslanyan et al. (2015), Huang et al. (2016), Oakley and Youngman (2017) and the references therein. Here we adopted the version which

emulates the likelihood surface with a Gaussian process, similar to previous studies including applications with a cosmological likelihood function (Aslanyan et al., 2015), a stochastic natural history model (Oakley and Youngman, 2017), the Hartman function and a hydrologic model (Wang et al., 2014) and two land surface models (Li et al., 2018). Our scheme resembles the adaptive surrogate modelling-based optimization (ASMO) approach (Wang et al., 2014; Li et al., 2018) in terms of both the nature of the problem (calibration of a process-based mechanistic simulator) and the general scheme of the calibration algorithm. However, aside from differences in initial sampling designs and error characterizations in these studies, there are two main differences of our scheme from ASMO.

First, we run full MCMC in between the adaptive sampling steps, and on the final response surface, instead of optimization search. Hence, we were able to provide full posterior probability density distribution of the parameters targeted for calibration instead of point estimates of optimum values as Li et al. (2018). The ASMO scheme has also been recently updated for distribution estimation using full MCMC runs (ASMO-PODE) and has been tested with Common Land Model (Gong and Duan, 2017). An important update in our study was that we used the error estimation (variance) provided by the GP model, instead of only using the mean estimates as Gong and Duan (2017) which allowed us to fully propagate the uncertainties to the post-PDA model predictions. Earlier work (not shown) illustrated that failing to propagate the emulator uncertainty (step 5b) results in overconfident posteriors that can easily miss the 'true' parameter in simulated data experiments.

A second addition to our scheme was that we included a further generalization of emulation of the sufficient statistics ($T$) surface. $T$ is, by definition, sufficient to estimate the simulator (process model) parameters in the MCMC. Unlike emulating the likelihood (this study, Oakley and Youngman (2017); Kandasamy et al. (2015)) or the posteriors (Gong and Duan, 2017), emulating $T$ allows us to estimate parameters that are not part of the process model but are part of the statistical data model (the likelihood) as well. In this study, we tested the sufficient statistics emulation for the SoilResp data and updated Gaussian likelihood precision parameter in the MCMC together with other process model parameters. This residual parameter includes both data error and model structural error, and it is not possible to distinguish one from the other with this approach (van Oijen, 2017). However, when we apply the same calibration scheme to different process models at the same site, because the observation error in the data are the same, the difference in the posteriors of this residual parameter (Fig. A7) could give us clues about the model structural errors of models relative to each other, as we demonstrate in this study as a proof-of-concept. However, in our study, use of multiplicative bias parameter further obscures the difference between observation and model structural error.

Indeed, implementation of a more formal way of accounting for model structural error (also called the discrepancy between model output and reality) in our emulator scheme is one of our planned next steps. Explicitly specifying a model discrepancy term and estimating it through MCMC would allow us to account for all sources of model predictive uncertainty (van Oijen, 2017). However, determining the expected form of discrepancy in order to learn about model parameters realistically could be difficult due to lack of mechanistic knowledge of the underlying processes (Brynjarsdóttir and O'Hagan, 2014). In that sense, accounting for discrepancy in model calibration is not an emulator approach specific issue. For a novel approach investigating model structural uncertainty through a modular modeling framework see Walker et al. (2018), which could be useful for modeling prior knowledge about discrepancy in ecosystem models in the future. Because of the unknowns about the discrepancy

functions, it is common to use Gaussian processes to model the discrepancy (Kennedy and O'Hagan, 2001). Even then, only with realistic prior constraints about the process, calibrated model predictions would be unbiased (Brynjarsdóttir and O'Hagan, 2014). For an example of addressing discrepancy in calibration that combines likelihood-emulation approach with importance sampling, see Oakley and Youngman (2017) where they inflated simulator uncertainty to account for simulator discrepancy instead of explicitly specifying a prior for it in order to make the likelihood tractable. When likelihood function becomes intractable or a sufficient statistic does not exist, techniques using likelihood-free inference (Gutmann and Corander, 2016) or computing approximately sufficient statistics could also be a remedy (Joyce and Marjoram, 2008).

Finally, the scheme used in this study is also compatible with various adaptive sampling designs (other than LHC), emulator models (other than GP), and MCMC algorithms (other than adaptive Metropolis-Hastings) like the ASMO-PODE scheme (Gong and Duan, 2017).

## 4.3 Bruteforce vs emulator

Both bruteforce and emulator methods reduced the uncertainty around the model predictions when real data was assimilated
with SIPNET. Bruteforce posteriors resolved finer than the emulator as expected due to the numerical approximation error in the emulator. Therefore, when computational time allows, bruteforce methods will result in more precise posteriors and are preferred over the emulator method. However, when the model run time or the volume of data to be assimilated does not allow running long MCMC iterations, it is possible to constrain parameters in orders of magnitude less time, with far fewer model evaluations, and with much greater parallelization using the emulator method. This speed-up puts model calibration within
reach for large, computationally-challenging models that are currently underconstrained.

In addition to just fitting the model, emulators make it practical to implement different hypotheses within a model, recalibrate the model, and test them against data repeatedly. Furthermore, emulators make it possible to calibrate complex models hierarchically, which would not be computationally feasible otherwise as hierarchical Bayesian modeling involves calibrating models many times at multiple spatial/temporal/experimental settings. For example, it is a known issue that site-level cali-
brations are not easily transferable to new sites or to larger scales (Post et al., 2017). In that sense, Hierarchical Bayesian approach is an important improvement over classical Bayesian model calibrations because it formally accounts for the spatial and temporal variability of ecosystems and provides a structure that will help us better understand the uncertainties involved at different levels of our study systems (Clark, 2005; Thomas et al., 2017).

## 4.4 Autocorrelation correction and multiple data constraints

A lack of independence in observation errors causes overfitting of the model parameters and underestimate prediction uncertainties (Ricciuto et al., 2008). It is not uncommon for calibration against one data set that is given a high weight (e.g. many more observations) to cause other model outputs to perform worse. Indeed, in our calibration study, model-data agreement for NEE improved while it was reduced for the SoilResp variable after the bruteforce calibration. The most common approaches to this problem involve arbitrary weights or *ad hoc* solutions to rebalance the influence of data. We addressed this issue with a
novel approach of explicitly modeling autocorrelation, which provides a more objective and statistically rigorous approach to

balancing the weights of different data. Although, the NEE and LE data still influenced the calibration more than the SoilResp data, assimilating multiple data streams and balancing their influence was important. For example, NEE is a result of both primary production and respiration processes, and the model outputs were sensitive to parameters involved in both of these processes. If we were to assimilate only NEE, estimated parameters contributing to NEE might have compensating errors (Post et al., 2017). However, including an additional constraint on model parameters contributing to either primary production or respiration could help us distinguish such compensation effects. Altogether, over-fitting of models is a common problem in Bayesian calibration, and both the autocorrelation correction and the use of the emulator method practically proved to be a

helpful strategy. Lastly, the effect of number of assimilated data streams on emulator performance is not explicitly tested in this study, however, calibration performance of the emulator should still be proportional to bruteforce with more or less data streams. For studies that inspect the effect of assimilating multiple data streams on model calibration performance see Keenan et al. (2013) and MacBean et al. (2017).

## 4.5   Scaling factors

In the calibration of ED2, instead of constraining the PFT parameters directly, we targeted scaling factors (SFs) for parameters that are common among PFTs which reduces the dimensionality considerably (i.e. instead of targeting $N_{parameters}$ x $M_{PFTs}$, we only target $N_{parameters}$). This experiment showed that the emulator method with SFs could constrain ED2 PFT parameters and improve model predictions. However, this approach assumes that the *relative* differences among PFTs are approximately correct, but that overall processes may be miscalibrated, and thus that the more likely parameter space for different PFTs will be

in the similar regions of their prior distributions. For example, if a density dependent mortality parameter is being targeted, the prior distributions for an early and a late successional type can be defined to represent their differentiation so that the posteriors would still be different when using the SF. In our study, PDA priors for each PFT were informed by meta-analysis, therefore accommodating for such differences amongst PFTs. By contrast, the SF approach by itself cannot, for example, converge on values in the first quartile for a certain parameter space for one PFT and in the third quartile for another PFT. We note that, the

SF approach is not specific to the emulator method, and could also be used with bruteforce algorithms to reduce dimensionality.

## 4.6   Approximation error vs clock-time

The emulator method we propose overcomes many hurdles in the Bayesian calibration of ecosystem models, especially in terms of computation time. The main cost of running the full model sequentially for the MCMC is avoided in the emulator approach, and the initial set of runs (or the iterative batches of runs) can be parallelized. Algorithms like Sequential Monte Carlo (or

Particle Filter) provide a partial solution since they allow parallelization, but they often require even larger number of model evaluations than a typical MCMC, particularly for higher dimensional problems (Arulampalam et al., 2002). Nevertheless, dimensionality can still be a problem for the emulator method as more knots will be needed to resolve the predicted surface as the number of parameters to be constrained increases. Our scaling experiment indicates that RCI decays quickly and starts leveling-off as the number of knots increases. In other words, one can stop increasing the number of knots at a stage where the

gain in terms of approximation error reduction being heavily traded-off with clock time is reached. Detecting such thresholds is feasible in practice if the emulator is refined iteratively.

    A similar threshold was also apparent for overall model calibration ability. While the gain, if any, in model improvement in terms of deviance was minimal from 480 to 960 knots, the clock time required was more than doubled in our scaling experiment. This experiment also suggested that the number of model parameters we chose to constrain was an adequate choice for our setting. Targeting a few additional model parameters did not result in substantial differences in terms of overall deviance, which was expected as the targeted parameters were chosen according to their contribution to the overall model uncertainty. Thus we are confronted with the fundamental trade-off where increasing the number of parameters requires that we need to propose

more knots to explore the parameter space, which increases runtime, and at some point these additional parameters provide diminishing returns. Understanding this trade-off is greatly facilitated by performing an uncertainty analysis before calibration, which allows parameters to be added to the calibration in order of their contribution to model uncertainty. Finally, we note that the shape of the clock time vs deviance trade-off curves will vary by model as they varied by number of model parameters.

    To fit the Gaussian process models in this study, we used the $mlegp$ R-package which was found to be performing well with

its default settings (Erickson et al., 2018). The comparison by Erickson et al. (2018) shows that there are faster (such as $laGP$) and computationally more stable (such as $GPfit$) R-packages available. However, $laGP$ performs worse than $mlegp$ unless thousands of design points are provided, and $GPfit$ is substantially slower than $mlegp$ as it is solely written in R whereas $mlegp$ is pre-compiled in C. Finally, other packages from other platforms (such as the GPy and scikit-learn modules of Python) could outperform $mlegp$ (Erickson et al., 2018), however, as PEcAn is mainly written in R, $mlegp$ was an adequate choice for

our workflow. Overall, the approximation error vs clock-time trade-off is not independent of the software/code used to fit the Gaussian process model.

    In this study, we tested emulator calibration with number of parameters that are comparable to previous studies with biosphere models, if not higher (Ray et al., 2015; Huang et al., 2016; Gong and Duan, 2017). However, running the emulator can also become infeasible. For example, with the current scheme calibrating 100 parameters would not be possible with $100^3$

knots, as $O(N^3)$ floating point operations needed for the Cholesky decomposition in GP would exceed memory and wall clock time capacities. That said, the $p^3$ scheme is just the rule-of-thumb that we employed in these experiments, and not an inherent limit of the emulator approach itself. The calibration of 100 parameters might be possible with much smaller number of knots ($\ll 10^6$) depending on the model. Using a sample size about 10 times (n=10d) the input dimension is a common recommendation in computer experiments with GP (Loeppky et al., 2009). But this is considered to be too small for most of the cases and

using 20 times (n = 20d) larger sample sizes are suggested instead (Erickson et al., 2018). Indeed, our scaling experiment also suggests calibrating the model with fewer knots ($< p^3$) would be possibe. In practice, we would advocate for performing an uncertainty analysis to reduce the dimensionality of the problem. In addition, the data would need to be strong enough to actually constrain such large number of parameters. Still, when dimensionality becomes too large, alternative emulators could be explored, such as the Nearest-Neighbor Gaussian Process model (which takes advantage of the fact that the nearest neighbors

contribute the most information while fitting the GP model, and could help reduce computational costs substantially for bigger datasets and much larger number of parameters Datta et al. (2016)).

## 5  Conclusions

Here we introduced a framework that addresses both the computational and statistical challenges of Bayesian model calibration. We introduced a number of novel approaches, such as: building an emulator on the sufficient statistics surface; an autocorrelation correction on the latent time series estimated through a state-space model; and introducing of a scaling factor to reduce dimensionality across PFTs. We also standardized and generalized this framework in an open source ecological informatics toolbox, PEcAn, for repeatability and use with other ecosystem models.

Our study furthers efforts toward reducing model uncertainties showing that the emulator method makes it possible to efficiently calibrate complex models. Here we demonstrated examples and evaluated performances with terrestrial ecosystem models but the application can be generalized to any "big model". Overall, this efficient data assimilation method allows us to conduct more calibration experiments in relatively much shorter times, enabling constraining of numerous models using the expanding amount and types of data.

*Code availability.*  All the code used in this study can be found at github.com/PecanProject/

*Competing interests.*  Authors declare no competing interests

*Acknowledgements.*  IF was funded by grants to MCD from National Science Foundation (NSF) Macrosystems grants (#1318164, #1241891) and NASA Terrestrial Ecosystems. We thank the PEcAn Project Team for helpful discussions and the cyberinfrastructure. The PEcAn project is supported by the NSF (ABI #1062547, ABI #1458021, DIBBS #1261582), NASA Terrestrial Ecosystems, the Energy Biosciences Institute,
and an Amazon AWS in Education Grant. Research at the Bartlett Experimental Forest is supported by the USDA Forest Service's Northern Research Station. We acknowledge additional support from the National Science Foundation (DEB #1114804), and from the Northeastern States Research Cooperative, and DOE NICCR grant DE-FC02-06ER64157 to PRM. We also thank Dr. Florian Hartig and PROFOUND TG14 group for early access to BayesianTools package, Dr. David Cameron and PROFOUND TG15 group for helpful discussions. We are greateful to Dr. Marcel van Oijen, the anonymous reviewers, and the Biogeosciences editor Dr. Sönke Zaehle for their comments and suggestions that helped us improve this paper to a great extent.

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

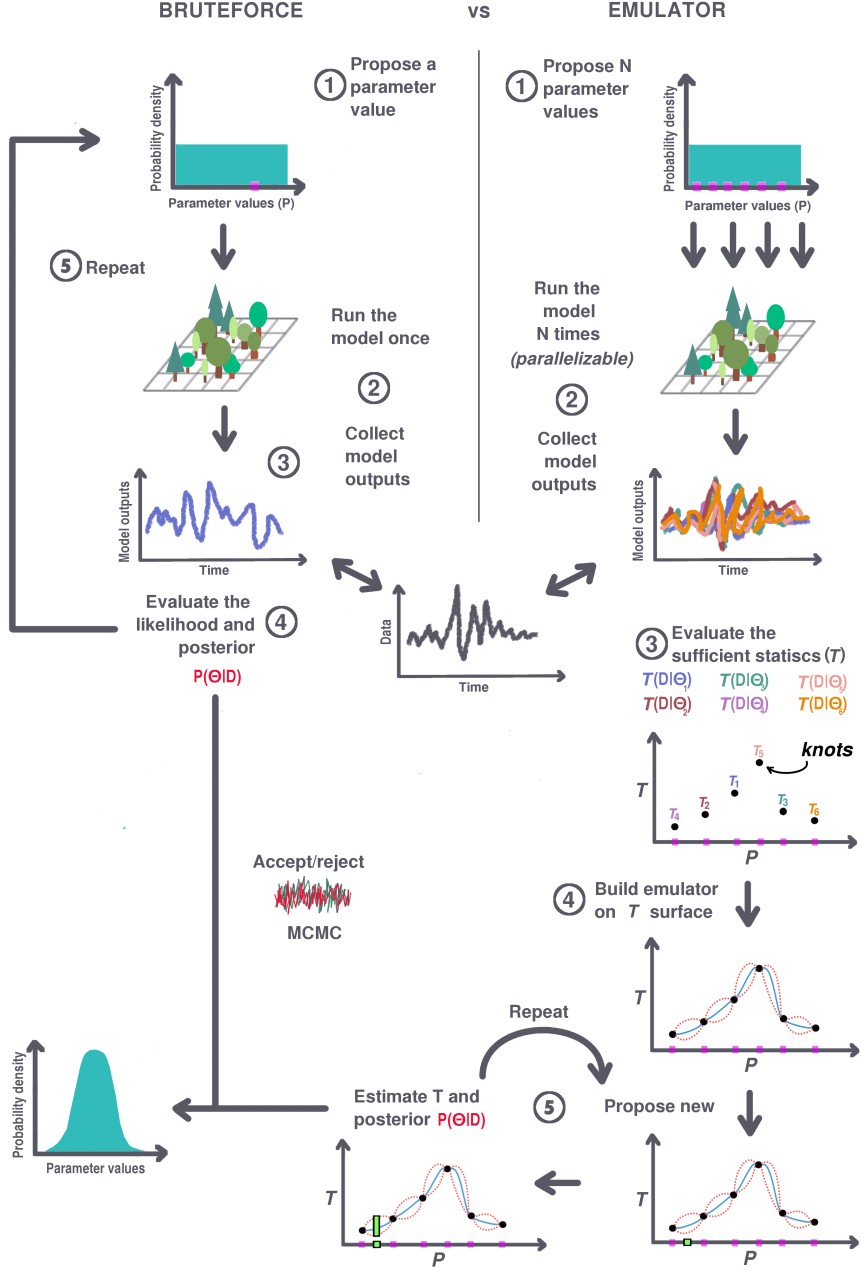

**Figure 1.** Comparison of bruteforce and emulator approaches for a univariate example. The computationally costly step of running the model is parallelizable for the emulator, whereas in the bruteforce approach it needs to be run at every MCMC iteration sequentially. Emulator is built on the pairs of the initial parameter set (pink points on x-axis; P) and the sufficient statistics ($T$) values on the y-axis. These design points in the P-T space, or knots (black dots) are obtained by evaluating the full model. Next, a Gaussian statistical process is fitted (blue solid line) with error estimates for prediction (red dashed lines). Once the emulator is constructed, a new parameter value will be proposed (green box on the x-axis). Finally, values that the response variable can take (green segment) given the newly proposed parameter will be estimated using the emulator.

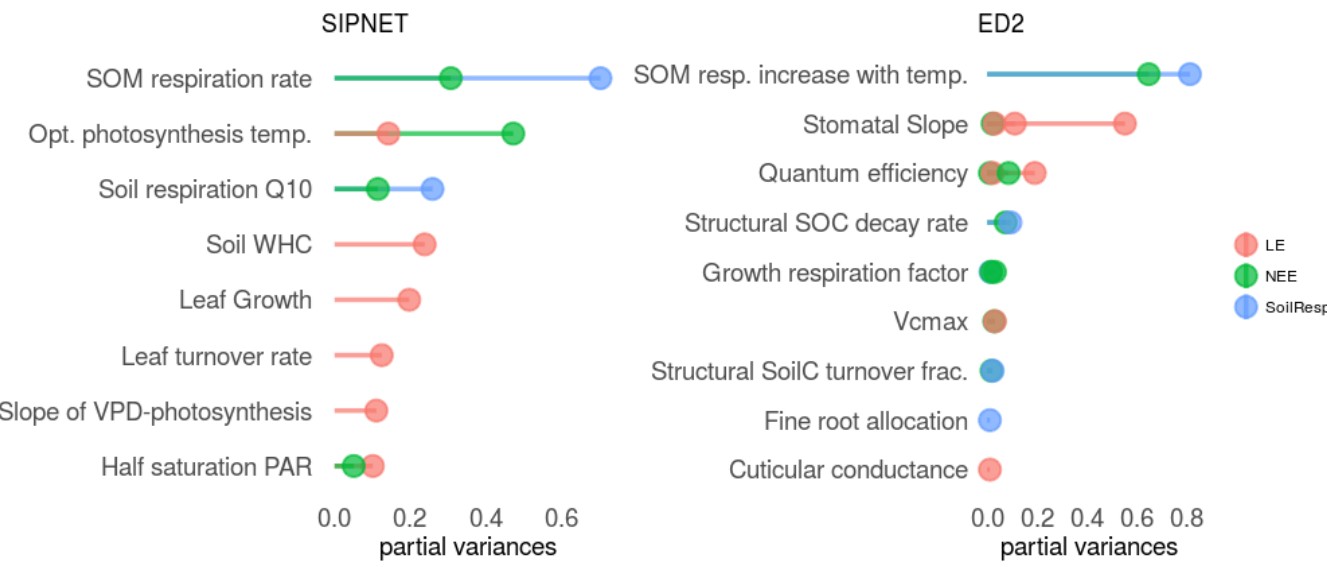

**Figure 2.** Results of uncertainty analysis in PEcAn for plant physiological and soil biogeochemistry parameters of SIPNET (left) and ED2 (right). The longer the bar the more that parameter contributes to the model predictive uncertainty. The parameters shown above that contribute more than 5% (1%) uncertainty were chosen to target in calibration of SIPNET (ED2) and are shown above.

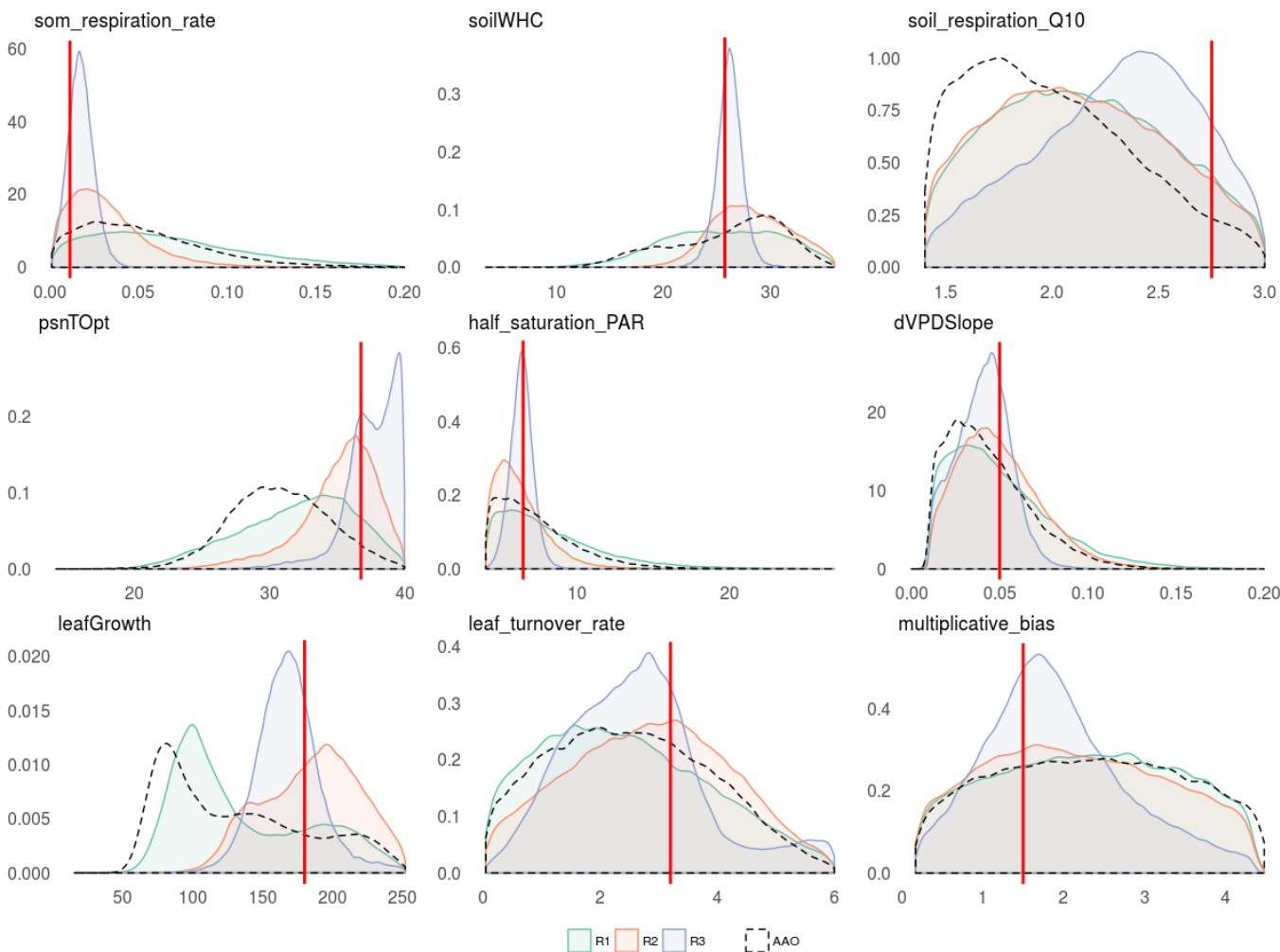

**Figure 3.** Emulator performance against synthetic data. Red vertical line represents the true parameter values that were used to create the synthetic dataset. Shaded distributions are the posteriors obtained after each emulator rounds. Dashed lines are the posteriors after a single emulator (all-at-once, AAO) round built with a total number of knots of all rounds (729 knots) instead of refining the emulator iteratively ($1^{st}$ round 243, $2^{nd}$ round 486, $3^{rd}$ round 729). All priors were uniform for these parameters, except the multiplicative bias parameter.

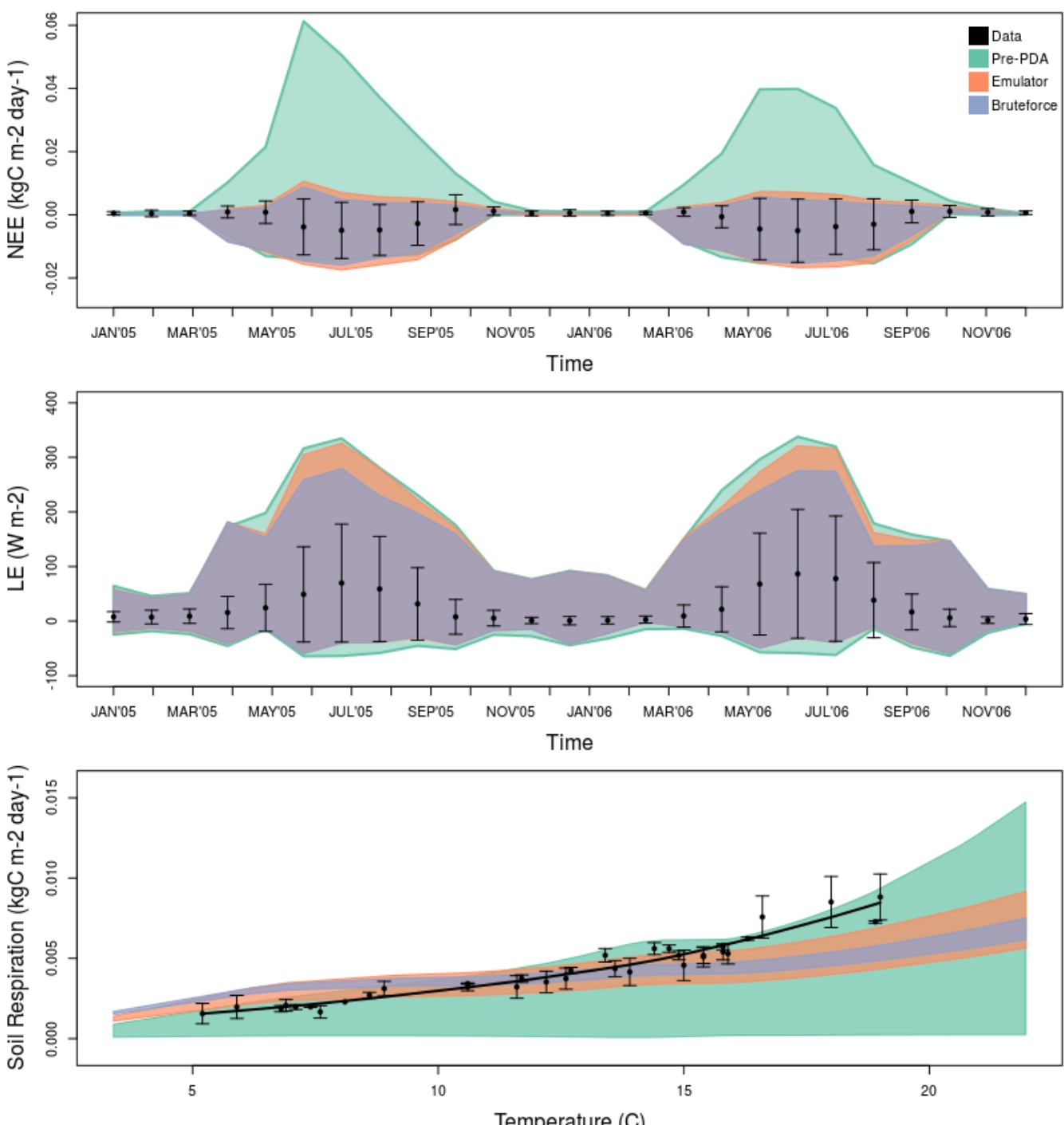

**Figure 4.** SIPNET performance against real data (black dots) after emulator (orange polygon) vs bruteforce (blue) calibration. The pre-PDA ensemble spread (green) was wider for all variables and reduced with both methods. (a) and (b) are monthly-smoothed time series (for unsmoothed version please see Fig. A1), while (c) shows the temperature - soil respiration response curve, plotted with locally weighted scatterplot smoothing (LOESS) line, and residuals from a fitted temperature response function as a conservative estimate of the error bars. All polygons show the 2.5% - 97.5% CI.

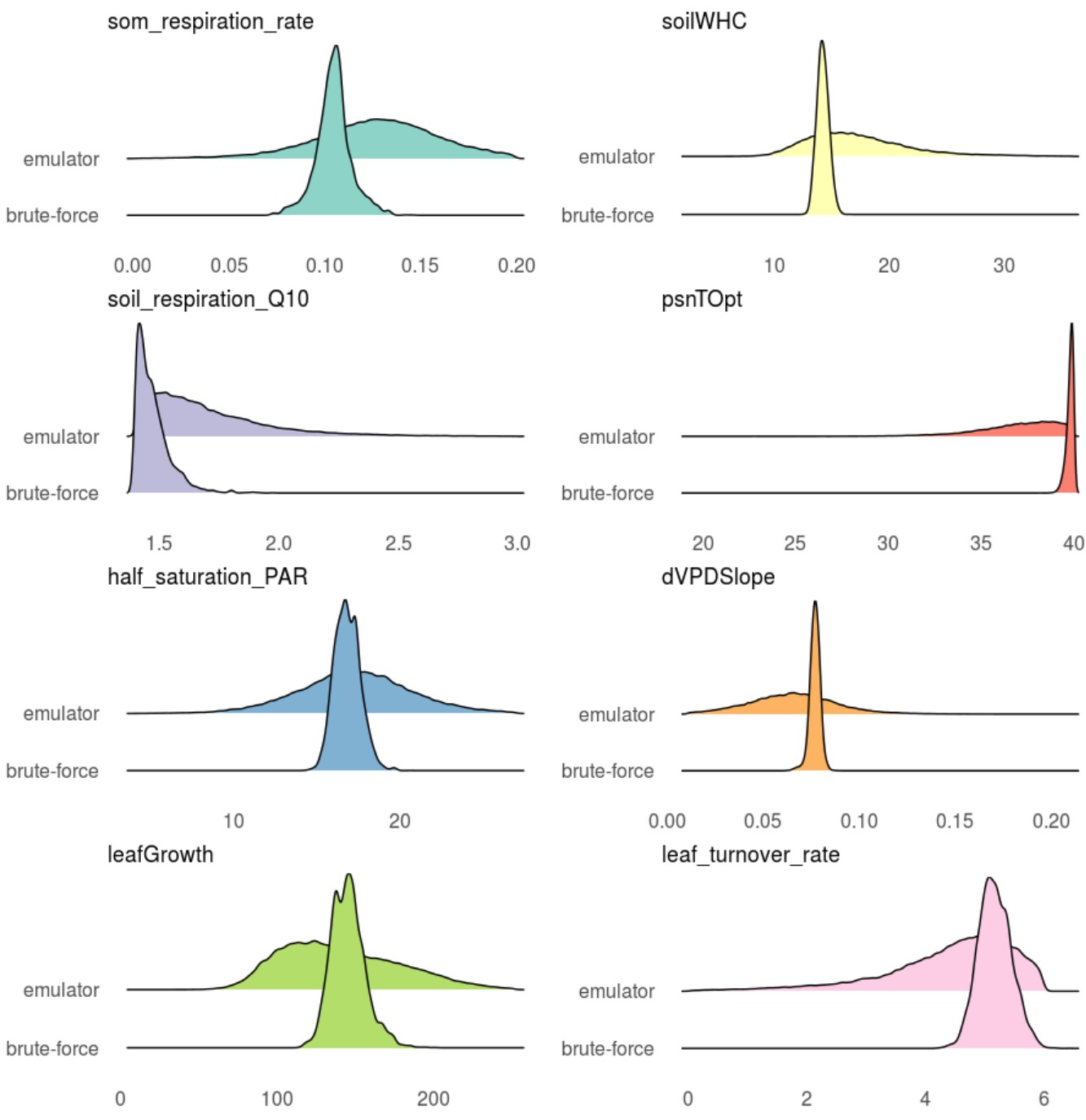

**Figure 5.** Posteriors from emulator vs bruteforce approach with SIPNET after calibration against real-world data.

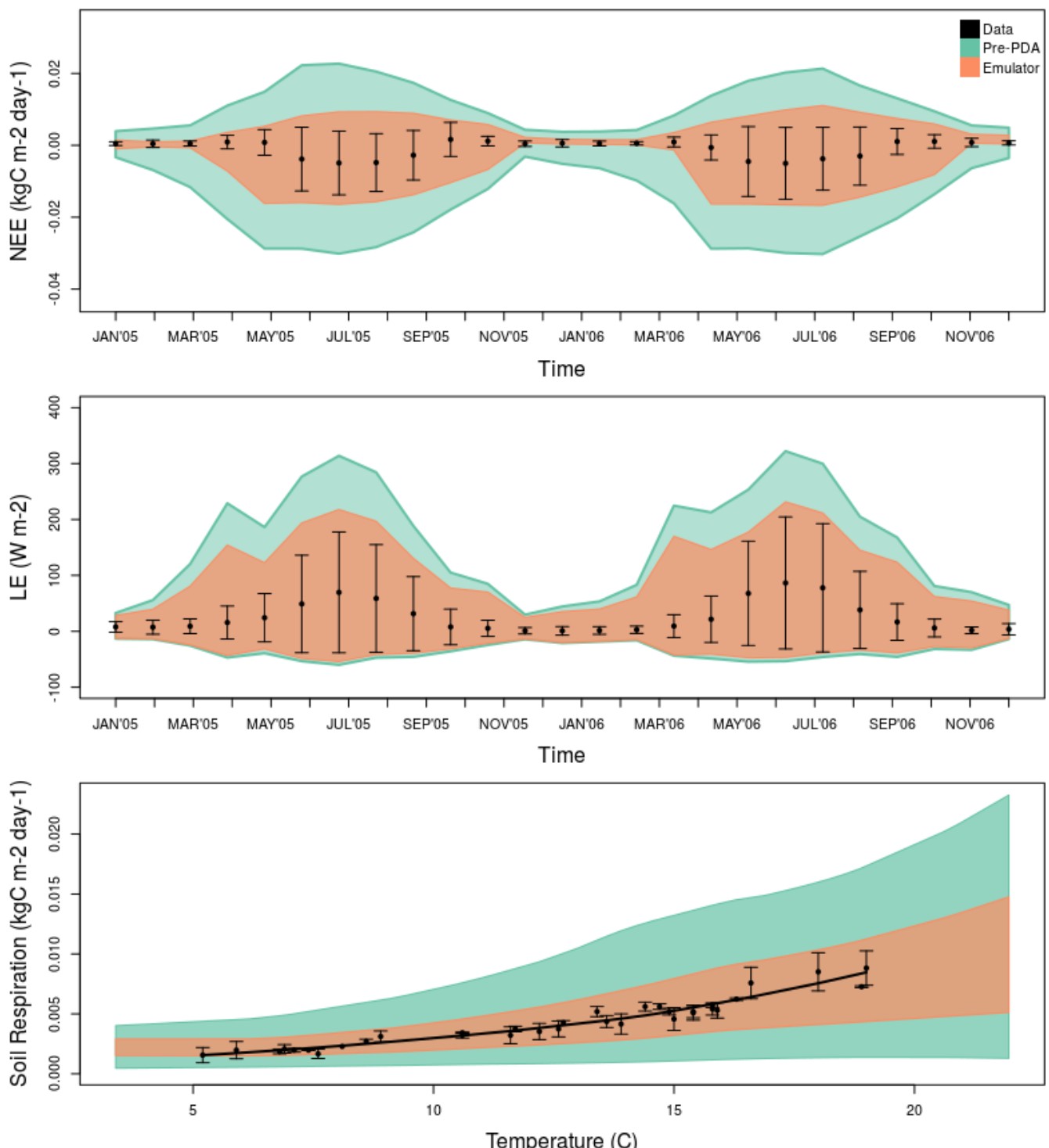

**Figure 6.** Pre-PDA vs post-PDA ED2 performance against real-world data. Panels and colors are same as Figure 4.

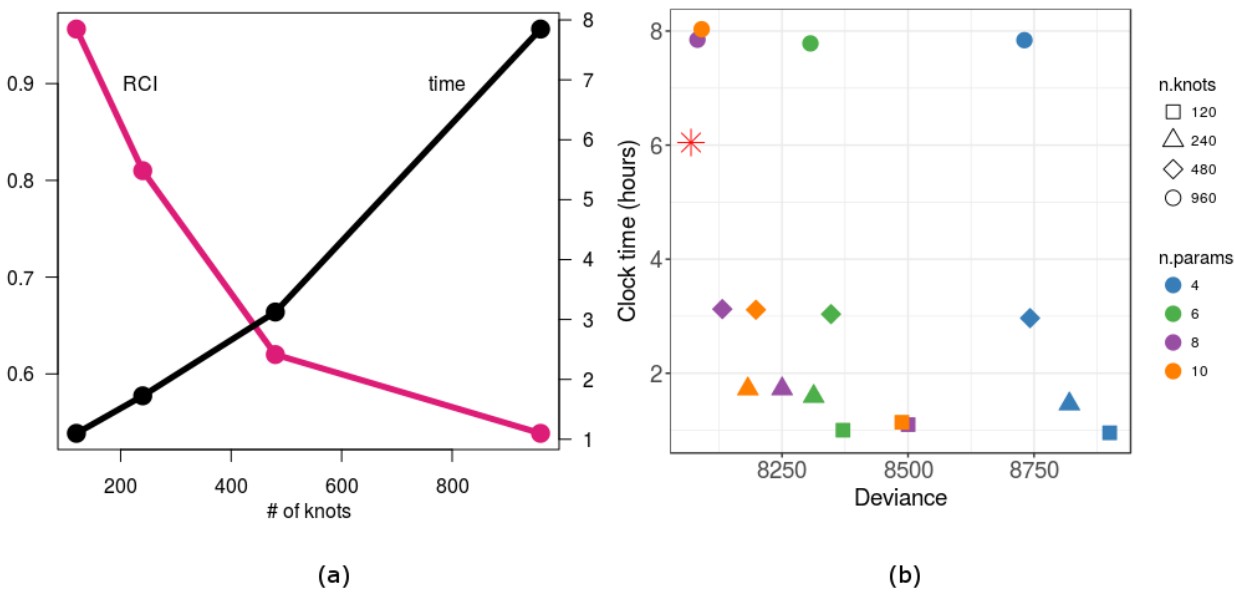

**Figure 7.** Results of the scaling experiment. (a) Trade-off between wall-clock time vs. the approximation error (relative confidence interval, RCI) with increasing emulator knots. (b) The tradeoff between improved model-data agreement vs. wall-clock time. Red star is the emulator design followed in this study for SIPNET with 8 model parameters and 729 knots. Underlying data for (b) can be found at Table A9.

## Appendix A

### A1  Study site

Bartlett Experimental Forest (44° 17′ N, 71° 03′ W) is a US Forest Service research forest located outside of Bartlett, NH in the White Mountains (Lee et al., 2018). Species composition is typical of northern hardwood forests and consists predominantly of *Acer rubrum* (red maple), *Fagus grandifolia* (American beech), *Betula papyrifera* (paper birch), and *Tsuga canadensis* (eastern hemlock). Climate is also typical of central New England with short summers (20 °C) and long cold winters (-8 °C). The site is generally moist, receiving approximately 1300 mm/yr of precipitation. Soils are sandy loam Spodosols and can become saturated during spring snowmelt.

An eddy-covariance tower (26.5m) was installed in November 2003 at a lowland site (272m) within the experimental forest. Topography near the eddy-covariance tower is flat to gently sloping but larger hills (1-3 km distant) surround the site. Canopy height is 19m with a mean stand age of approximately 100 yr. The eddy-covariance system consists of a LI-6262 $CO_2$/$H_2O$ infra-red gas analyser (LiCor, Lincoln, NE) and SAT-211/3K 3-axis sonic anemometer (Applied Technologies, Longmont, Colo.). Measurements were made at 5 Hz and fluxes were estimated every 30 minutes. The meteorological data used in this analysis were derived from measurements made at the eddy-covariance tower for years 2005-2006. These include air temperature above the canopy (22.3 m), soil temperature, relative humidity, precipitation, above canopy PAR and wind speed.

The Bartlett tower footprint contains twelve vegetation inventory plots that follow the Forest Inventory and Analysis (FIA) design consisting of four circular 10 m radius subplots: one central and three evenly spaced at a radius of 36.5 meters. Vegetation plots were established in May 2004 and used to initialize ED2. Bradford et al. (2010) provided soil carbon and live aboveground biomass estimates for Bartlett which we used to initialize SIPNET.

Soil respiration measurements were made manually in each plot (n=12) at permanently installed rings that are 10cm in diameter using a soil CO2 flux chamber (LiCOR 6400-9). Soil temperature and moisture were measured concurrently using a soil temperature probe and a TDR probe. During 2006, soil respiration censuses were made approximately every 4-5 days from day 138 to day 325 for a total of 39 chamber censuses.

### A2  SIPNET model

The simplified Photosynthesis and Evapotranspiration model (SIPNET) is a simple ecosystem model which can be used to interpret carbon water exchange between vegetation and the atmosphere. SIPNET has been developed from the PnET family of models to facilitate model comparisons to flux towers (Braswell et al., 2005; Sacks et al., 2006). SIPNET runs at a half-hourly time step. It represents relatively few processes (has two vegetation carbon pools, a single aggregated soil carbon pool, and a simple soil moisture sub-model), making it easier to evaluate which data contributes how much to the parameterization of each process. As a result of this setup, SIPNET is a fast model ($\sim$ 5.5 sec per MCMC iteration in PEcAn including model execution, and writing and reading model outputs), which makes it suitable for application of bruteforce methods.

Forest inventory data collected in the tower footprint were used to set initial conditions in SIPNET. We fitted Bayesian models using the allometric equations available in the literature (Jenkins et al., 2004) to estimate the aboveground biomass

**Table A1.** Initial state values used for SIPNET runs.

| Pool | Value | Units |
|---|---|---|
| Above- and below-ground woody biomass | 9600 | gC / m$^2$ ground area |
| Initial leaf area | 0 | m$^2$ leaves / m$^2$ ground area |
| Litter biomass | 200 | gC / m$^2$ ground area |
| Soil biomass | 1600 | gC / m$^2$ ground area |

**Table A2.** The prior and posterior distributions of the constrained SIPNET parameters.

| Parameter | Prior | Posterior (Emulator) | Posterior (Bruteforce) |
|---|---|---|---|
| SOM respiration rate | unif(0.001, 0.3) | weibull(1.62, 0.13) | norm(0.1, 0.009) |
| Soil Respiration Q10 | unif(1.4, 3.0) | lnorm(0.697, 0.24) | lnorm(0.39, 0.046) |
| Soil WHC | unif(0.1, 36.0) | lnorm(2.95, 0.31) | lnorm(2.7, 0.035) |
| Half saturation PAR | unif(4.0, 27.0) | weibull(3.74, 17.5) | lnorm(2.8, 4.5e-02) |
| dVPDSlope | unif(0.01, 0.25) | weibull(2.26, 7e-02) | norm(0.08, 2.6e-03) |
| Seasonal leaf growth | unif(0.0, 252.0) | norm(150.6, 46.8) | norm(145, 10.8) |
| psnTOpt | unif(5.0, 40.0) | norm(12.07, 35.7) | weibull(336, 39.9) |
| Leaf turnover rate | unif(0.03, 6.0) | norm(5.14, 1.9) | lnorm(1.64, 5e-02) |

at Bartlett through PEcAn's allometry module. These values were in agreement with live aboveground biomass estimates by Bradford et al. (2010) whose soil carbon pool estimates were also used to set the initial values in our SIPNET runs (Table A1).

## 5   A1   Ecosystem Demography Model

The Ecosystem Demography model version 2.1 (ED2) is a terrestrial biosphere model that couples plant community dynamics to biogeochemical models of associated soil fluxes of carbon, water, and nitrogen (Moorcroft et al., 2001; Medvigy et al., 2009). ED2 is explicitly designed to scale from the individual to the region and to account for community processes, such as disturbance and resource competition, in a manner analogous to forest gap models. ED2 achieves this with a size and age structured (SAS) approximation to a forest gap model which accounts for the vertical size distribution within a stand/patch and the distribution of different stand ages across the landscape. This hierarchical SAS allows ED to be compared to data operating at multiple scales but in practice this means that a single ED run will simulate a large number of different patches, each with a number of trees of different sizes and species. The resulting computational expenses and complexity of drivers and outputs make ED2 an ideal example of the challenges of model-data fusion. The initialization of vegetation and soil for ED2 was done using the same forest inventory data and soil carbon measurements described for SIPNET. The species occurring in the inventory data were mapped to ED2 PFTs following Dietze and Moorcroft (2011).

**Table A3.** Calibrated SIPNET parameters and the 'true' values used to produce the synthetic data.

| Parameter | Definition | Units | True Values |
|---|---|---|---|
| SOM Respiration rate | Soil organic matter respiration rate coefficient | $Day^{-1}$ | 0.01 |
| Optimum photosynthesis rate | Optimum temperature for photosynthesis | Celcius | 36.75 |
| Soil Respiration Q10 | Scalar determining effect of temperature on soil heterotrophic respiration | ratio | 2.75 |
| Soil WHC | Soil water holding capacity | cm | 25.75 |
| Seasonal leaf growth | Amount of leaf growth following leaf-out | $gC / m^2$ | 180 |
| Leaf turnover rate | Average turnover rate of leaves | $y^{-1}$ | 3.2 |
| Slope-VPD | Slope of VPD-posthesis relationship | $kPa^{-1}$ | 0.05 |
| Half saturation PAR | Photosynthetically active ratioan at which photosynthesis occurs at 1/2 theoretical minimum | Einsteins $m^{-2}$ $day^{-1}$ | 6.46 |
| Multiplicative bias | Soil respiration scaling constant | unitless | 1.5 |

**Table A4.** Calibrated ED2 parameters.

| Parameter | Definition | Units |
|---|---|---|
| Stomatal slope | Slope of relatin between stomatal conductance and A | ratio |
| Quantum efficiency | Efficiency with which light is converted into fixed carbon | fraction |
| Vcmax | Maximum rubisco carboxylation capacity | umol CO2 m-2 s-1 |
| Cuticular conductance | Leaf (cuticular) conductance when stomata fully closed | umol H2O m-2 s-1 |
| Growth respiration factor | Proportion of daily carbon gain lost to growth respiration | fraction |
| Fine root allocation | Ratio of fine root to leaf biomass | ratio |
| r_stsc | Fraction of structural pool decomposition going to heterotrophic respiration | fraction |
| Decay rate stsc | Intrinsic decay rate of structural pool soil carbon | 1/day |
| Resp. temperature increase | Determines how rapidly heterotrophic respiration increases with increasing temperature | 1/K |
| Multiplicative bias | Soil respiration scaling constant | unitless |

**Table A5.** The PDA prior (meta-analysis posterior) approximated parametric distributions of the targeted ED2 parameters.

Plant Functional Type Physiological Parameters

|  | t.EH | t.LC | t.LH | t.NMH | t.NP |
|---|---|---|---|---|---|
| stomatal slope | gamma(19.7, 2.97) | weibull(2, 10) | weibull(2, 10) | weibull(2, 10) | weibull(2, 10) |
| quantum efficiency | gamma(16.6, 279) | norm(0.08, 0.014) | weibull(2.9, 0.07) | lnorm(-3.28, 0.08) | gamma(82, 1.4e+03) |
| Vcmax | norm(74.9, 9.8) | weibull(1.7, 80) | norm(60.5, 11.9) | gamma(37.8, 0.53) | weibull(2.2, 80) |
| cuticular conductance | lnorm(9.4, 0.7) | lnorm(9.4, 0.7) | lnorm(9.4, 0.7) | norm(9988, 497) | lnorm(9.4, 0.7) |
| growth respiration factor | beta(4.06, 7.2) | beta(2.63, 6.52) | beta(4.06, 7.2) | beta(2.63, 6.52) | beta(2.63, 6.52) |
| fine root allocation | gamma(16.59, 23.32) | lnorm(-0.25, 1) | gamma(9.13, 8.22) | gamma(9.44, 8.82) | lnorm(-0.25, 1) |

Soil Biogeochemistry (decomposition) parameters

| r_stsc | beta(1, 1) |
|---|---|
| decay rate stsc | unif(0.005, 0.75) |
| resp. temperature increase | unif(0.05, 0.2) |

t.EH: temperate Early Hardwood, t.LC: temperate Late Conifer, t.LH: temperate Late Hardwood, t.NMH: temperate North Mid-Hardwood, t.NP: temperate Northern Pine

**Table A6.** The emulator-PDA approximated parametric posterior distributions of the targeted ED2 parameters.

Plant Functional Type Physiological Parameters

|  | t.EH | t.LC | t.LH | t.NMH | t.NP |
|---|---|---|---|---|---|
| stomatal slope | lnorm(1.48, 0.13) | gamma(4.01, 1.6) | gamma(4.01, 1.6) | gamma(4.01, 1.6) | gamma(4.01, 1.6) |
| quantum efficiency | lnorm(-2.8, 0.11) | norm(0.08, 6.3e-03) | gamma(35.8, 541) | lnorm(-3.3, 0.04) | lnorm(-2.8, 0.05) |
| Vcmax | norm(47.3, 3.45) | gamma(2.83, 1.04) | norm(27.1, 4.17) | norm(42.9, 2.85) | weibull(2.4, 6.4) |
| cuticular conductance | norm(9.85, 0.385) | norm(9.85, 0.385) | norm(9.85, 0.385) | norm(10308, 273) | norm(9.85, 0.385) |
| growth respiration factor | beta(3.59, 7.47) | beta(2.29, 6.8) | beta(3.59, 7.47) | beta(2.29, 6.8) | beta(2.29, 6.8) |
| fine root allocation | gamma(30.7, 7.47) | lnorm(-0.3, 0.73) | gamma(16.7, 15.6) | gamma(17.3, 16.8) | lnorm(-0.3, 0.73) |

Soil Biogeochemistry (decomposition) parameters

| r_stsc | beta(1, 1.98) |
|---|---|
| decay rate stsc | lnorm(-2.97, 1.02) |
| resp. temperature increase | lnorm(-2.16, 0.28) |

t.EH: temperate Early Hardwood, t.LC: temperate Late Conifer, t.LH: temperate Late Hardwood, t.NMH: temperate North Mid-Hardwood, t.NP: temperate Northern Pine

**Table A7.** Links to the Workflow IDs. The input/output files associated with each workflow can be accessed via the history table on the following link "pecan2.bu.edu/pecan/history.php". Or each workflow can be accessed directly by replacing the workflow ID at the end of the following link: "pecan2.bu.edu/pecan/08-finished.php?workflowid=1000008379". The left frame on the page can be used to navigate through PEcAn settings, input and output files. If you wish to conduct further visualizations or analysis on the MCMC samples, you can first select the "mcmc.list.pda***.Rdata" file (*** being the ensemble IDs given by the workflow) under the "PEcAn Files" dropdown menu on the left frame. By clicking "Show File" button you can download the raw MCMC outputs to your own machines. If you would like to display posterior density distributions, first select either soil or plant physiology under the "PFTs/PFT" menu on the left frame. Next, under the "PFTs/Output" dropdown menu, select "posteriors.pda.***.pdf" files and click "Show PFT Output". The red line would be the posterior density plot and the black line would be the approximated parameteric distributions (such as the ones reported in Table A2 and A6) fitted by PEcAn's approx.posterior function that can be found under pecan/modules/meta.analysis/R/approx.posterior.R

| Model | Experiment | Workflow ID |
|---|---|---|
| SIPNET | Pre-PDA EA/UA | 1000008379 |
| SIPNET | Emulator PDA - Synthetic Data | 1000009295 |
| SIPNET | Emulator PDA - Real Data | 1000009249 |
| SIPNET | Emulator Post-PDA EA | 1000009309 |
| SIPNET | Bruteforce PDA - Real Data (chain 1) | 1000008530 |
| SIPNET | Bruteforce PDA - Real Data (chain 2) | 1000008531 |
| SIPNET | Bruteforce PDA - Real Data (chain 3) | 1000008532 |
| SIPNET | Bruteforce Post-PDA EA | 1000008923 |
| ED2 | Pre-PDA EA/UA | 1000009051 |
| ED2 | Emulator PDA - Real Data | 1000009052 |
| ED2 | Emulator Post-PDA EA | 1000009052 |

PDA: Parameter Data Assimilation, EA: Ensemble Analysis, UA: Uncertainty Analysis

**Table A8.** Links to the Workflow IDs of scaling experiments. Parameters targeted are in this order cumulatively: som_respiration_rate, soil_respiration_Q10, soilWHC, psnTOpt (4), leafGrowth, leaf_turnover_rate (6), half_saturation_PAR, dVPDSlope (8), AmaxFrac, dVpdExp (10)

| Model | # of params | # of knots | Workflow ID |
|-------|-------------|------------|-------------|
| SIPNET | 4 | 960 | 1000009310 |
|        |   | 480 | 1000009311 |
|        |   | 240 | 1000009312 |
|        |   | 120 | 1000009313 |
| SIPNET | 6 | 960 | 1000009314 |
|        |   | 480 | 1000009315 |
|        |   | 240 | 1000009316 |
|        |   | 120 | 1000009317 |
| SIPNET | 8 | 960 | 1000009318 |
|        |   | 480 | 1000009319 |
|        |   | 240 | 1000009320 |
|        |   | 120 | 1000009321 |
| SIPNET | 10 | 960 | 1000009322 |
|        |   | 480 | 1000009323 |
|        |   | 240 | 1000009324 |
|        |   | 120 | 1000009325 |

**Table A9.** Scaling experiment results showing the trade-off between wall-clock time vs. the approximation error with increasing emulator knots.

| m | n | Model run time (sec) | | | GP fitting (sec) | | | 100K MCMC (sec) | | | Deviance |
|---|---|---|---|---|---|---|---|---|---|---|---|
| | | $1^{st}$ | $2^{nd}$ | $3^{rd}$ | $1^{st}$ | $2^{nd}$ | $3^{rd}$ | $1^{st}$ | $2^{nd}$ | $3^{rd}$ | |
| 4 | 120 | 182 | 188 | 184 | 2 | 4 | 12 | 772 | 948 | 1144 | 9489 |
| | 240 | 366 | 364 | 359 | 5 | 27 | 92 | 941 | 1340 | 1764 | 9255 |
| | 480 | 733 | 748 | 744 | 28 | 228 | 707 | 1592 | 2502 | 3614 | 9230 |
| | 960 | 1453 | 1511 | 1505 | 204 | 1736 | 6615 | 2523 | 4862 | 7815 | 9308 |
| 6 | 120 | 182 | 180 | 185 | 2 | 6 | 14 | 795 | 1017 | 1221 | 8371 |
| | 240 | 365 | 368 | 366 | 5 | 27 | 85 | 1039 | 1519 | 1962 | 8284 |
| | 480 | 735 | 777 | 737 | 28 | 215 | 731 | 1544 | 2488 | 3675 | 8310 |
| | 960 | 1521 | 1471 | 1514 | 209 | 1785 | 6858 | 2360 | 4503 | 7799 | 8150 |
| 8 | 120 | 197 | 199 | 198 | 2 | 5 | 12 | 905 | 1116 | 1323 | 9825 |
| | 240 | 410 | 392 | 392 | 7 | 32 | 109 | 1152 | 1611 | 2107 | 8643 |
| | 480 | 745 | 749 | 754 | 30 | 236 | 747 | 1625 | 2596 | 3766 | 8100 |
| | 960 | 1517 | 1532 | 1502 | 217 | 1949 | 6678 | 2532 | 4827 | 7498 | 8062 |
| 10 | 120 | 187 | 187 | 187 | 2 | 7 | 15 | 988 | 1254 | 1277 | 9573 |
| | 240 | 376 | 368 | 418 | 5 | 29 | 92 | 1235 | 1610 | 2075 | 8682 |
| | 480 | 752 | 769 | 766 | 26 | 204 | 787 | 1681 | 2732 | 3489 | 8559 |
| | 960 | 1491 | 1507 | 1490 | 208 | 2015 | 6643 | 2721 | 5010 | 7831 | 8106 |

**m** parameters (*m = {4, 6, 8, 10}*), **k** knots (*k = {120, 240, 480, 960}*)

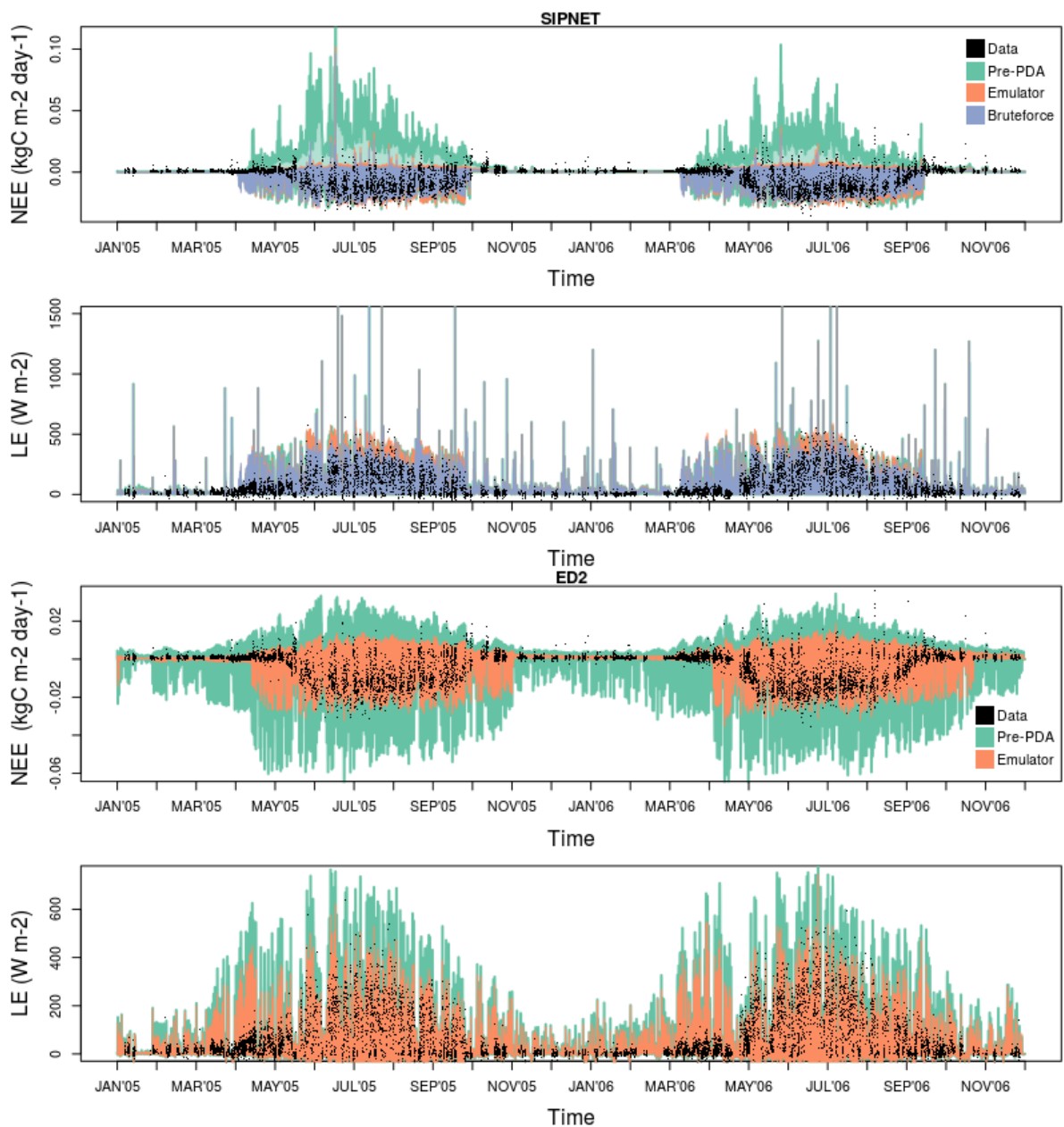

**Figure A1.** Un-smoothed, half-hourly time series comparison for NEE and LE predictions, before and after calibration.

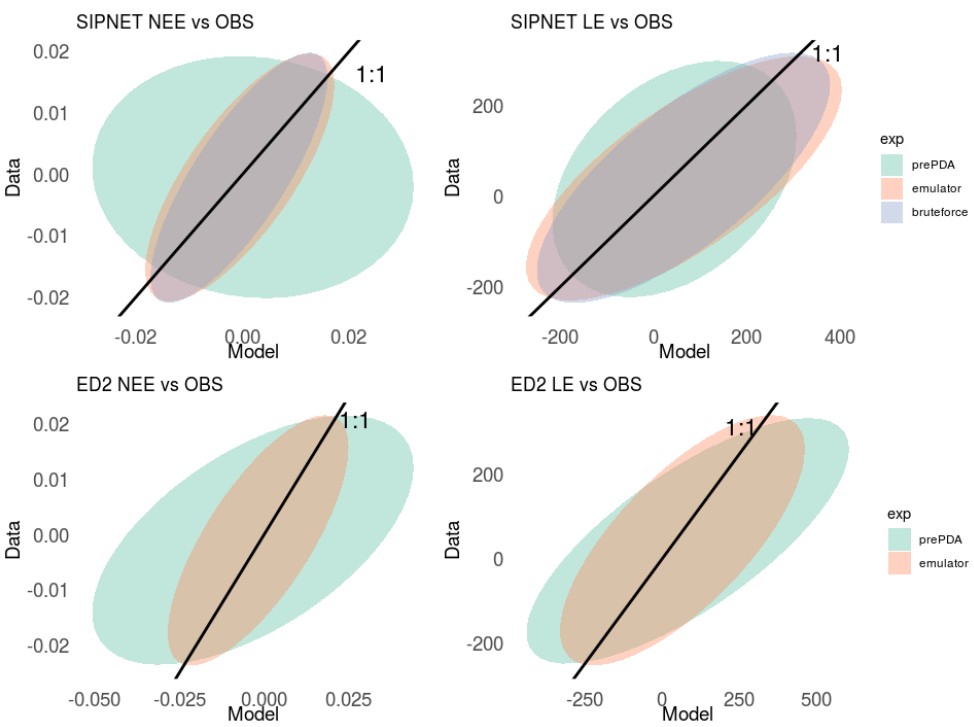

**Figure A2.** Predicted vs observed comparison with concentration ellipses. Top row: SIPNET, bottom row: ED2. Units same as Fig. A1, NEE: kgC m-2 day-1, LE: W m-2.

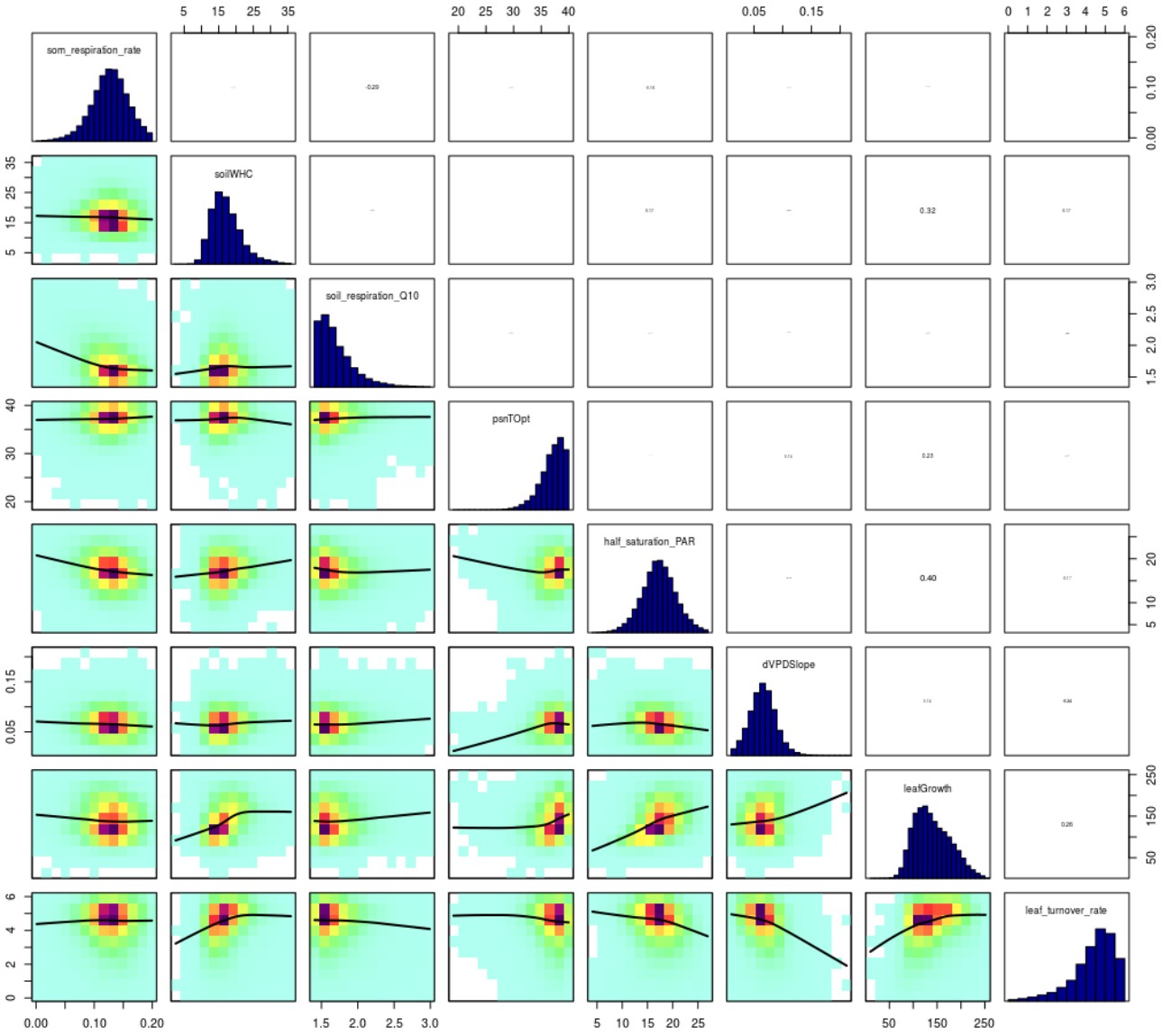

**Figure A3.** Correlations in the posterior samples after emulator MCMC (SIPNET). The lower left and upper right triangles show the correlation density, and the Pearson correlation coefficients between the parameters on the diagonal respectively (Hartig et al., 2017).

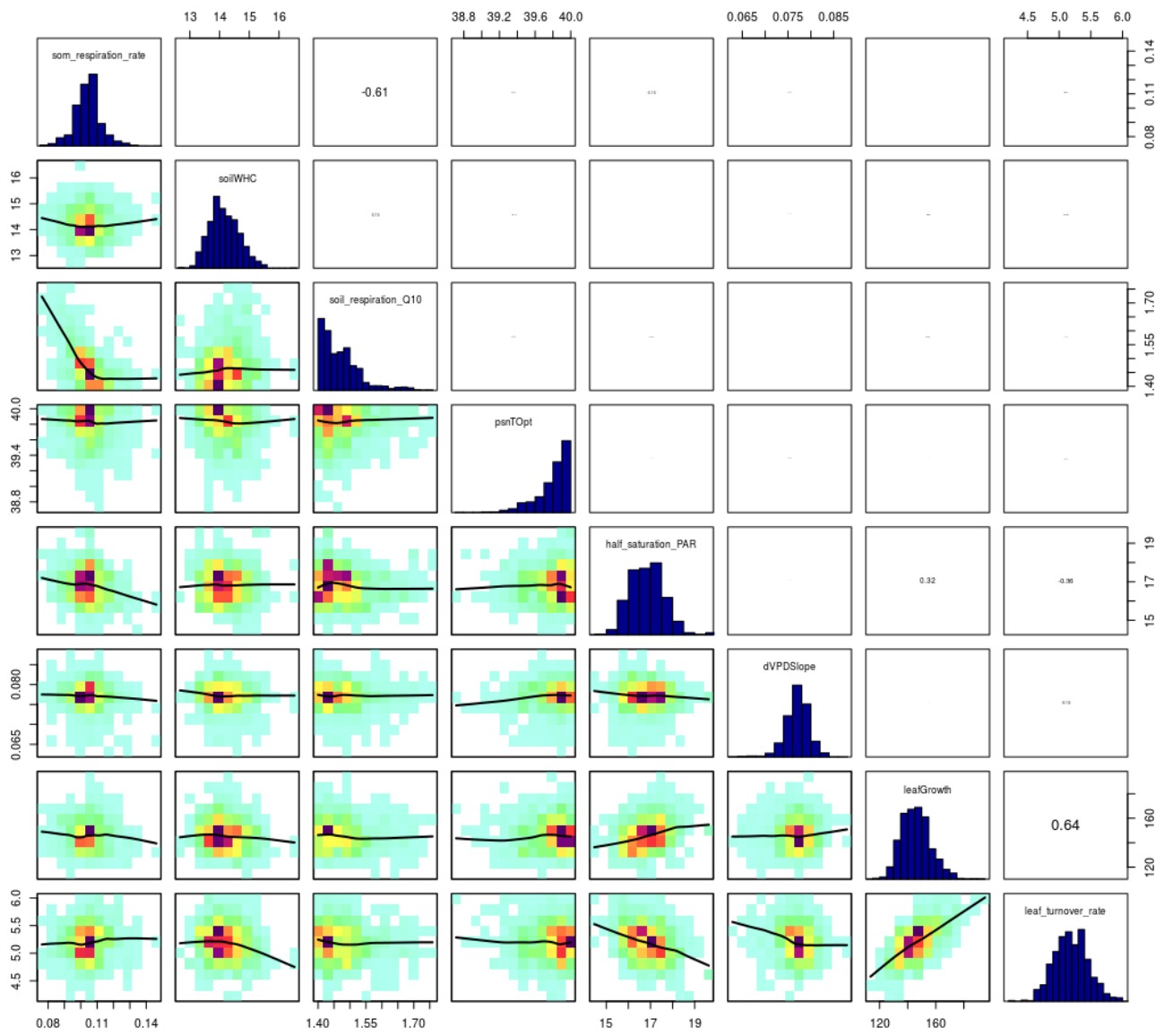

**Figure A4.** Correlations in the posterior samples after bruteforce MCMC (SIPNET).

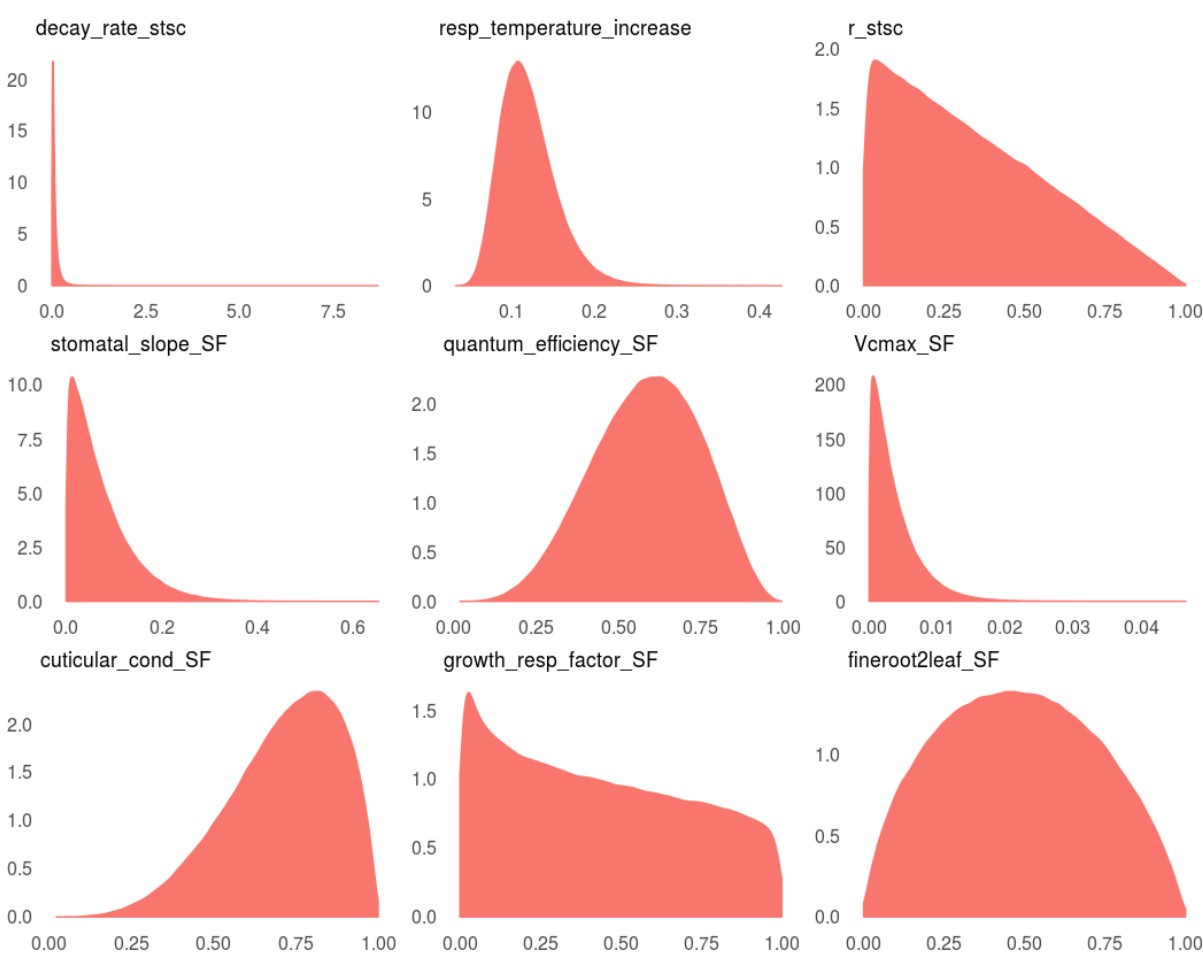

**Figure A5.** ED decomposition and scaling factor posteriors density distributions. Parameters common to all ED2 PFTs ending with suffix "SF" were targeted through the scaling factor.

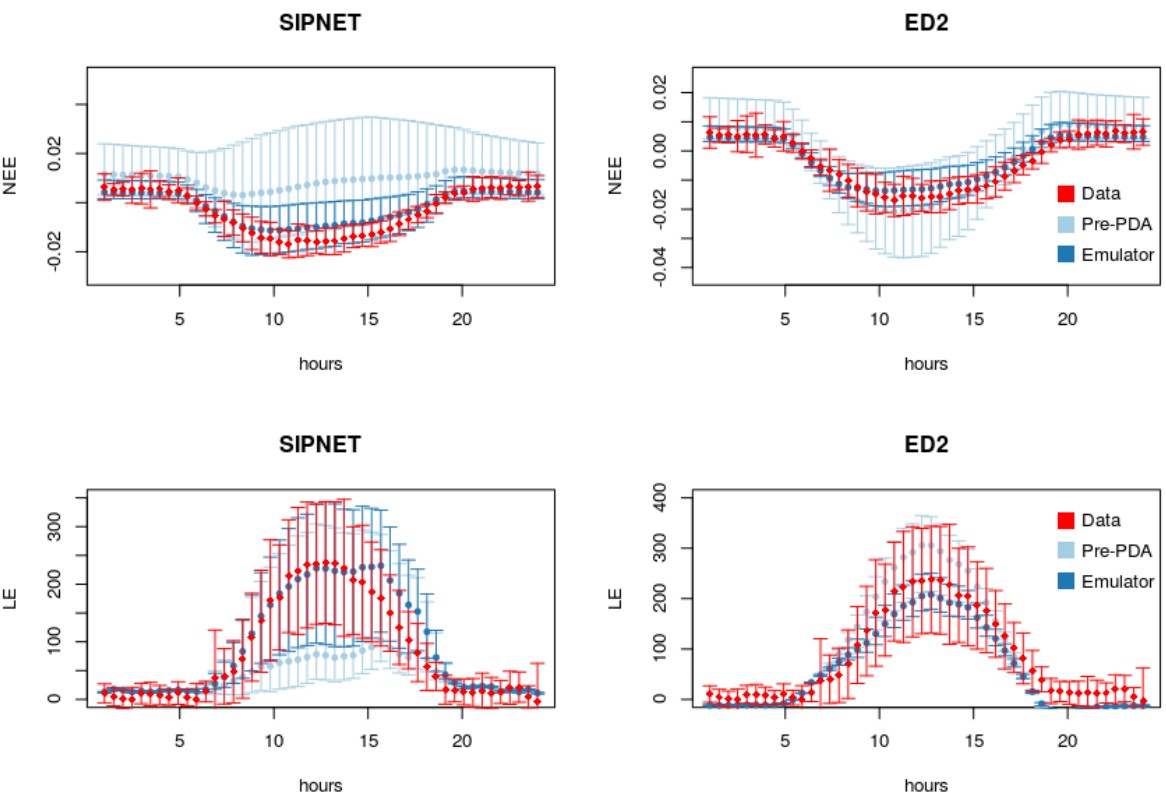

**Figure A6.** Diurnal cycles of NEE and LE fluxes for June-July-August months over the simulation period (2005-2006) before and after the calibration. Error bars represent the variation over the JJA period. Units same as above, NEE: kgC m-2 day-1, LE: W m-2.

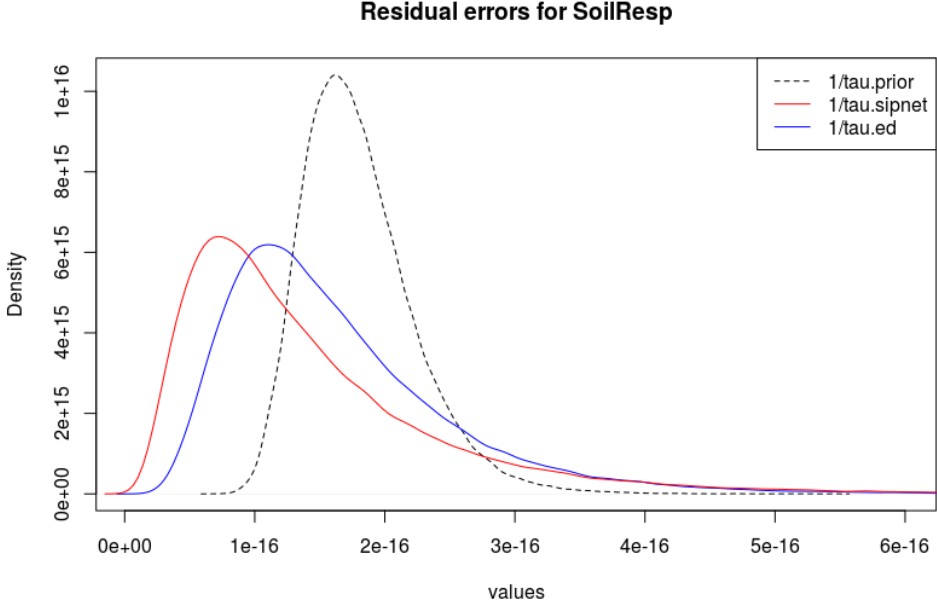

**Figure A7.** Posterior probability density distribution of variance (reciprocal of the precision, $1/\tau$) parameter of the Soil Respiration likelihood after emulator-PDA.

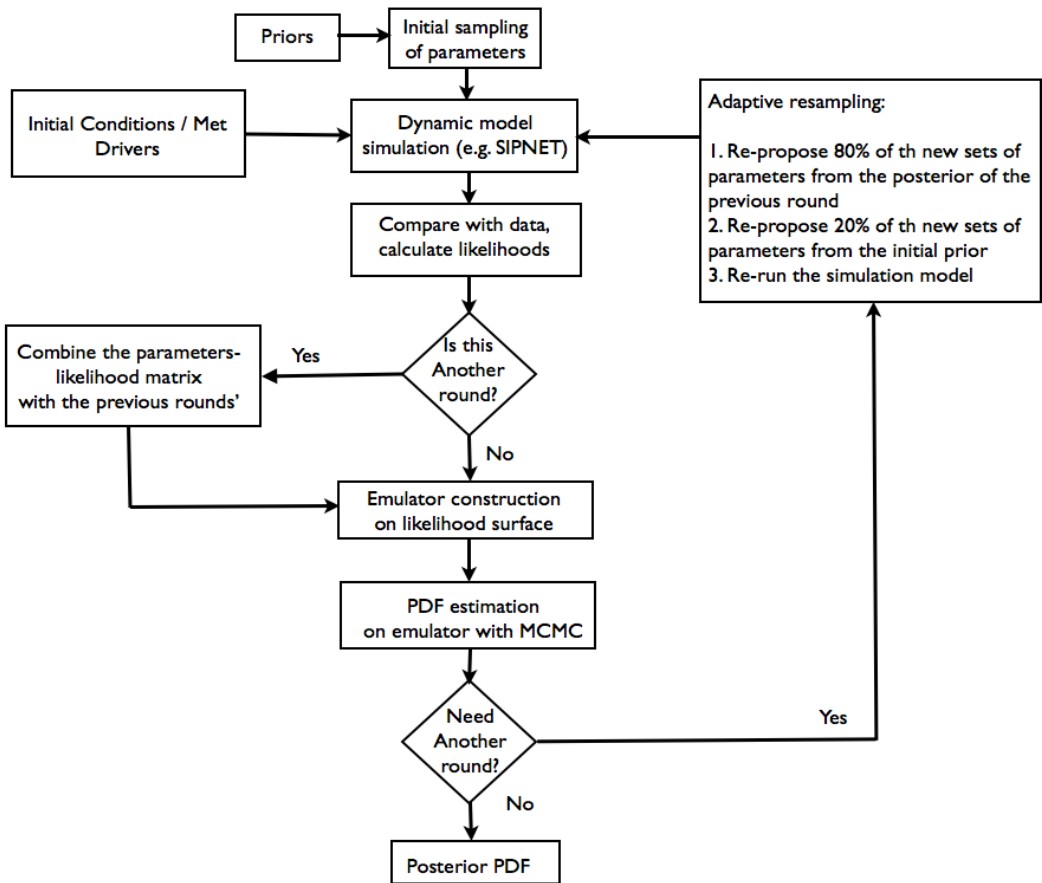

**Figure A8.** Schematic diagram of emulator workflow.