# Peer review of "Linking big models to big data: efficient ecosystem model calibration through Bayesian model emulation"

_Biogeosciences, 2018_

## Author Comment (AC1) · 28 Feb 2018

Dear reviewers and colleagues,

Unfortunately during the re-formatting step, a misalignment occurred in Table 2's columns. Please consider the following format for it.

We apologize for the inconvenience.

Istem Fer, On behalf of all Co-Authors
* * *
Table 2. Performance statistics of ensemble means before and after the PDA for both models and output variables. While root-mean-square-error (RMSE) scores evaluate the absolute deviations of model predictions than data, deviance (-2 x log-likelihood) scores evaluate predictive ability. For both metrics lower scores are better.

420    SIPNET$_E$: Emulator PDA. SIPNET$_B$: Bruteforce PDA. Bold RMSE values for NEE and SoilResp were rescaled by $10^9$ for easier comparison.

| | | NEE | | LE | | SoilResp | |
|---|---|---|---|---|---|---|---|
| | | pre-PDA | post-PDA | pre-PDA | post-PDA | pre-PDA | post-PDA |
| RMSE | SIPNET$_E$ | **140** | **45** | 89 | 81 | **18** | **21** |
| | SIPNET$_B$ | | **43** | | 77 | | **32** |
| | ED2 | **122** | **68** | 124 | 89 | **29** | **18** |
| Deviance | SIPNET$_E$ | 2745 | 1073 | 9879 | 8469 | -1333 | -1372 |
| | SIPNET$_B$ | | 944 | | 8331 | | -1315 |
| | ED2 | 3152 | 1523 | 9914 | 9103 | -1380 | -1390 |

35                          18

**Fig. 1.** Corrected Table 2 columns

---

## Referee Comment (RC1) · M. van Oijen (Referee) · 19 Mar 2018

**GENERAL COMMENTS**

This paper shows how Bayesian calibration of computationally demanding ecosystem models can be sped up using various techniques. The authors test their methods on two ecosystem models, SIPNET and ED2, using synthetic calibration data and flux data from a forest site. A small fraction of the models' parameters (theta) is calibrated by means of Markov Chain Monte Carlo (MCMC) simulation. Instead of running the ecosystem models at each iteration of the MCMC to calculate the likelihood function, L(theta), the authors first derive emulators for the contribution to L(theta) of different

data sets (representing different measurement variables) at any point in parameter space. Building the emulators requires running the original models on a set of training points, but thereafter, in the MCMC, the fast emulators are used at every iteration instead of the slow models.

Contrary to statements made in the paper, the techniques used by the authors are for the most part not novel. There is in fact a substantial literature on replacing the likelihood function with more efficient calculation methods, and I shall give pointers to the literature below. Overall it seems that the literature is very poorly referenced in this paper.

However, in the field of ecosystem modelling, several techniques described by the authors have been used hardly at all, so the paper can be valuable in introducing the ideas to a new audience. Moreover, the tests carried out by the authors demonstrate the effectiveness of the techniques very well, and they present very clear figures and tables. But to introduce new methodological ideas to people, the language should be clear and consistent, and that is not the case here. There is a worrying lack of understanding of the difference between the concepts of 'error' and 'uncertainty'. The first refers to deviation from truth, the second to incomplete knowledge, but in this paper the terms are occasionally treated as synonyms, which makes the Introduction highly unclear. [Proper terminology for these concepts and others can, for example, be found in the review of Bayesian methods by Van Oijen (2017), where also additional references on MCMC, emulation and hierarchical modelling in ecosystem modelling can be found.]

Missing references to the literature include the following. Guttmann & Corander (2016) gave a useful overview of many different ways to replace the likelihood function with faster alternatives (their Table 1 is helpful). Oakley & Youngman (2017) showed many of the same methods as the present authors do. They even provided a six-step procedure that is almost identical to the steps outlined in Fig. 1 and section 2.1 of the present paper. For many examples of likelihood-emulation using Gaussian processes

etc. in cosmology, see Aslanyan et al. (2015) and references 7-24 therein (which also tend to focus on how much computations are made faster by likelihood-emulation). And just as in the present paper, Aslanyan et al.'s procedure alternates between posterior estimation and emulator improvement. Jandorov et al. (2014) used the same refinement employed in the present paper, of emulating sufficient statistics instead of the overall likelihood directly. In contrast to that, Kandasamy & Schneider show that instead of emulating the likelihood, it is also possible to emulate the product of prior and likelihood (i.e. the posterior up to a constant), an approach not mentioned by the present authors. On pages 6-7 of Yurko et al. (2015), some mathematical details are provided of using a 'GP emulator-modified likelihood function'. I further recommend that the authors inspect the literature on ABC and especially on Bayesian quadrature and Bayesian optimization to find ideas that may help refine their approach, and ground it in the wider literature. Further, as perhaps an unmentioned predecessor of calibrating scale parameters rather than the original parameters, see the ecosystem model Bayesian calibration approach of Van Oijen et al. (2011), where every separate data stream came with its own bias parameter.

The Introduction mentions that "Parameter error refers to the uncertainty about the true values of the model parameters", which is quite wrong. Parameter error means assigning a value to a parameter which differs from reality, e.g. stating that the light-use efficiency is 1 g MJ-1 when in reality it is 2 g MJ-1. Not knowing whether it is 1 or 2 or anything else is uncertainty. It is therefore also incorrect to state, as the authors do, that "parameter error asymptotically goes to zero with enough data". It is the conditional uncertainty that goes to zero, not the error. Every experimentalist knows that having any number of biased measurements makes no parameter converge to its correct value - and all measurements have their hidden or unhidden biases. There is no safe way to "estimate observation error from data".

The treatment of the subject matter in the Introduction is further hampered by poor terminology regarding parameters. Terms like "parameter", "parameter vector", "parameter set[s]" are used arbitrarily and inconsistently. [As an exercise for the reader: show that lines 98 and 147 cannot both apply.] Note that a set is unordered and a vector is ordered, so a point in parameter space can not be a "parameter set". And "covariances among parameters" are not real quantities but statistical quantities that capture part of our uncertainty and that change when more data come in. Therefore the covariances are in no way "accounted for". Please note that your subject matter of Bayesian calibration using MCMC is unfamiliar to many readers, so getting an idea of what is going on requires using precise language. Apologies for these pedantic remarks, but in my experience people stumble over the smallest inconsistency when learning Bayesian methods.

Can you elaborate on the limitations of your approach? What is the maximum number of parameters (p) that can be calibrated in general, and for your two models in particular? You set the number of model-runs at p3. Does that mean that calibrating 100 parameters is unfeasible because it would require 106 model evaluations just to build the emulator? And how exactly does PEcAn calculate the contributions of different parameters to overall uncertainty, i.e. what was the screening algorithm?

Published methods for Bayesian calibration increasingly take into account that models are imperfect. There is a discrepancy between model output and reality, even at the best possible setting of model parameter values. This discrepancy is often modeled as a Gaussian Process for which - in the Bayesian calibration - the hyperparameters are estimated together with the regular model parameters. Likelihood-emulation precludes including discrepancy-estimation because model outputs are not calculated during the MCMC. Please add a discussion of this limitation of your approach.

SPECIFIC COMMENTS

There are linguistic errors (plural subjects with singular verbs, missing definite articles etc.) on lines 54, 55, 92, 93, 100, 183, 201, 248, 294, 306 (twice), 309, 323, 351, 372, 418, 434, 436, 443, 454-455, 482 (twice), 483, 484, 485, 507, 511, 520, 539 (twice),

581. I will have missed many more errors - it might be good to involve a native speaker.

The last sentence of the Abstract (l. 34-36) can be deleted without loss of content.

How is the "Euclidean distance between confidence intervals" determined?

Why were 729 knots used for p=8 parameters of SIPNET, given that you state the need for p3 knots (729=93, not 83)?

Two of the references are not placed in their proper alphabetical position, and the reference to Hartig et al. (2012) is missing.

Can you explain the results shown in Tables A2 and A5? How can posterior distributions for parameters following MCMC neatly fall into parameterised probability distributions (which also are often of different type than their priors)? And what were the posterior covariances?

REFERENCES

Aslanyan, G. et al. (2015). Learn-as-You-Go Acceleration of Cosmological Parameter Estimates. J. Cosmology and Astroparticle Physics 09: 005. https://doi.org/10.1088/1475-7516/2015/09/005. [see also http://arxiv.org/abs/1506.01079]

Chowdhury, A. & Terejanu, G. (2016). An Enhanced Metropolis-Hastings Algorithm Based on Gaussian Processes. In: Model Validation and Uncertainty Quantification, Volume 3, 227-33. Conference Proceedings of the Society for Experimental Mechanics Series. Springer, Cham. https://doi.org/10.1007/978-3-319-29754-5_22.

Gutmann, M.U. & Corander, J. (2016). Bayesian Optimization for Likelihood-Free Inference of Simulator-Based Statistical Models. J. Machine Learning Res. 17: 1-47.

Jandarov, R. et al. (2014). Emulating a Gravity Model to Infer the Spatiotemporal Dynamics of an Infectious Disease. J. Roy. Stat. Soc., Series C, Appl. Statist. 63: 423-44. https://doi.org/10.1111/rssc.12042.

Kandasamy, K. & Schneider, J. (2015). Bayesian Active Learning for Posterior Estimation. Proc. 24th Int. Conf. Artifical Intelligence: 3605-3611.

Oakley, J.E. & Youngman, B.D. (2017). Calibration of Stochastic Computer Simulators Using Likelihood Emulation. Technometrics 59: 80-92. https://doi.org/10.1080/00401706.2015.1125391. [Preprint at http://arxiv.org/abs/1403.5196]

Van Oijen, M. et al. (2011). A Bayesian framework for model calibration, comparison and analysis: Application to four models for the biogeochemistry of a Norway spruce forest, Agric. Forest Met. 151: 1609-1621. 10.1016/j.agrformet.2011.06.017.

Van Oijen, M. (2017). Bayesian Methods for Quantifying and Reducing Uncertainty and Error in Forest Models. Current Forestry Reports 3: 269-80. https://doi.org/10.1007/s40725-017-0069-9.

Yurko, J.P. et al. (2015). Demonstration of Emulator-Based Bayesian Calibration of Safety Analysis Codes: Theory and Formulation. Science and Technology of Nuclear Installations. https://doi.org/10.1155/2015/839249.
* * *

---

## Short Comment (SC1) · 9 Apr 2018

We are grateful Dr. Van Oijen's expertise and comments. We thank him for pointing out the relevant literature. While we mainly focused on use of emulators or surrogate models in ecosystem modeling studies, he is right that our citations should widen outside of the literature in ecosystem modeling as we aim to further the use of these techniques in the ecosystem modeling community. We would be happy to include his recommendations and revise the novelty statements in the light of these references.

**The Introduction mentions that "Parameter error refers to the uncertainty about the true values of the model parameters", which is quite wrong. Parameter error**

**means assigning a value to a parameter which differs from reality, e.g. stating that the light- use efficiency is 1 g MJ-1 when in reality it is 2 g MJ-1. Not knowing whether it is 1 or 2 or anything else is uncertainty. It is therefore also incorrect to state, as the authors do, that "parameter error asymptotically goes to zero with enough data". It is the conditional uncertainty that goes to zero, not the error. Every experimentalist knows that having any number of biased measurements makes no parameter converge to its correct value - and all measurements have their hidden or unhidden biases. There is no safe way to "estimate observation error from data".**

We share Dr. Van Oijen's concerns about the consistency of concepts. It is important for us that discussions of these concepts, and methods for their analyses, become more common practice in ecosystem modeling studies. We completely agree with reviewer's definitions of error and uncertainty: "Parameter error is the difference from [its correct] value, and parameter uncertainty is not knowing what that value is" (Van Oijen, 2017). We intended to refer to "uncertainty about parameter errors", thank you for catching it. Therefore, we also agree with that it is the parameter uncertainty that goes to zero. We would be happy to revise these statements to the satisfaction of the reviewer, including additional citations.

**The treatment of the subject matter in the Introduction is further hampered by poor terminology regarding parameters. Terms like "parameter", "parameter vector", "parameter set[s]" are used arbitrarily and inconsistently. [As an exercise for the reader: show that lines 98 and 147 cannot both apply.] Note that a set is unordered and a vector is ordered, so a point in parameter space can not be a "parameter set".**

We thank Dr. Van Oijen for pointing out the usage of parameter vector vs parameter set. We were using "sets of parameters" for a number of parameter combinations that went in for a particular LHC ensemble, as in [nKNOTS x nPARAMS] (being nrows x ncols). For fitting the GP, the order among rows is not important, whereas each row (a

parameter set in that sense) is a(n ordered) vector of parameters (e.g. usage on L189). However, the reviewer is right that their current usage is confusing, if not wrong. We will refer the latter [1 x nPARAMs] as "a parameter vector" and the former [nKNOTS x nPARAMS] as "a parameter set" and will not use "sets of parameters" unless we refer to multiple parameter sets (e.g. multiple parameter sets for multiple iterative emulator rounds). We will go through the text and make sure these are introduced and referred consistently.

**And how exactly does PEcAn calculate the contributions of different parameters to overall uncertainty, i.e. what was the screening algorithm?**

The uncertainty analysis in PEcAn uses a simple one-at-a-time (OAT) approach. OAT approach involves multiple model runs while holding all parameters at their median except one each time, and evaluating how it translates to differences in model outputs. The parameters are varied at their parameter data assimilation (PDA) analysis priors' (which could be original priors or if the parameter was constrained by the meta analysis, they could be meta analysis posteriors in PEcAn) median and at six PDA prior quantiles equivalent to $\pm[1,2,3]\sigma$ in the standard normal. More details are given in previous papers as cited (LeBauer et al., 2013; Dietze et al., 2014).

**How is the "Euclidean distance between confidence intervals" determined?**

"Euclidean distance between confidence intervals" were determined simply by calculating the mean Euclidean distance between 2.5%- 97.5% CIs of post-emulator and post-bruteforce PDA ensembles at each time point. For example, for half-hourly time step and two years (2005-2006) of flux outputs, we have 35040 points in our model output time series. Then there will be 35040 values of $(CI_{E,L} - CI_{B,L})^2$ where E stands for emulator, B stands for bruteforce ensemble and L stands for lower CI limit. The same is calculated for upper CI limit and sum of their mean is used as a score for relative confidence interval (RCI) coverage per variable:

$$RCI_{VAR} = \text{mean}((CI_{E,L} - CI_{B,L})^2) + \text{mean}((CI_{E,U} - \text{CI}_{B,U})^2)$$

Then the sum over variables (in our case, $RCI_{FINAL} = RCI_{NEE} + RCI_{LE} + RCI_{SoilResp}$) gives us the final RCI score. We expect this score to get smaller as the approximation error of emulator decreases with increasing number of knots and the post-emulator PDA ensemble CIs overlaps more with post-bruteforce CIs. While this metric is not perfect and certain scenarios can even result in misleading scores (e.g. a narrower -than bruteforce- emulator CI coverage can give the same score with an equally wider -than bruteforce- emulator CI coverage according to this calculation, which is an unlikely scenario but just to give an example), in our experience it contains information about how close the CI coverages of post PDA ensembles for both approaches are. We would be happy to include more details and discussion on this in the text and provide more supplemental figures as an example. Please below see a visualization of emulator CI coverage approaching to bruteforce CI coverage with increasing number of knots for NEE (Fig 1). Please note that this is a smoothed time-series for ease of visual inspection. Otherwise, these are from the exact same runs from our scaling experiment with 8 parameters that are used in RCI calculations reported in Figure 7a in the paper.

**Can you explain the results shown in Tables A2 and A5? How can posterior distributions for parameters following MCMC neatly fall into parameterised probability distributions (which also are often of different type than their priors)? And what were the posterior covariances?**

We would like to clarify that the results reported in Tables S2 and S5 are fitted parametric distributions to the MCMC samples. We wanted to provide an approximate parametric distribution for the reader for ease of use. Otherwise, all the raw MCMC samples are accessible via our workflow directories for more interested users/readers. (e.g. from the following url http://pecan2.bu.edu/pecan/08-finished.php?workflowid=1000008503 -please note that this takes a while to load- the reader can first select an "mcmc.list.pda***.Rdata" file under the "PEcAn Files" dropdown menu on the left frame. By clicking "Show File" button they can download the raw

MCMC outputs to their own machines for further analyses.) We can include a more detailed explanation in the text and in the supplement, we thank the reviewer for pointing this out.

Correlation density plots were provided in the supplementary but were not discussed in the text. We will list the strongest correlations in the text more explicitly and add discussion accordingly.

Finally, we would be happy to include further discussion on limitations of our approach and address all comments of Dr. Van Oijen in a final response given the chance of revision.

Istem Fer

Dietze, M.C., Serbin, S.P., Davidson, C., Desai, A.R., Feng, X. Kelly, R., Kooper, R., LeBauer, D., Mantooth, J., McHenry, K., and Wang, D.: A quantitative assessment of a terrestrial biosphere model's data needs across North American biomes, J. Geophys. Res. Biogeosci., 119, 286–300, doi:10.1002/2013JG002392, 2014.

LeBauer, D. S., Wang, D., Richter, K. T., Davidson, C. C., and Dietze, M. C.: Facilitating feedbacks between field measurements and ecosystem models, Ecological Monographs, 83: 133–154. doi:10.1890/12-0137.1, 2013.

[Figure]

**Fig. 1.** Post emulator-PDA ensemble CI converging to bruteforce CI with increasing number of knots

---

## Referee Comment (RC2) · Anonymous Referee #2 · 27 Apr 2018

Fer et al present an approach to speed the inversion of parameters in ecosystem models via the construction of emulators. The methods are not novel, but application of the method in the field of biogeosciences is in its infancy and the example experiment provided here may be useful in designing further approaches. For this however, it would be important to have a deeper description of the results and a clearer link to the conclusions.

1) For example, the comparison between the results of the emulated and the real SIP-NET show that the distributions and central moments of the posteriors are different. This is seen in:

a. Figure 3, where there is not "superior" approach across parameters: sometimes is R3, sometimes AAO, sometimes both R2 and R3 are equally good.

b. Figure 5, where 50% of the emulated SIPNET parameters are (statistically?) different from the central moment of the distribution of the "bruteforce" model calibrations and all of the emulated estimates have substantially higher ranges.

Both these results suggest that some further developments have to be investigated in order to rely on posteriors from emulators. It would be key to investigate why the emulators are overall inflating uncertainty and missing the optimum in particular parameters (equifinality? Non-linearities in model functions controlled by those parameters?).

2) Overall I miss quantitative statistical information about the fitness (model performance) stemming from the parameters obtained via the emulator and the "bruteforce" method against (1) synthetic data and (2) observations (e.g. Nash Sutcfliffe or the Kling Gupta Efficiency). This should also be illustrated by scatter plots and figures that show not only the subdaily but also the seasonal cycle in synthetic/real-world data against models.

Knowing the time it takes for the calculations to get done is indeed of technical relevance. But here the most relevant aspects (at least in the perspective of BGD) are centered on how the different model realizations stemming from the emulator approach against the traditional approach change the retrieval of optimal parameters (and posterior uncertainties) and in the eddy covariance flux predictions (for which many relevant information is mostly found in supplements). These are especially important to understand the limitations and caveats of the current proof-of-concept exercise (evaluation of the synthetic exercise). Another missing important aspect is to understand how the overall results change when contaminating the synthetic dataset with noise (with the same characteristics such as the real observations).

3) The argumentation behind the sufficient statistics is not sustained by the experiment. It is not analyzed how does the emulator performance changes by the inclusion of more

BGD
or less data streams.

Some minor points for attention and discussion:

-> Related to Equation 3, please see the analysis and discussion in Lasslop et al 2008.

-> There are a few uninformative visuals, like Figure 4 top 2 panels; Figure 3, the som\_resp\_rate; that could be replaced by more informative elements (new figures, or tables).

Finally, the results from ED2 seem to be encouraging regarding improving the parameterizations of very computationally expensive models. But the results on the uncertainties undermine its ability to provide a proper representation of the parametric uncertainties.

Last, I would like to express my appreciation to the Authors for providing the tools and following an open source philosophy. Thank you.

REFERENCES Gupta, Hoshin V., Harald Kling, Koray K. Yilmaz, Guillermo F. Martinez. Decomposition of the mean squared error and NSE performance criteria: Implications for improving hydrological modelling. Journal of Hydrology, Volume 377, Issues 1-2, 20 October 2009, Pages 80-91. DOI: 10.1016/j.jhydrol.2009.08.003. ISSN 0022-1694

Kling, H., M. Fuchs, and M. Paulin (2012), Runoff conditions in the upper Danube basin under an ensemble of climate change scenarios. Journal of Hydrology, Volumes 424-425, 6 March 2012, Pages 264-277, DOI:10.1016/j.jhydrol.2012.01.011 Lasslop et al., 2008: https://www.biogeosciences.net/5/1311/2008/bg-5-1311-2008.pdf
**Discussion** paper

---

## Author Comment (AC2) · 17 May 2018

We thank both referees for their comments and suggestions. We would also like thank the associate editor for diligently inviting reviewers who are most well suited. We find the comments very helpful. In the light of these comments, we realized that we were missing an opportunity to improve our workflow, we now highlight this change and report our results below.

Also it is clear from both referee comments that we should enhance our literature citation, and revise text accordingly. Below please find our responses and suggested changes in the manuscript.

**Referee's Comment (RC) 1 - Contrary to statements made in the paper, the techniques used by the authors are for the most part not novel. There is in fact a substantial literature on replacing the likelihood function with more efficient calculation methods, and I shall give pointers to the literature below. Overall it seems that the literature is very poorly referenced in this paper. However, in the field of ecosystem modelling, several techniques described by the authors have been used hardly at all, so the paper can be valuable in introducing the ideas to a new audience.**

**RC 2 - The methods are not novel, but application of the method in the field of biogeosciences is in its infancy and the example experiment provided here may be useful in designing further approaches.**

Authors' Comment (AC) - We thank the reviewers for this remark. In terms of our novelty statements, we wanted to explain that this paper is the culmination of work that has started approximately 12 years ago (please see AGU talk abstract by Dietze et al., 2009) and it was rather novel even across disciplines back then. We acknowledge the fact that this is not the case anymore and offer our apologies for missing key papers.

However, as both reviewers highlighted, a decade after they were first introduced, the techniques described in this paper have been used hardly at all in the field of ecosystem modeling. This is not surprising given that applications of these techniques require a non-trivial amount of computational and statistical expertise, not to mention a steep debugging curve of both models and algorithms. In this paper, we report the integration of a standardized ecological application of these methods in an open-source ecological informatics toolbox for the general use of the ecosystem modeling community. It is exactly our hope that the experiments and the implementations provided here may foster more use and development of novel types of model emulators.

We have revised the novelty statements and provided citations from the literature as needed. We appreciate the constructive comments of the reviewers, which have improved both our workflow and manuscript.

**RC 1 - To introduce new methodological ideas to people, the language should be clear and consistent, and that is not the case here. There is a worrying lack of understanding of the difference between the concepts of 'error' and 'uncertainty'. The first refers to deviation from truth, the second to incomplete knowledge, but in this paper the terms are occasionally treated as synonyms, which makes the Introduction highly unclear. Proper terminology for these concepts and others can, for example, be found in the review of Bayesian methods by Van Oijen (2017), where also additional references on MCMC, emulation and hierarchical modelling in ecosystem modelling can be found.**

**The Introduction mentions that "Parameter error refers to the uncertainty about the true values of the model parameters", which is quite wrong. Parameter error means assigning a value to a parameter which differs from reality, e.g. stating that the light- use efficiency is 1 g MJ-1 when in reality it is 2 g MJ-1. Not knowing whether it is 1 or 2 or anything else is uncertainty. It is therefore also incorrect to state, as the authors do, that "parameter error asymptotically goes to zero with enough data". It is the conditional uncertainty that goes to zero, not the error. Every experimentalist knows that having any number of biased measurements makes no parameter converge to its correct value - and all measurements have their hidden or unhidden biases. There is no safe way to "estimate observation error from data".**

AC - As mentioned in our previous short comment, we share the reviewer's concerns about the consistency of concepts. It is important for us that discussions of these concepts, and methods for their analyses become more common practice in ecosystem modeling studies. We completely agree with reviewer's definitions of error and uncertainty, and revised these sections in the introduction as follows:

Authors' changes in the manuscript L64-82 :

The Bayesian approach also distinguishes between parametric, model structural and data uncertainties, which is critical for ecological forecasting. Parameter uncertainty refers to the uncertainty about the true values of the model parameters due to data deficiency and model simplification (McMahon et al., 2009; Van Oijen, 2017). As models are simplified representations of reality, it is often not possible to measure the true value of an ecosystem model parameter precisely in the field, regardless of the measurement errors (Van Oijen, 2017). However, measurements can still provide estimates for parameter values that makes the model represent the reality better (Van Oijen, 2017). Hence, it is possible to reduce parameter uncertainty with more measurements, conditioned upon the model structure and the measurement error (Van Oijen, 2017; Dietze, 2017a). Therefore, the parameter uncertainty should be reflected by probability distributions and propagated into model predictions. By reducing parameter uncertainties, PDA helps us identify where we need further data collection and improved model representations.

By contrast, process or model structural uncertainty refers to the uncertainty about how to represent ecological processes in models. As every model is a simplification of reality, there will always be underrepresented processes or insufficiently modeled interactions in ecological models (Van Oijen, 2017; McMahon et al., 2009; Clark, 2005). With more observations, we can advance our theoretical understanding and better characterize ecological processes, but process uncertainty does not necessarily decrease with more data, the way parameter uncertainty does (Dietze, 2017a; Gupta et al., 2012; Clark, 2005). As process uncertainty is part of our imperfect models, it is part of the uncertainty associated with the model predictions.

Unlike process and parameter uncertainties, data (observation) uncertainty does not need to be propagated into model predictions. Observation error

is a result of the limited precision and accuracy of the measurement instruments, hence, the uncertainty about it is not part of the process that we are trying to model (Van Oijen, 2017; McMahon et al., 2009). In Bayesian PDA, observation uncertainty should be treated independent of the deviations of model predictions from data as part of the likelihood for observations to inform model predictions without biases (Dietze, 2017a). For a more in depth terminology for these concepts in the context of process-based models and Bayesian methods, see review by Van Oijen (2017).

**RC 1 - The treatment of the subject matter in the Introduction is further hampered by poor terminology regarding parameters. Terms like "parameter", "parameter vector", "parameter set[s]" are used arbitrarily and inconsistently. [As an exercise for the reader: show that lines 98 and 147 cannot both apply.] Note that a set is unordered and a vector is ordered, so a point in parameter space can not be a "parameter set". And "covariances among parameters" are not real quantities but statistical quantities that capture part of our uncertainty and that change when more data come in. Therefore the covariances are in no way "accounted for". Please note that your subject matter of Bayesian calibration using MCMC is unfamiliar to many readers, so getting an idea of what is going on requires using precise language. Apologies for these pedantic remarks, but in my experience people stumble over the smallest inconsistency when learning Bayesian methods.**

AC - We are grateful for such remarks, and have revised the text accordingly.

L98-99: In the emulator approach, we first propose a set of parameter underlinevectors (knots) according to a statistical design (Fig. 1). Then, we run the full model with this set of underlinedesign points in parameter space, and compare the model outputs with data.

L113-115: Instead of constructing an emulator for the raw model output, we adopt the approach of constructing an emulator of the likelihood – the statistical assessment of the probability of the data given a vector of model parameters which forms the basis for both frequentist and Bayesian inference.

L147-148: (1) Propose initial $N_{KNOTS}$ design points in the parameter space

(2) Run full model with each parameter vector (parallelizable over $N_{KNOTS}$)

L155: (5a) Propose a new vector of process-model parameter values (each parameter vector defines a point in multivariate parameter space)

L177: The second step (2) is to evaluate the full model using the proposed design points in parameter space (knots), and it is the only step where we run the full model.

L189: This allows us to not only accept/reject a proposed parameter vector (5e) but also sample the $\tau$ conditional on that parameter vector (step 5f).

L214: In the MCMC, we use the GP to estimate $T$ for both the current and proposed parameter vector (5b).

L216: To propagate this interpolation uncertainty, it is important to draw the $T$ stochastically from the GP, and draw new values for both the current and proposed parameter vector at each iteration.

L221: This is in contrast with traditional optimization and MCMC algorithms that only leverage the current vector of parameter values when proposing new parameters.

L61: As opposed to piecewise evaluation of different parts of the model against different data sets, a Bayesian framework allows the evaluation of the whole model at once against all data sources, reflecting the connections between variables and the covariances among parameters (Dietze, 2017a).

**RC 1 - Can you elaborate on the limitations of your approach? What is the maximum number of parameters ($p$) that can be calibrated in general, and for your two models in particular? You set the number of model-runs at $p^3$. Does that mean that calibrating 100 parameters is unfeasible because it would require $10^6$ model evaluations just to build the emulator?**

AC - With the current ($p^3$) scheme calibrating 100 parameters would be infeasible as it would require $10^6$ model evaluations just to build the emulator. With ED2, running the model $10^6$ times is not feasible at all, unless iterative emulator rounds are massively parallelized. With SIPNET, the Cholesky decomposition within the GP, rather than the model evaluations, would become limiting for $10^6$ design points. In that case, emulators other than GP (e.g. NNGP) could be considered as we discuss in the manuscript.

That said, the ($p^3$) scheme is just the rule-of-thumb that we employed in these experiments, and not an inherent limit of the emulator approach itself. The calibration of 100 parameters might be possible with a much smaller number of knots ($\ll 10^6$) depending on the model. For example, our scaling experiment (Figure 7b) shows that, in terms of deviance, it was possible to constrain 6 SIPNET parameters to a reasonable extent with 120 knots in total (likewise, 8 and 10 SIPNET parameters with 240 parameters in total). A common recommendation in computer experiments with GP is to use a sample size about 10 times (n = 10d) the input dimension (Loeppky, Sacks Welch, 2009). Others found this is often too small and suggest 20 times (n = 20d) larger sample size (Erickson, Ankenman Sanchez, 2018).

Therefore, calibrating 100 model parameters with 100 x 20 design points could be possible in theory. In practice, we would advocate for performing an uncertainty analysis to reduce the dimensionality of the problem to the subset these 100 parameters that contribute most to model uncertainty. In addition, the data would need to be strong enough to actually constrain 100 parameters. We would be happy to extend the discussion in section 4.6 to explicitly report these numbers (about sample sizes) with references.

**RC 1 - How exactly does PEcAn calculate the contributions of different parameters to overall uncertainty, i.e. what was the screening algorithm?**

AC - The uncertainty analysis in PEcAn uses a one-at-a-time (OAT) approach. An OAT approach involves multiple model runs while holding all parameters at their median except one each time, and evaluating how it translates to differences in model outputs. The parameters are varied at their parameter data assimilation (PDA) analysis priors' (which could be original priors or, if the parameter was constrained by the meta analysis, they could be meta analysis posteriors in PEcAn) median and at six PDA prior quantiles equivalent to $\pm[1, 2, 3]\sigma$ in the standard normal. Details are given in previous papers as cited (LeBauer et al., 2013; Dietze et al., 2014). Plans are in place to develop a more general multivariate uncertainty analysis in the future once the multivariate version of our trait meta-analysis is in place (Shiklomanov et al in review).

**RC 1 - There are linguistic errors (plural subjects with singular verbs, missing definite articles etc.) on lines 54, 55, 92, 93, 100, 183, 201, 248, 294, 306 (twice), 309, 323, 351, 372, 418, 434, 436, 443, 454-455, 482 (twice), 483, 484, 485, 507, 511, 520, 539 (twice), 581.**

We thank the reviewer for noticing and noting these errors. We went through the text more carefully and believe we have corrected these errors.

L54: In Bayesian calibration it is possible to use more than one type of data to simultaneously constrain multiple output variables in a model.

L55: Using multiple data constraints is particularly helpful because model errors can compensate for each other and single variables often do not provide robust constraints.

L92-93: Thus, it is particularly advantageous to consider techniques that are both parallel in nature and which have substantial "memory"

L100: Next, we fit a statistical approximation through the design points

where we evaluated the model.

L183: In this case, $T$ for a Gaussian likelihood would be the sum of squared residuals, $\Sigma(y - \mu)^2$, where $y$ is the observation and $\mu$ is the model prediction.

L201: GP assumes that the covariance between any set of points in parameter space is multivariate Gaussian, and that the correlation between points decreases as the distance between them increases

L248: The error distribution of flux data is known to be both heteroskedastic, with variance increasing with the magnitude of the flux, and to have a double exponential distribution rather than a normal (Richardson et al., 2006).

L294: Unlike SIPNET, it is possible to run ED2 simulations with more than one competing PFT.

L306: The use of literature constraints ensures that the posterior parameter estimates fall within underlinea biologically plausible range, and reduces the problem of equifinality, as parameters that are already well constrained cannot vary much, and thus cannot trade-off with poorly constrained parameters.

L309: The scaling factors used for common ED2 PFT parameters all have Beta(1,1) prior distributions.

L323: In the end, 9 and 10 parameters were targeted in SIPNET and ED2, respectively (i.e. in the case of ED2, 9 model parameters are shown in Fig. 2, plus the multiplicative bias parameter)

L351: In our scaling experiment, we evaluate the trade-off between the number of model runs and the approximation error by comparing the 8-parameter SIPNET bruteforce calibration to emulator calibrations with varying numbers of $k$ knots ($k = 120, 240, 480, 960$).

L372: Shaded distributions are the posteriors obtained after each round of emulation.

L418: While root-mean-square-error (RMSE) scores evaluate the deviations of model predictions from data, deviance (-2 x log-likelihood) scores evaluate predictive ability.

L434: However, the time-series plot of LE for SIPNET (Fig. 4, middle panel) shows that SIPNET largely overestimates the winter moisture fluxes whereas ED2 does not (Fig. 6, middle panel).

L436: Both pre- and post-PDA ED2 performance for SoilResp were better than SIPNET (bottom panels).

L443: As expected, the post-PDA ensemble CI approaches the bruteforce post-PDA CI. In other words, the RCI asymptotically converges to zero, while the clock time increases with the number of knots.

L454-455: With a lower number of knots fewer parameters were well-constrained, but with too few parameters we traded-off the ability to get a good fit.

L482-485: First, we ran the full MCMC in between the adaptive sampling steps, and on the final response surface, instead of an optimization search. Hence, we were able to provide full posterior probability distributions of the parameters targeted for calibration, instead of point estimates of optimum values as Li et al. (2018). The ASMO scheme has also been recently updated for distribution estimation using full MCMC runs (ASMO-PODE) and has been tested with Common Land Model (Gong and Duan, 2017).

L507: In addition to just fitting the model, emulators make it practical to implement different hypotheses within a model, re-calibrate the model, and test them against data repeatedly.

L511: For example, it is a known issue that site-level calibrations are not easily transferable to new sites or to larger scales (Post et al., 2017).

L520: A lack of independence in observation errors causes overfitting of the model parameters and underestimates prediction uncertainty.

L539: This experiment showed that the emulator method with SFs could constrain ED2 PFT parameters and improve model predictions.

L581: Future directions may include exploring alternative emulators, such as the Nearest-Neighbor Gaussian Process model (which takes advantage of the fact that nearest neighbors contribute the most information while fitting a GP model), and could help reduce computational costs substantially for bigger datasets and larger numbers of parameters.

**RC 1 - The last sentence of the Abstract (l. 34-36) can be deleted without loss of content.**

AC - We can delete this sentence.

**RC 1 - How is the "Euclidean distance between confidence intervals" determined?**

AC - Please also see the previously posted short comment for more details. Realizing that this was not clear in the manuscript, we added the following text:

L354:

To do this, we compared the post-emulator PDA ensemble confidence interval errors relative (RCI) to the post-bruteforce PDA ensemble CI in terms of mean Euclidean distance between their 2.5% - 97.5% CIs. For each experiment with $k$ different knots and variable $(CI_{E,L,k} - CI_{B,L,k})^2$ values were calculated where $E$ stands for emulator, $B$ stands for bruteforce ensemble
and $L$ stands for lower CI limit. The same is calculated for upper CI limit and sum of their mean is used as a score for relative confidence interval (RCI) coverage per variable:

$$RCI_{VAR,k} = mean((CI_{E,L,k} - CI_{B,L,k})^2) + mean(CI_{E,U,k} - CI_{B,U,k})^2)]$$

Next, each RCI vectors ($RCI_{VAR} = RCI_{VAR,960}, RCI_{VAR,480}, RCI_{VAR,240}, RCI_{VAR,120}$) are normalized by dividing by their mean to obtain values independent of the units. Then, the sum over variables (in our case, $RCI_{FINAL} = RCI_{NEE} + RCI_{LE} + RCI_{SoilResp}$) gives us the final RCI score.

L443: As expected, the post-PDA ensemble CI approaches to bruteforce post-PDA CI, in other words the RCI asymptotically converges to zero, while the clock time to increases with the number of knots (Fig. 7a; also see Fig. S6 for time-series plot that shows emulator CI coverage approaching the bruteforce CI coverage with increasing number of knots).

We also include additional supplementary figures showing coverage convergence, similar to the one we presented in the short comment.

**RC 1 - Why were 729 knots used for $p = 8$ parameters of SIPNET, given that you state the need for $p^3$ knots ($729 = 9^3$, not $8^3$)?**

AC - Because we counted the multiplicative bias parameter in the $p$. So, 8 SIPNET parameters plus the multiplicative bias parameter, $p = 9$ for SIPNET. 9 ED2 parameters (6 of 9 being scaling factors for common PFT parameters) plus the multiplicative bias parameter, $p = 10$ for ED2. Thank you for pointing this out. We will state this in the text more explicitly and add figures for bias parameter posteriors as well.

L323: In the end, 9 and 10 parameters were targeted in SIPNET and ED2, respectively. To be more specific, the 8 (9) model parameters for SIPNET

(ED2) that are shown in Fig. 2, plus the multiplicative bias parameter were targeted in the PDA, therefore $9^3$ ($10^3$) knots were proposed iteratively with the emulator approach.

**RC 1 - Two of the references are not placed in their proper alphabetical position, and the reference to Hartig et al. (2012) is missing.**

AC - We corrected these in the revised manuscript.

**RC 1 - Can you explain the results shown in Tables A2 and A5? How can posterior distributions for parameters following MCMC neatly fall into parameterised probability distributions (which also are often of different type than their priors)? And what were the posterior covariances?**

AC - As explained in the short comment, the results reported in Tables S2 and S5 are fitted parametric distributions to the marginal MCMC samples. We wanted to provide an approximate parametric distribution for the reader for ease of use. Otherwise, all the raw MCMC samples are accessible via PEcAn for more interested readers. We will now extend the explanation before Table S6 to:

> Table S6 caption:
>
> Links to the Workflow IDs. The input/output files associated with each workflow can be accessed via the history table at https://pecan2.bu.edu/pecan/history.php. Or each workflow can be accessed directly by replacing the workflowID in the following link:
>
> https://pecan2.bu.edu/pecan/08-finished.php?workflowid=1000008503 (please note that this takes a while to load)
>
> The left frame on the page can be used to navigate through PEcAn settings, input and output files. If you wish to conduct further visualization or analysis on the MCMC samples, you can first select the "mcmc.list.pda***.Rdata" file

(*** being the ensemble IDs given by the workflow) under the "PEcAn Files" dropdown menu on the left frame. By clicking "Show File" button you can download the raw MCMC outputs to your own machines.

If you would like to display posterior density distributions, first select either soil or plant physiology the under the "PFTs/PFT" menu on the left frame. Next, under the "PFTs/Output" dropdown menu select "posteriors.pda.***.pdf" files and click "Show PFT Output". The red line would be the posterior density plot and the black line would be the approximated parametric distributions (such as the ones reported in Table S2 and S5) fitted by PEcAn's approx.posterior function that can be found under pecan/modules/meta.analysis/R/approx.posterior.R

L428: Fitted parametric posterior distributions of ED2 are given in the supplement (Fig. S1, Table S5.) In addition all raw MCMC samples ("mcmc.list.pda***.Rdata") and posterior distribution plots ("posteriors.pda.***.pdf") are available from the respective workflow directories (see Table S6).

L397: The strongest correlations between leaf growth and leaf turnover rate, and soil respiration rate and soil respiration Q10 parameters were also detectable in emulator posteriors (emulator Fig S4, bruteforce Fig S5).

Important note on an improvement/fix from the authors: Before, we were using these fitted parametric distributions to i) propose new knots in an iterative round, ii) produce post-PDA ensembles. In other words, we were sampling from the marginal distributions, and missing further constraint from covariances. We are now sampling the joint posterior distributions for both proposing new knots and generating post-PDA ensembles. At the end of responses, some of our figures are redrawn with new results (please note that other figures will also be redrawn in the revised manuscript). We also include the following explanation at the end of section 2.1 Emulator-based calibration:

L226: In this study, new points were added by proposing 20% of the new knots from the original prior distribution and 80% from the joint posterior of the previous emulator round (via re-sampling the MCMC samples in between rounds).

**Enhancing Literature references:**

**RC 1 - Missing references to the literature include the following.**

AC - We thank the reviewer for going the extra mile and briefly summarizing relevant aspects in all these references. We will include most of them in the manuscript in regarding places.

**RC 1 - Further, as perhaps an unmentioned predecessor of calibrating data-scaling parameters, see the ecosystem model Bayesian calibration approach of Van Oijen et al. (2011), where every separate data stream came with its own bias parameter.**

> L276-278: The bias term is included to account for the scaling from the discrete soil collars to the stand as a whole (Van Oijen et al., 2011). This term was also introduced because observed soil chamber fluxes were typically over twice the ecosystem respiration estimated from the eddy-covariance tower (Phillips et al., 2017). As in previous studies, this parameter is also estimated in the calibration (Van Oijen et al., 2011).

**RC 1 - Jandarov et al. (2014) used the same refinement employed in the present paper, of emulating sufficient statistics instead of the overall likelihood directly.**

AC - If we understood their study correctly, Jandarov et al.'s (2014) approach is related but different than ours. As the spatiotemporal data they were dealing with was high dimensional, likelihood-based inference for their model was becoming intractable. Their

approach consists of obtaining summary statistics from forward simulator runs, and emulating the Euclidean distances between the summary statistics of their simulated data and the summary statistics of the real data. In other words, they compared model and data on a more aggregated level in their calibration. They chose these key summary statistics by expert opinion to capture important characteristics of their modeled process (disease dynamics). The emulated Euclidean distances is then treated as the likelihood function in their study. Whereas in this study, we compared model and data directly, and emulated sufficient statistics of the likelihood. Here, sufficient statistics has a formal mathematical definition (Fisher, 1992; Mikusheva, 2011).

L179: Next (step 3), a sufficient statistic ($T$) is calculated by comparing each model output to each data set (Fig. 1). Statistic $T$ is sufficient for the job of estimating the unknown parameters "when no other statistic calculated from the same sample provides any additional information" (Fisher, 1992).

**RC 1- Oakley Youngman (2017) showed many of the same methods as the present authors do.**

L113-115: Instead of constructing an emulator for the raw model output, we adopt the approach of constructing an emulator of the likelihood – the statistical assessment of the probability of the data given a set of model parameters which forms the basis for both frequentist and Bayesian inference. Emulating the likelihood has the advantage that likelihood surfaces are generally smooth and univariate (Oakley and Youngman, 2017).

L471-473: The efficiency of this workflow could potentially be increased further by other adaptive sampling designs, and this remains an important area for further research. For example, Oakley and Youngman (2017) used an initial set of simulator runs to screen-out low likelihood regions to reduce the parameter space before the calibration. For a review of adaptive

sampling methods, and emulator design methodologies in general, see Forrester and Keane (2009).

**RC 1 - For many examples of likelihood-emulation using Gaussian processes etc. in cosmology, see Aslanyan et al. (2015) and references 7-24 therein (which also tend to focus on how much computations are made faster by likelihood-emulation).**

L476-480 (section 4.2): In this study, we focused on calibrating process-based mechanistic simulators (ecosystem models) using computationally cheaper emulators. Variations of emulator approach are many, and can be found in Jandarov et al., (2014); Aslanyan et al. (2015), Huang et al. (2016), Oakley and Youngman (2017) and the references therein. Here we adopted the version which emulates the likelihood surface with a Gaussian process, similar to previous studies including applications with a cosmological likelihood function (Aslanyan et al., 2015), a stochastic natural history model (Oakley and Youngman, 2017), the Hartman function and a hydrologic model (Wang et al., 2014) and two land surface models (Li et al., 2018). Our scheme also resembles the adaptive surrogate modelling-based optimization approach (ASMO; Wang et al., 2014; Li et al., 2018) in terms of both the nature of the problem (calibration of a process-based mechanistic simulator) and the general scheme of the calibration algorithm. However, aside from differences in initial sampling designs and error characterizations in these studies, there are two main differences of our scheme from ASMO.

**RC 1 - Kandasamy Schneider show that instead of emulating the likelihood, it is also possible to emulate the product of prior and likelihood (i.e. the posterior up to a constant), an approach not mentioned by the present authors.**

AC - We might be looking at the wrong paper (because we found a paper from Kandasamy, Schneider and Poczos by the same name and year, not from Kandasamy Schneider), but this paper also emulates the likelihood surface (they estimate posteriors through emulated likelihoods). However, a paper we are already citing (Gong and Duan, 2017) does emulate posterior surface. Both papers are now cited (see next comment).

**(this is a comment by R2, included here as well for completeness) RC 2 - The argumentation behind the sufficient statistics is not sustained by the experiment.**

AC - We thank the reviewer for pointing this out. We will now extend the following section in the discussion. Please also see our next response.

L493-496: A second addition to our scheme was that we included a further generalization of emulation of the sufficient statistics (T) surface. T is, by definition, sufficient to estimate the simulator (process model) parameters in the MCMC. Unlike emulating the likelihood (this study, Oakley and Youngman, 2017; Kandasamy, Schneider and Poczos, 2015) or the posteriors (Gong and Duan, 2017), emulating T allows us to estimate parameters that are not part of the process model but are part of the statistical data model (the likelihood) as well. In this study, we tested the sufficient statistics emulation for the SoilResp data and updated Gaussian likelihood precision parameter in the MCMC together with other process model parameters. This residual parameter includes both data error and model structural error, and it is not possible to distinguish one from the other with this approach (Van Oijen, 2017). However, when we apply the same calibration scheme to different process models at the same site, because the observation error in the data are the same, the difference in the posteriors of this residual parameter (Fig. S2)* could give us clues about the model structural errors of models relative to each other, as we demonstrate in this study as a proof-of-

concept. However, in our study, use of multiplicative bias parameter further obscures the difference between observation and model structural error.

*Please note that Fig S2 will be redrawn with the revised workflow and will be mentioned explicitly in the results section.

**RC 1 - Published methods for Bayesian calibration increasingly take into account that models are imperfect. There is a discrepancy between model output and reality, even at the best possible setting of model parameter values. This discrepancy is often modeled as a Gaussian Process for which - in the Bayesian calibration - the hyperparameters are estimated together with the regular model parameters. Likelihood-emulation precludes including discrepancy-estimation because model outputs are not calculated during the MCMC. Please add a discussion of this limitation of your approach.**

AC - This is an important point. First, it is worth noting that our current scheme does allow the inclusion of this discrepancy in terms of a bias and variance terms that are estimated together with the regular model parameters. Indeed, the ability to fit the variance term at the same time as the parameters is precisely why we switched to emulating summary statistics. That said, it is true that in the current implementation the bias term is assumed to be a fixed constant, not varying dynamically, and the soil respiration variance is assumed to be homoskedastic (though the tower fluxes are not). Second, we would argue that our approach does not preclude a more flexible bias specification. Indeed, while beyond the scope of the current paper, conceptually it should be possible to use a bias-variance decomposition to separate our single emulator of the error surface into two separate emulators for bias and variance terms. Similarly, our approach does not preclude specifying a likelihood with a temporally autocorrelated error (which is functionally equivalent to a GP error model in the time dimension), and augmenting the emulator with the autocorrelation parameter similar to how we augmented the emulator with the bias term. As discussed in the paper we instead chose to approximate this as an effective sample size correction, both for computational efficiency and because accounting for autocorrelation in an asymmetric heteroskedastic Laplace is more complicated than doing so in a multivariate Normal. Finally, in PEcAn, we are working towards a more general framework for model-data integration that takes into account initial condition / driver / parameter / model structural uncertainty in calibration and prediction. However, this is still work in progress.

We will include the following in the discussion, at the end of section 4.2 after the paragraph in the response above:

> L497: Indeed, implementation of a more formal way of accounting for model structural error (also called the discrepancy between model output and reality) in our emulator scheme is one of our planned next steps. Explicitly specifying a model discrepancy term and estimating it through MCMC would allow us to account for all sources of model predictive uncertainty (Van Oijen, 2017). However, determining the expected form of discrepancy in order to learn about model parameters realistically could be difficult due to lack of mechanistic knowledge of the underlying processes (Brynjarsdottir and O'Hagan, 2014). In that sense, accounting for discrepancy in model calibration is not an emulator approach specific issue. For a novel approach investigating model structural uncertainty through a modular modeling framework see Walker et al. (2018), which could be useful for modeling prior knowledge about discrepancy in ecosystem models in the future. Because of the unknowns about the discrepancy functions, it is common to use Gaussian processes to model the discrepancy (Kennedy and O'Hagan, 2001). Even then, only with realistic prior constraints about the process, calibrated model predictions will be unbiased (Brynjarsdottir and O'Hagan, 2014). For an example of addressing discrepancy in calibration that combines likelihood-emulation approach with importance sampling, see Oakley and Youngman (2017) where they inflated simulator uncertainty to account for simulator discrepancy instead of explicitly specifying a prior for it in order to make the likelihood tractable. When likelihood function becomes intractable, techniques using likelihood-free inference could also be a remedy (Gutmann and Corander, 2016).

**Cited references (that were not mentioned in previous correspondence or in the manuscript):**

Brynjarsdottir and O'Hagan, 2014, Learning about physical parameters: the importance of model discrepancy, 30, IOP, doi:10.1088/0266-5611/30/11/114007

Dietze et al., Beyond MCMC: Data-constraint and error propagation in a dynamic terrestrial biosphere model through Bayesian model emulation, American Geophysical Union, Fall Meeting 2009, http://adsabs.harvard.edu/abs/2009AGUFM.B44A..02D

Erickson, C.B., Ankenman, B.E., Sanchez, S.M., 2018, Comparison of Gaussian process modeling software, European Journal of Operational Research, 266(1), 179-192 https://doi.org/10.1016/j.ejor.2017.10.002

Fisher, R., A., On the mathematical foundations of theoretical statistics, Philosophical Transactions Of The Royal Society A.222:309-368, doi:10.1098/rsta.1922.0009

Gupta, H. V., M. P. Clark, J. A. Vrugt, G. Abramowitz, and M. Ye (2012), Towards a comprehensive assessment of model structural adequacy, Water Resour. Res., 48, W08301, doi:10.1029/2011WR011044.

Loeppky, J. L., Sacks, J., Welch, W. J. (2009). Choosing the sample size of a computer experiment: A practical guide. Technometrics, 51(4), 366–376. doi:10.1198/TECH.2009.08040.

Mikusheva, A., Lecture 4 Sufficient Statistics, course materials for 14.381 Statistical Methods in Economics, Fall 2011. MIT OpenCourseWare (http://ocw.mit.edu), Massachusetts Institute of Technology. Last access on 14/05/2018.

[Figure]

Rasmussen, C. E. (1996). Evaluation of Gaussian Processes and Other Methods for Non-linear Regression. PhD thesis, Dept. of Computer Science, University of Toronto. http://www.kyb.mpg.de/publications/pss/ps2304.ps

Walker, A. P., Ye, M., Lu, D., De Kauwe, M. G., Gu, L., Medlyn, B. E., Rogers, A., and Serbin, S. P.: The Multi-Assumption Architecture and Testbed (MAAT v1.0): Code for ensembles with dynamic model structure including a unified model of leaf-scale C3 photosynthesis, Geosci. Model Dev. Discuss., https://doi.org/10.5194/gmd-2018-71, in review, 2018.

---

## Author Comment (AC3) · 17 May 2018

**RC 2 - The comparison between the results of the emulated and the real SIPNET show that the distributions and central moments of the posteriors are different. This is seen in:**

**a. Figure 3, where there is not "superior" approach across parameters: sometimes is R3, sometimes AAO, sometimes both R2 and R3 are equally good.**

AC - After fixing errors in our algorithm, we have repeated this test with three changes: 1) we now sample the MCMC in the iterative rounds (instead of drawing from marginal

distributions), 2) we now directly plot the MCMC samples (instead of fitted parametric distributions), 3) we now use the contaminated synthetic data. Please see our other response regarding the contamination of the synthetic dataset and Fig 1 below. In the revised figure emulator performance improved and R3 was almost always the best posterior distribution in terms of resolving the true parameter.

**b. Figure 5, where 50% of the emulated SIPNET parameters are (statistically?) different from the central moment of the distribution of the "bruteforce" model calibrations and all of the emulated estimates have substantially higher ranges.**

AC - We redrew this figure after sampling joint posterior distribution and directly plotting MCMC samples themselves instead of using the approximated parametric distributions. Please see our next response and the new figure below (Fig 2).

**Both these results suggest that some further developments have to be investigated in order to rely on posteriors from emulators. It would be key to investigate why the emulators are overall inflating uncertainty and missing the optimum in particular parameters (equifinality? Non-linearities in model functions controlled by those parameters?).**

AC - We thank the reviewer for this remark. Both reviewers' comments indeed helped us investigate further developments in our workflow and visualization. With this latest improvement/fix, the differences between emulator and bruteforce posteriors are diminished further and the emulator medians are notably closer to the optima (Fig 2 below). Our two answers to why emulators are overall inflating uncertainty are the following:

1) There is room for further improvement in the workflow. As re-sampling from the joint posterior distributions rather than the marginal distributions helped with gaining more constraint, other improvements could be thought of: e.g. adaptive sampling design could be further improved, emulator could be passed to more effective algorithms than MH-MCMC, different settings in the mlegp package could be tested to optimize the Gaussian Process (GP) fitting, a different GP package could be used/written (we added

a citation comparing Gaussian process modeling software; Erickson, Ankenman and Sanchez, 2018) etc. We discuss these in the text, and we will be actively working towards such improvements in the future.

> (before the last paragraph of section 4.6): To fit the Gaussian process models in this study, we used the $mlegp$ R-package which was found to be performing well with its default settings (Erickson, Ankenman and Sanchez, 2018). The comparison by Erickson et al. (2018) shows that there are faster (such as $laGP$) and computationally more stable (such as $GPfit$) R-packages available. However, $laGP$ performs worse than $mlegp$ unless thousands of design points are provided, and $GPfit$ is substantially slower than $mlegp$ as it is solely written in R whereas $mlegp$ is pre-compiled in C. Finally, other packages from other platforms (such as the GPy and scikit-learn modules of Python) could outperform $mlegp$ (Erickson, Ankenman and Sanchez, 2018), however, as PEcAn is mainly written in R, $mlegp$ was an adequate choice for our workflow. Overall, we note that approximation error vs clock-time trade-off is not independent of the software/code used to fit the Gaussian process model.

2) The changes to the MCMC algorithm to accommodate emulator interpolation uncertainty, which is the source of the emulators inflating the uncertainty with respect to bruteforce, is an important feature of our algorithm not something that needs to be fixed. As we do not run the full model everywhere in the parameter space, it is important that the emulator not only interpolates between the points in the response surface, but also reflects uncertainty where the model was not run (a reason why we chose gaussian process as the emulator in the first place). If more design points are added, the uncertainty reduces further as shown in Figure 7 in the manuscript. Failure to formally incorporate this interpolation uncertainty (a mistake we ourselves made early in the development of this algorithm) leads to falsely overconfident posteriors that often

exclude the 'true' value.

Equifinality is a problem for bruteforce methods as well. While it is possible that this is slightly exacerbated in the emulator approach, as there is more "wiggle room" for parameter combinations, the formal uncertainty propagation due to GP approximation errors is the main reason for higher uncertainty in emulator posteriors here.

In terms of non-linearities in model functions, GPs are well-suited for the task of emulating non-linear surfaces, and are shown to be performing well regardless of the degree of non-linearities in the fitted surface (Rasmussen, 1996; Rasmussen and Williams, 2006), but are known to have trouble with discontinuities in surfaces. That said, we have no evidence to suggest there are discontinuities in our likelihood surfaces, and indeed the smoothness of most likelihood surfaces is one of the reasons we emulate likelihoods / summary statistics rather than raw model output.

Overall, we agree that further developments should be investigated in this area of research. However, despite the differences, we believe it is encouraging to see emulator posteriors do not exclude the parameter space that the bruteforce suggests, and often agree well with bruteforce posteriors.

**RC 2 - Overall I miss quantitative statistical information about the fitness (model performance) stemming from the parameters obtained via the emulator and the "bruteforce" method against (1) synthetic data and (2) observations (e.g. Nash Sutcfliffe or the Kling Gupta Efficiency). This should also be illustrated by scatter plots and figures that show not only the subdaily but also the seasonal cycle in synthetic/real-world data against models.**

**Knowing the time it takes for the calculations to get done is indeed of technical relevance. But here the most relevant aspects (at least in the perspective of BGD) are centered on how the different model realizations stemming from the emulator approach against the traditional approach change the retrieval of optimal parameters (and posterior uncertainties) and in the eddy covariance flux predictions**

**(for which many relevant information is mostly found in supplements). These are especially important to understand the limitations and caveats of the current proof-of-concept exercise (evaluation of the synthetic exercise).**

AC - We thank the reviewer for pointing out NSE and KGE statistics used in the hydrology literature. We agree with the reviewer that reporting quantitative statistical information about model performance is important. Indeed, we provide performance metrics in the paper currently (please see Table 2 in the manuscript). We report RMSE values, which is related to NSE. In addition, we also report deviance values which takes into account the chosen likelihood and are a more relevant approach regarding the Bayesian framework (Hooten and Hobbs, 2015 as cited in the manuscript). For emulator approach, we report both these metrics against synthetic and real-world data. For bruteforce approach, we only report these metrics against real-world data to provide comparison to the emulator performance. As we already know the true values for synthetic data, we did not feel the need to run bruteforce approach to evaluate emulator performance there. But our workflow is ready to do that in case requested.

We did not present predicted vs observed scatter plots for two reasons: 1) The temporal trends are not visible from such plots, therefore, we decided to go with the more informative unsmoothed time-series plots of both predictions and observations. 2) Such scatter plots are easier to visualize when data is plotted against single model run, while the Bayesian approach produces an ensemble probability distribution of runs. Model ensemble means or medians could be used, but, we wanted to provide the CIs, incorporating the posterior parameter uncertainty. Concentration ellipses could also be used, such as in the an example below (Fig 3). However, we still think temporal trends would be missed by such plots, and including both plots in the main text that essentially test for the same thing would be unnecessary. That is also the reason why we provided diurnal cycle plots in the supplementary, while there is the whole time-series plot in the main text.

Seasonal plots of the model ensembles could also easily be made, but seasonal plots

of the data are not possible without gapfilling the data using some other statistical or mechanistic model, which would then result in a model-model comparison rather than a model-data comparison.

**RC 2 - Another missing important aspect is to understand how the overall results change when contaminating the synthetic dataset with noise (with the same characteristics such as the real observations).**

AC - We thank the reviewer for pointing this out. While it is true that our original analysis did not contaminate the synthetic dataset with noise, our synthetic data had certain characteristics as the real observations. Namely, it had the same gaps as the filtered flux data, and the same coarser time-step and small sample size (n=39) as the real SoilResp data.

We agree that testing against a noisy synthetic dataset is an important test. We contaminated the synthetic dataset with noise based on the uncertainties in the fits to field data and repeated the experiment. Emulator approach showed similar performance with the contaminated synthetic data. Please see Fig 1 below. We believe this is a more proper test than our current version. Therefore, we changed the synthetic data experiment in the main text with the one against the contaminated synthetic data.

> L335: We generated a random parameter set for the SIPNET parameters shown in Fig. 2, and ran the model forward with these values (Table S3). In order to give the synthetic data real characteristics, model outputs were reformatted to have the same gaps, time-steps and sample sizes as the data used in this study. Then, the likelihood parameters were calculated from the synthetic dataset, and next, further noise was added by drawing values from their respective likelihood functions to obtain the final synthetic dataset.

**RC 2- It is not analyzed how does the emulator performance changes by the**

**inclusion of more or less data streams.**

AC - While we agree that effect of including more or less data streams in model cali­bration is an interesting question and an active area of research, we find it to be a more general data assimilation question rather than being emulator specific, and out of the scope of this particular study. We cite two papers that are looking into this question (MacBean et al., 2016; Cameron et al., in prep.). We designed our framework to make assimilation of multiple data streams possible. With more or less data streams, calibra­tion performance of the emulator should still be proportional to bruteforce rather than showing large independent emulator-specific differences.

**RC 2 - There are a few uninformative visuals, like Figure 4 top 2 panels; Figure 3, the som_resp_rate; that could be replaced by more informative elements (new figures, or tables).**

AC - We thank the reviewer for the suggestions. We now replaced Figure 3, som_resp_rate panel with a more informative x-axis range, please see Fig 1 (with contaminated synthetic data) below.

We also agree that top 2 panels of Figure 4 in the manuscript are "busy". However, we have found the current unsmoothed time-series plot to be more informative than smoothed ones or predicted and observed scatter plots with a 1-1 line as it shows the overall temporal trend without biases as mentioned above. The smoothed figures could be drawn only with a gapfilled flux data. However, as it was the unfilled data that the model was calibrated against, this causes some data points to fall out of the the calibrated model CIs as an artifact of gapfilling and smoothing.

Below, we provide both a concentration ellipses version and a smoothed time-series version of model-data comparison (Fig 3 and 4 respectively). We propose to include the smoothed time-series version in the main text, and include the unsmoothed version and the version with ellipses in the supplement for the interested readers, unless the editor suggests otherwise.

**RC 2 - Related to Equation 3, please see the analysis and discussion in Lasslop et al 2008.**

AC - We thank the reviewer for pointing out this paper. Treatment for the asymmetric heteroskedasticity of the flux data is critical for parameter estimation. While Lasslop et al. (2008) argue that the double exponential distribution of fluxes is largely due to a superposition of Gaussian distributions, they showed that distributions of all error estimates still have a Laplacian distribution. It is also assumed that random errors on eddy covariance data would be approximately normal when integrated over a day (Richardson et al., 2010), but in this study we assimilate all fluxes at the half-hourly scale. Therefore, we think the asymmetric heteroskedastic Laplacian distribution choice in our study is justified. We have added a reference to Lasslop et al. (2008).

L248-250: The error distribution of flux data are known to be both heteroskedastic, with variance increasing with the magnitude of the flux, and to have a double exponential distribution (Richardson et al., 2006; Lasslop et al., 2008). In previous studies, the error distributions of high flux magnitudes and fluxes averaged over time were modeled as Gaussian (Lasslop et al., 2008; Richardson et al., 2010). However, as we assimilate all flux magnitudes at half-hourly time-step we modeled the error distributions of NEE and LE fluxes as asymmetric heteroskedastic Laplacian distribution.

**RC 2 - The argumentation behind the sufficient statistics is not sustained by the experiment.**

AC - We thank the reviewer for pointing this out. We will now extend the following section in the discussion. Please also see our next response.

L493-496: A second addition to our scheme was that we included a further generalization of emulation of the sufficient statistics (T) surface. T is,
by definition, sufficient to estimate the simulator (process model) parameters in the MCMC. Unlike emulating the likelihood (this study, Oakley and Youngman, 2017; Kandasamy, Schneider and Poczos, 2015) or the posteriors (Gong and Duan, 2017), emulating T allows us to estimate parameters that are not part of the process model but are part of the statistical data model (the likelihood) as well. In this study, we tested the sufficient statistics emulation for the SoilResp data and updated Gaussian likelihood precision parameter in the MCMC together with other process model parameters. This residual parameter includes both data error and model structural error, and it is not possible to distinguish one from the other with this approach (Van Oijen, 2017). However, when we apply the same calibration scheme to different process models at the same site, because the observation error in the data are the same, the difference in the posteriors of this residual parameter (Fig. S2) could give us clues about the model structural errors of models relative to each other, as we demonstrate in this study as a proof-of-concept. However, in our study use of multiplicative bias parameter further obscures the difference between observation and model structural error.

**(this is a comment by R1 included here for completeness) RC 1 - Published methods for Bayesian calibration increasingly take into account that models are imperfect. There is a discrepancy between model output and reality, even at the best possible setting of model parameter values. This discrepancy is often modeled as a Gaussian Process for which - in the Bayesian calibration - the hyperparameters are estimated together with the regular model parameters. Likelihood-emulation precludes including discrepancy-estimation because model outputs are not calculated during the MCMC. Please add a discussion of this limitation of your approach.**

AC - This is an important point. First, it is worth noting that our current scheme does allow the inclusion of this discrepancy in terms of a bias and variance terms that are

estimated together with the regular model parameters. Indeed, the ability to fit the variance term at the same time as the parameters is precisely why we switched to emulating summary statistics. That said, it is true that in the current implementation the bias term is assumed to be a fixed constant, not varying dynamically, and the soil respiration variance is assumed to be homoskedastic (though the tower fluxes are not). Second, we would argue that our approach does not preclude a more flexible bias specification. Indeed, while beyond the scope of the current paper, conceptually it should be possible to use a bias-variance decomposition to separate our single emulator of the error surface into two separate emulators for bias and variance terms. Similarly, our approach does not preclude specifying a likelihood with a temporally autocorrelated error (which is functionally equivalent to a GP error model in the time dimension), and augmenting the emulator with the autocorrelation parameter similar to how we augmented the emulator with the bias term. As discussed in the paper we instead chose to approximate this as an effective sample size correction, both for computational efficiency and because accounting for autocorrelation in an asymmetric heteroskedastic Laplace is more complicated than doing so in a multivariate Normal. Finally, in PEcAn, we are working towards a more general framework for model-data integration that takes into account initial condition / driver / parameter / model structural uncertainty in calibration and prediction. However, this is still work in progress.

We will include the following in the discussion, at the end of section 4.2 after the paragraph in the response above:

> L497: Indeed, implementation of a more formal way of accounting for model structural error (also called the discrepancy between model output and reality) in our emulator scheme is one of our planned next steps. Explicitly specifying a model discrepancy term and estimating it through MCMC would allow us to account for all sources of model predictive uncertainty (Van Oijen, 2017). However, determining the expected form of discrepancy in order to learn about model parameters realistically could be difficult due to lack of mechanistic knowledge of the underlying processes (Brynjarsdottir and O'Hagan, 2014). In that sense, accounting for discrepancy in model calibration is not an emulator approach specific issue. For a novel approach investigating model structural uncertainty through a modular modeling framework see Walker et al. (2018), which could be useful for modeling prior knowledge about discrepancy in ecosystem models in the future. Because of the unknowns about the discrepancy functions, it is common to use Gaussian processes to model the discrepancy (Kennedy and O'Hagan, 2001). Even then, only with realistic prior constraints about the process, calibrated model predictions will be unbiased (Brynjarsdottir and O'Hagan, 2014). For an example of addressing discrepancy in calibration that combines likelihood-emulation approach with importance sampling, see Oakley and Youngman (2017) where they inflated simulator uncertainty to account for simulator discrepancy instead of explicitly specifying a prior for it, in order to make the likelihood tractable. When likelihood function becomes intractable, techniques using likelihood-free inference could also be a remedy (Gutmann and Corander, 2016).

**Cited references (that were not mentioned in previous correspondence or in the manuscript):**

Brynjarsdottir and O'Hagan, 2014, Learning about physical parameters: the importance of model discrepancy, 30, IOP, doi:10.1088/0266-5611/30/11/114007

Dietze et al., Beyond MCMC: Data-constraint and error propagation in a dynamic terrestrial biosphere model through Bayesian model emulation, American Geophysical Union, Fall Meeting 2009, http://adsabs.harvard.edu/abs/2009AGUFM.B44A..02D

Erickson, C.B., Ankenman, B.E., Sanchez, S.M., 2018, Comparison of Gaussian process modeling software, European Journal of Operational Research, 266(1), 179-192 https://doi.org/10.1016/j.ejor.2017.10.002

Fisher, R., A., On the mathematical foundations of theoretical statistics, Philosophical Transactions Of The Royal Society A.222:309-368, doi:10.1098/rsta.1922.0009

Gupta, H. V., M. P. Clark, J. A. Vrugt, G. Abramowitz, and M. Ye (2012), Towards a comprehensive assessment of model structural adequacy, Water Resour. Res., 48, W08301, doi:10.1029/2011WR011044.

Loeppky, J. L., Sacks, J., Welch, W. J. (2009). Choosing the sample size of a computer experiment: A practical guide. Technometrics, 51(4), 366–376. doi:10.1198/TECH.2009.08040.

Mikusheva, A., Lecture 4 Sufficient Statistics, course materials for 14.381 Statistical Methods in Economics, Fall 2011. MIT OpenCourseWare (http://ocw.mit.edu), Massachusetts Institute of Technology. Last access on 14/05/2018.

Rasmussen, C. E. (1996). Evaluation of Gaussian Processes and Other Methods for Non-linear Regression. PhD thesis, Dept. of Computer Science, University of Toronto. http://www.kyb.mpg.de/publications/pss/ps2304.ps

Walker, A. P., Ye, M., Lu, D., De Kauwe, M. G., Gu, L., Medlyn, B. E., Rogers, A., and Serbin, S. P.: The Multi-Assumption Architecture and Testbed (MAAT v1.0): Code for ensembles with dynamic model structure including a unified model of leaf-scale C3 photosynthesis, Geosci. Model Dev. Discuss., https://doi.org/10.5194/gmd-2018-71, in review, 2018.
* * *
BGD

[Figure]

**Fig. 1.** New Figure 3, contaminated synthetic data version.

[Figure]

**Fig. 2.** New Figure 5, after sampling joint posterior distribution instead of marginal distributions in between iterative emulator rounds.

[Figure]

**Fig. 3.** Predicted vs observed, concentration ellipses version.

[Figure]

[Figure]

**Fig. 4.** Predicted vs observed, smoothed time-series version.

---

## Author Response (AR1)

We would like to express our gratitude to both referees and the associate editor once more for carefully handling of our manuscript. We found the comments very helpful.

The two biggest changes in the revised manuscript from the previous version are: 1) We are now sampling the joint posterior distributions for both proposing new knots in iterative rounds and generating post-PDA ensembles. 2) We now use a synthetic data with noise in our synthetic data experiment.

We repeated our experiments, and revised the text and figures in the light of referee suggestions. We believe both our manuscript and workflow is now improved.

Below please find our point-by-point responses, some of which are reproduced from our previous responses here for completeness. The page (p) and line numbers (L) refer to the marked-up version of the manuscript.

Referee's Comment (RC) 1 - Contrary to statements made in the paper, the techniques used by the authors are for the most part not novel. There is in fact a substantial literature on replacing the likelihood function with more efficient calculation methods, and I shall give pointers to the literature below. Overall it seems that the literature is very poorly referenced in this paper. However, in the field of ecosystem modelling, several techniques described by the authors have been used hardly at all, so the paper can be valuable in introducing the ideas to a new audience.

**RC 2 - The methods are not novel, but application of the method in the field of biogeosciences is in its infancy and the example experiment provided here may be useful in designing further approaches.**

Authors' Comment (AC) - We thank the reviewers for this remark. In terms of our novelty statements, we wanted to explain that this paper is the culmination of work that has started approximately 12 years ago (please see AGU talk abstract by Dietze et al., 2009) and it was rather novel even across disciplines back then. We acknowledge the fact that this is not the case anymore and offer our apologies for missing key papers. We have revised the novelty statements accordingly: p1.L6, p1.L8-10, p4.L10, p4.L12, p8.L22, p14.L10-18

However, as both reviewers highlighted, a decade after they were first introduced, the techniques described in this paper have been used hardly at all in the field of ecosystem modeling. This is not surprising given that applications of these techniques require a non-trivial amount of computational and statistical expertise, not to mention a steep debugging curve of both models and algorithms. In this paper, we report the integration of a standardized ecological application of these methods in an open-source ecological informatics toolbox for the general use of the ecosystem modeling community. It is exactly our hope that the experiments and the implementations provided here may foster more use and development of novel types of model emulators.

RC 1 - To introduce new methodological ideas to people, the language should be clear and consistent, and that is not the case here. There is a worrying lack of understanding of the difference between the concepts of 'error' and 'uncertainty'. The first refers to deviation from truth, the second to incomplete knowledge, but in this paper the terms are occasionally treated as synonyms, which makes the Introduction highly unclear. Proper terminology for these concepts and others can, for example, be found in the review of Bayesian methods by Van Oijen (2017), where also additional references on MCMC, emulation and hierarchical modelling in ecosystem modelling can be found.

The Introduction mentions that "Parameter error refers to the uncertainty about the true values of the model parameters", which is quite wrong. Parameter error means assigning a value to a parameter which differs from reality, e.g. stating that the light- use efficiency is 1 g MJ-1 when in reality it is 2 g MJ-1. Not knowing whether it is 1 or 2 or anything else is uncertainty. It is therefore also incorrect to state, as the authors do, that "parameter error asymptotically goes to zero with enough data". It is the conditional uncertainty that goes to zero, not the error. Every experimentalist knows that having any number of biased measurements makes no parameter converge to its correct value - and all measurements have their hidden or unhidden biases. There is no safe way to "estimate observation error from data".

AC - We completely agree with reviewer's definitions of error and uncertainty, and revised these sections in the introduction: p2.L24-L33, p3.L1-L18

RC 1 - The treatment of the subject matter in the Introduction is further hampered by poor terminology regarding parameters. Terms like "parameter", "parameter vector", "parameter set[s]" are used arbitrarily and inconsistently. [As an exercise for the reader: show that lines 98 and 147 cannot both apply.] Note that a set is unordered and a vector is ordered, so a point in parameter space can not be a "parameter set". And "covariances among parameters" are not real quantities but statistical quantities that capture part of our uncertainty and that change when more data come in. Therefore the covariances are in no way "accounted for". Please note that your subject matter of Bayesian calibration using MCMC is unfamiliar to many readers, so getting an idea of what is going on requires using precise language. Apologies for these pedantic remarks, but in my experience people stumble over the smallest inconsistency when learning Bayesian methods.

AC - We are grateful for such remarks, and have revised the text accordingly: p3.L32-L34, p4.L13, Algorithm 1: step 1-2-5a, p5.L7, p6.L4, p6.L25, p6.L31-32, p7.L3-4

RC 1 - Can you elaborate on the limitations of your approach? What is the maximum number of parameters (\$p\$) that can be calibrated in general, and for your two models in particular? You set the number of model-runs at \$p^3\$. Does that mean that calibrating 100 parameters is unfeasible because it would require \$10^6\$ model evaluations just to build the emulator?

AC - With the current (\$p^3\$) scheme calibrating 100 parameters would be infeasible as it would require \$10^6\$ model evaluations just to build the emulator. With ED2, running the model \$10^6\$ times is not feasible at all, unless iterative emulator rounds are massively parallelized. With SIPNET, the Cholesky decomposition within the GP, rather than the model evaluations, would become limiting for \$10^6\$ design points. In that case, emulators other than GP (e.g. NNGP) could be considered as we discuss in the manuscript.

That said, the ( $p^3$ ) scheme is just the rule-of-thumb that we employed in these experiments, and not an inherent limit of the emulator approach itself. The calibration of 100 parameters might be possible with a much smaller number of knots ( $ll $10^6$ ) depending on the model. For example, our scaling experiment (Figure 7b) shows that, in terms of deviance, it was possible to constrain 6 SIPNET parameters to a reasonable extent with 120 knots in total (likewise, 8 and 10 SIPNET parameters with 240 parameters in total). A common recommendation in computer experiments with GP is to use a sample size about 10 times (n = 10d) the input dimension (Loeppky, Sacks & Welch, 2009). Others found this is often too small and suggest 20 times (n = 20d) larger sample size (Erickson, Ankenman & Sanchez, 2018).

Therefore, calibrating 100 model parameters with 100 x 20 design points could be possible in theory. In practice, we would advocate for performing an uncertainty analysis to reduce the dimensionality of the problem to the subset these 100 parameters that contribute most to model uncertainty. In addition, the data would need to be strong enough to actually constrain 100 parameters. We now extended the text in section 4.6 to include this discussion, p18.L1-12.

**RC 1 - How exactly does PEcAn calculate the contributions of different parameters to overall uncertainty, i.e. what was the screening algorithm?**

AC - The uncertainty analysis in PEcAn uses a one-at-a-time (OAT) approach. An OAT approach involves multiple model runs while holding all parameters at their median except one each time, and evaluating how it translates to differences in model outputs. The parameters are varied at their parameter data assimilation (PDA) analysis priors' (which could be original priors or, if the parameter was constrained by the meta analysis, they could be meta analysis posteriors in PEcAn) median and at six PDA prior quantiles equivalent to \pm[1,2,3]\$\sigma\$ in the standard normal. Details are given in previous papers as cited (LeBauer et al., 2013; Dietze et al., 2014). Plans are in place to develop a more general multivariate uncertainty analysis in the future once the multivariate version of our trait meta-analysis is in place (Shiklomanov et al in review).

RC 1 - There are linguistic errors (plural subjects with singular verbs, missing definite articles etc.) on lines 54, 55, 92, 93, 100, 183, 201, 248, 294, 306 (twice), 309, 323, 351, 372, 418, 434, 436, 443, 454-455, 482 (twice), 483, 484, 485, 507, 511, 520, 539 (twice), 581.

We thank the reviewer for noticing and noting these errors. We went through the text more carefully and believe we have corrected these errors: p2.L15, p2.L16, p3.L27, p3.L34, p5.L14, p6.L13, p7.L23, p9.L7, p9.L16-19, p10.L23-25, p13.L5, p13.L11, p13.L18, p15.L32, p16.L2, p16.L27, p16.L27, p16.L28.

**RC 1 - The last sentence of the Abstract (I. 34-36) can be deleted without loss of content.**

AC - We deleted this sentence, p1.L20.

**RC 1 - How is the "Euclidean distance between confidence intervals" determined?**

AC - Thank you for this question. Realizing that this was not clear in the manuscript, we added more text to the manuscript: p10.L27-30, p11.L1-4.

**RC 1 - Why were 729 knots used for \$p=8\$ parameters of SIPNET, given that you state the need for \$p^3\$ knots (\$729=9^3\$, not \$8^3\$)?**

AC - Because we counted the multiplicative bias parameter in the *\$p\$*. So, 8 SIPNET parameters plus the multiplicative bias parameter, *\$p* = 9*\$* for SIPNET. 9 ED2 parameters (6 of 9 being scaling factors for common PFT parameters) plus the multiplicative bias parameter, *\$p* = 10*\$* for ED2. Thank you for pointing this out. We now state this in the text more explicitly (p9.L28-30).

**RC 1 - Two of the references are not placed in their proper alphabetical position, and the reference to Hartig et al. (2012) is missing.**

AC - We corrected these in the revised manuscript.

**RC 1 - Can you explain the results shown in Tables A2 and A5? How can posterior distributions for parameters following MCMC neatly fall into parameterised probability distributions (which also are often of different type than their priors)?**

AC - As explained previous comments, the results reported in Tables S2 and S5 are fitted parametric distributions to the marginal MCMC samples. We wanted to provide an approximate parametric distribution for the reader for ease of use. Otherwise, all the raw MCMC samples are

accessible via PEcAn for more interested readers (p12.L6-7). We now extended the explanation before Table S6 as well.

Please note that we now plot the Figures 3 and 5 with the raw samples instead of the fitted parametric distributions (previous version).

**Enhancing Literature references:**

**RC 1 - Missing references to the literature include the following.**

AC - We thank the reviewer for going the extra mile and briefly summarizing relevant aspects in all these references. We now included them in the manuscript in regarding places.

RC 1 - As perhaps an unmentioned predecessor of calibrating data-scaling parameters, see the ecosystem model Bayesian calibration approach of Van Oijen et al. (2011), where every separate data stream came with its own bias parameter.

AC - p8.L20-22

**RC 1 - Jandarov et al. (2014) used the same refinement employed in the present paper, of emulating sufficient statistics instead of the overall likelihood directly.**

AC - If we understood their study correctly, Jandarov et al.'s (2014) approach is related but different than ours. As the spatiotemporal data they were dealing with was high dimensional, likelihood-based inference for their model was becoming intractable. Their approach consists of obtaining summary statistics from forward simulator runs, and emulating the Euclidean distances between the summary statistics of their simulated data and the summary statistics of the real data. In other words, they compared model and data on a more aggregated level in their calibration. They chose these key summary statistics by expert opinion to capture important characteristics of their modeled process (disease dynamics). The emulated Euclidean distances is then treated as the likelihood function in their study. Whereas in this study, we compared model and data directly, and emulated sufficient statistics of the likelihood. Here, sufficient statistics has a formal mathematical definition, p5.L10

RC 1- Oakley & Youngman (2017) showed many of the same methods as the present authors do.

AC - p4.L15, p14.L3-4, p14.L10, p14.L29

RC 1 - For many examples of likelihood-emulation using Gaussian processes etc. in cosmology, see Aslanyan et al. (2015) and references 7-24 therein (which also tend to focus on how much computations are made faster by likelihood-emulation).

AC - p14.L8-12

**RC 1 - Kandasamy & Schneider show that instead of emulating the likelihood, it is also possible to emulate the product of prior and likelihood (i.e. the posterior up to a constant), an approach not mentioned by the present authors.**

AC - We might be looking at the wrong paper (because we found a paper from Kandasamy, Schneider and Poczos by the same title and year, not from Kandasamy & Schneider), but this paper also emulates the likelihood surface (they estimate posteriors through emulated likelihoods). However, a paper we are already citing (Gong and Duan, 2017) does emulate posterior surface. Both papers are now cited, p14.L28.

**RC 2 - The argumentation behind the sufficient statistics is not sustained by the experiment.**

AC - We thank the reviewer for this important point. We now extended the text in the discussion (p14.L27 onwards). Please also see our next response.

RC 1 - Published methods for Bayesian calibration increasingly take into account that models are imperfect. There is a discrepancy between model output and reality, even at the best possible setting of model parameter values. This discrepancy is often modeled as a Gaussian Process for which - in the Bayesian calibration - the hyperparameters are estimated together with the regular model parameters. Likelihood-emulation precludes including discrepancy-estimation because model outputs are not calculated during the MCMC. Please add a discussion of this limitation of your approach.

AC - We thank the reviewer for pointing this out. First, it is worth noting that our current scheme does allow the inclusion of this discrepancy in terms of a bias and variance terms that are estimated together with the regular model parameters. Indeed, the ability to fit the variance term at the same time as the parameters is precisely why we switched to emulating summary statistics. That said, it is true that in the current implementation the bias term is assumed to be a fixed constant, not varying dynamically, and the soil respiration variance is assumed to be homoskedastic (though the tower fluxes are not). Second, we would argue that our approach does not preclude a more flexible bias specification. Indeed, while beyond the scope of the current paper, conceptually it should be possible to use a bias-variance decomposition to separate our single emulator of the error surface into two separate emulators for bias and variance terms. Similarly, our approach does not preclude specifying a likelihood with a temporally autocorrelated error (which is functionally equivalent to a GP error model in the time dimension), and augmenting the emulator with the autocorrelation parameter similar to how we

augmented the emulator with the bias term. As discussed in the paper we instead chose to approximate this as an effective sample size correction, both for computational efficiency and because accounting for autocorrelation in an asymmetric heteroskedastic Laplace is more complicated than doing so in a multivariate Normal. Finally, in PEcAn, we are working towards a more general framework for model-data integration that takes into account initial condition / driver / parameter / model structural uncertainty in calibration and prediction. However, this is still work in progress.

We now extended our discussion, at the end of section 4.2, p15.L6 onwards.

**RC 2 - Figure 3, where there is not "superior" approach across parameters: sometimes is R3, sometimes AAO, sometimes both R2 and R3 are equally good.**

AC - We have repeated this test with three changes: 1) we now sample the MCMC in the iterative rounds (instead of drawing from marginal distributions), 2) we now directly plot the MCMC samples (instead of fitted parametric distributions), 3) we now use the contaminated synthetic data. Please see our other response regarding the contamination of the synthetic dataset and new Figure 3. In the revised figure, R3 is always the best posterior distribution in terms of resolving the true parameter.

**RC - 2 Figure 5, where 50\% of the emulated SIPNET parameters are (statistically?) different from the central moment of the distribution of the "bruteforce" model calibrations and all of the emulated estimates have substantially higher ranges.**

AC - We redrew this figure after sampling joint posterior distribution and directly plotting MCMC samples themselves instead of using the approximated parametric distributions. Please see our next response and the new Figure 5 in the revised manuscript.

Both these results suggest that some further developments have to be investigated in order to rely on posteriors from emulators. It would be key to investigate why the emulators are overall inflating uncertainty and missing the optimum in particular parameters (equifinality? Non-linearities in model functions controlled by those parameters?).

AC - We thank the reviewer for this remark. Both reviewers' comments indeed helped us investigate further developments in our workflow and visualization. With this latest improvement/fix, the differences between emulator and bruteforce posteriors are diminished further and the emulator medians are notably closer to the optima. Our two answers to why emulators are overall inflating uncertainty are the following:

1) There is room for further improvement in the workflow. As re-sampling from the joint posterior distributions rather than the marginal distributions helped with gaining more constraint, other

improvements could be thought of: e.g. adaptive sampling design could be further improved, emulator could be passed to more effective algorithms than MH-MCMC, different settings in the *mlegp* package could be tested to optimize the Gaussian Process (GP) fitting, a different GP package could be used/written (we now added a citation comparing Gaussian process modeling software; p17.L25-32) etc. We discuss these in the text, and we will be actively working towards such improvements in the future.

2) The changes to the MCMC algorithm to accommodate emulator interpolation uncertainty, which is the source of the emulators inflating the uncertainty with respect to bruteforce, is an important feature of our algorithm not something that needs to be fixed. As we do not run the full model everywhere in the parameter space, it is important that the emulator not only interpolates between the points in the response surface, but also reflects uncertainty where the model was not run (a reason why we chose gaussian process as the emulator in the first place). If more design points are added, the uncertainty reduces further as shown in Figure 7 in the manuscript. Failure to formally incorporate this interpolation uncertainty (a mistake we ourselves made early in the development of this algorithm) leads to falsely overconfident posteriors that often exclude the 'true' value.

Equifinality is a problem for bruteforce methods as well. While it is possible that this is slightly exacerbated in the emulator approach, as there is more "wiggle room" for parameter combinations, the formal uncertainty propagation due to GP approximation errors is the main reason for higher uncertainty in emulator posteriors here.

In terms of non-linearities in model functions, GPs are well-suited for the task of emulating non-linear surfaces, and are shown to be performing well regardless of the degree of non-linearities in the fitted surface (Rasmussen, 1996; Rasmussen and Williams, 2006), but are known to have trouble with discontinuities in surfaces. That said, we have no evidence to suggest there are discontinuities in our likelihood surfaces, and indeed the smoothness of most likelihood surfaces is one of the reasons we emulate likelihoods / summary statistics rather than raw model output.

Overall, we agree with the reviewer that further developments should be investigated in this area of research. However, despite the differences, we believe it is encouraging to see emulator posteriors do not exclude the parameter space that the bruteforce suggests, and often agree well with bruteforce posteriors.

RC 2 - Overall I miss quantitative statistical information about the fitness (model performance) stemming from the parameters obtained via the emulator and the "bruteforce" method against (1) synthetic data and (2) observations (e.g. Nash Sutcfliffe or the Kling Gupta Efficiency). This should also be illustrated by scatter plots and figures that show not only the subdaily but also the seasonal cycle in synthetic/real-world data against models.

AC - We thank the reviewer for pointing out NSE and KGE statistics used in the hydrology literature. We agree with the reviewer that reporting quantitative statistical information about model performance is important. Indeed, we provide performance metrics in the paper currently (please see Table 2 in the manuscript). We report RMSE values, which is related to NSE. In addition, we also report deviance values which takes into account the chosen likelihood and are a more relevant approach regarding the Bayesian framework (Hooten & Hobbs, 2015 as cited in the manuscript). For emulator approach, we report both these metrics against synthetic and real-world data. For bruteforce approach, we only report these metrics against real-world data to provide comparison to the emulator performance. As we already know the true values for synthetic data, we did not feel the need to run bruteforce approach to evaluate emulator performance there. But our workflow is ready to do that in case requested.

We did not present predicted vs observed scatter plots for two reasons: 1) The temporal trends are not visible from such plots. Also, including both plots (time-series and scatter) in the main text that essentially test for the same thing (observed vs. predicted) would be unnecessary. That was also the reason why we provided diurnal cycle plots in the supplementary, while there is the whole time-series plot in the main text. Therefore, we decided to go with the more informative unsmoothed time-series plots of both predictions and observations. 2) Such scatter plots are easier to visualize when data is plotted against single model run, while the Bayesian approach produces an ensemble probability distribution of runs. In that case, concentration ellipses might be useful.

Seasonal plots of the data require gapfilling the data using some other statistical or mechanistic model, which result in a model-model comparison rather than a model-data comparison. However, also considering the next suggestion by the reviewer (please see our next response), we agree to provide the seasonal (monthly smoothed) time-series in the main text (please see new Figure 4 and 6 in the revised manuscript), move the unsmoothed time series to the appendix, and include an additional observed vs. predicted ellipses plot of the in the appendix.

**RC 2 - There are a few uninformative visuals, like Figure 4 top 2 panels; Figure 3, the som\_resp\_rate; that could be replaced by more informative elements (new figures, or tables).**

AC - We thank the reviewer for the suggestions. We now replaced Figure 3, som\_resp\_rate panel with a more informative x-axis range, please see new Figure 3 (with contaminated synthetic data) in the revised manuscript.

As mentioned above, we have found the previous unsmoothed time-series plot to be more informative than smoothed ones. The smoothed figures could be drawn only with a gapfilled flux data. However, as it was the unfilled data that the model was calibrated against, this causes some data points to fall out of the the calibrated model CIs as an artifact of gapfilling and smoothing.

But we also agree that top 2 panels of Figure 4 (and likewise Figure 6) in the manuscript were "busy". We now included the smoothed time-series version in the main text, and the unsmoothed version in the supplement for the interested readers.

**RC 2 - Another missing important aspect is to understand how the overall results change when contaminating the synthetic dataset with noise (with the same characteristics such as the real observations).**

AC - We thank the reviewer for pointing this out. While it is true that our original analysis did not contaminate the synthetic dataset with noise, our synthetic data had certain characteristics as the real observations. Namely, it had the same gaps as the filtered flux data, and the same coarser time-step and small sample size (n=39) as the real SoilResp data.

We agree that testing against a noisy synthetic dataset is an important test. We contaminated the synthetic dataset with noise based on the uncertainties in the fits to field data and repeated the experiment. Emulator approach showed similar performance with the contaminated synthetic data (please see new Figure 3 in the revised manuscript). We believe this is a more proper test than our previous version. Therefore, we changed the synthetic data experiment in the main text with the one against the contaminated synthetic data, p10.L6-10.

**RC 2- It is not analyzed how does the emulator performance changes by the inclusion of more or less data streams.**

AC - While we agree that effect of including more or less data streams in model calibration is an interesting question and an active area of research, we consider it to be a more general data assimilation question rather than being emulator specific, and out of the scope of this particular study. We designed our framework to make assimilation of multiple data streams possible. With more or less data streams, calibration performance of the emulator should still be proportional to bruteforce rather than showing large independent emulator-specific differences. We now cite two papers that are looking into this question specifically (Keenan et al., 2013; MacBean et al., 2016) and explicitly state this in the text, p16.L20-22

**RC 2 - Related to Equation 3, please see the analysis and discussion in Lasslop et al 2008.**

AC - We thank the reviewer for pointing out this paper. Treatment for the asymmetric heteroskedasticity of the flux data is critical for parameter estimation. While Lasslop et al. (2008) argue that the double exponential distribution of fluxes is largely due to a superposition of Gaussian distributions, they showed that distributions of all error estimates still have a Laplacian

distribution. It is also possible to assume that random errors on eddy covariance data would be approximately normal when integrated over a day (Richardson et al., 2010), but in this study we assimilate all fluxes at the half-hourly scale. Therefore, we think the asymmetric heteroskedastic Laplacian distribution choice in our study is justified. We still find it useful to add a reference to Lasslop et al. (2008), p7.L25-30.

Brynjarsdottir and O'Hagan, 2014, Learning about physical parameters: the importance of model discrepancy, 30, IOP, doi:10.1088/0266-5611/30/11/114007

Dietze et al., Beyond MCMC: Data-constraint and error propagation in a dynamic terrestrial biosphere model through Bayesian model emulation, American Geophysical Union, Fall Meeting 2009, http://adsabs.harvard.edu/abs/2009AGUFM.B44A..02D

Erickson, C.B., Ankenman, B.E., Sanchez, S.M., 2018, Comparison of Gaussian process modeling software, European Journal of Operational Research, 266(1), 179-192 https://doi.org/10.1016/j.ejor.2017.10.002

Fisher, R., A., On the mathematical foundations of theoretical statistics, Philosophical Transactions Of The Royal Society A.222:309-368, doi:10.1098/rsta.1922.0009

Gupta, H. V., M. P. Clark, J. A. Vrugt, G. Abramowitz, and M. Ye (2012), Towards a comprehensive assessment of model structural adequacy, Water Resour. Res., 48, W08301, doi:10.1029/2011WR011044.

Loeppky, J. L., Sacks, J., & Welch, W. J. (2009). Choosing the sample size of a computer experiment: A practical guide. *Technometrics*, *51*(4), 366–376. doi:10.1198/ TECH.2009.08040.

Mikusheva, A., Lecture 4 Sufficient Statistics, course materials for 14.381 Statistical Methods in Economics, Fall 2011. MIT OpenCourseWare (http://ocw.mit.edu), Massachusetts Institute of Technology. Last access on 14/05/2018.

Rasmussen, C. E. (1996). Evaluation of Gaussian Processes and Other Methods for Non-linear Regression. PhD thesis, Dept. of Computer Science, University of Toronto. http://www.kyb.mpg.de/publications/pss/ps2304.ps

Walker, A. P., Ye, M., Lu, D., De Kauwe, M. G., Gu, L., Medlyn, B. E., Rogers, A., and Serbin, S. P.: The Multi-Assumption Architecture and Testbed (MAAT v1.0): Code for ensembles with dynamic model structure including a unified model of leaf-scale C3 photosynthesis, Geosci. Model Dev. Discuss., https://doi.org/10.5194/gmd-2018-71, in review, 2018.

**Linking big models to big data: efficient ecosystem model calibration through Bayesian model emulation**

Istem Fer1, Ryan Kelly2, Paul R. Moorcroft3, Andrew D. Richardson4,5, Elizabeth M. Cowdery1, and Michael C. Dietze1

1Department of Earth and Environment, Boston University, Boston, MA 02215, USA
2RK Analytics, Durham, NC 27712, USA
3Department Organismic and Evolutionary Biology, Harvard University, Cambridge, MA 02138, USA
4School of Informatics, Computing and Cyber Systems, Northern Arizona University Flagstaff, AZ 86011, USA
5Center for Ecosystem Science and Society, Northern Arizona University, Flagstaff, AZ 86011, USA

Correspondence: Istem Fer (fer.istem@gmail.com)

[revised manuscript text omitted]

The Bayesian approach also distinguishes between uncertainties due to process, data, and parameter errors, parametric, 25 model structural and data uncertainties, which is critical for ecological forecasting. Parameter error uncertainty refers to the uncertainty about the true values of the model parameters (e.g. variation among individuals, across sites) due to data deficiency (McMahon et al., 2009). By definition, parameter error asymptotically goes to zero with enough data (Dietze, 2017a) and model simplification (McMahon et al., 2009; van Oijen, 2017). As models are simplified representations of reality, it is often not possible to measure the true value of an ecosystem model parameter precisely in the field, regardless of the measurement

30 errors (van Oijen, 2017). However, until such time, the measurements can still provide estimates for parameter values that makes the model represent the reality better (van Oijen, 2017). Hence, it is possible to reduce parameter uncertainty with more measurements, conditioned upon the model structure and the measurement error (van Oijen, 2017; Dietze, 2017a). Therefore, the parameter uncertainty should also be be reflected by probability distributions and propagated into model predictions. By

reducing parameter errorsuncertainties, PDA helps us identify where we need further data collection and improved model representations.

By contrast, process error refers to our inability to capture the ecological processes with deterministic modelsdue to the many or model structural uncertainty refers to the uncertainty about how to represent ecological processes in models. As every

- 5 model is a simplification of reality, there will always be underrepresented processes or insufficiently modeled interactions in ecology (Clark, 2005; McMahon et al., 2009) ecological models (van Oijen, 2017; McMahon et al., 2009; Clark, 2005). With more observations, we can advance our theoretical understanding and better characterize process variability, but all models are approximations of reality, so process error ecological processes, but process uncertainty does not necessarily decrease asymptotically to zerowith more data, the way parameter uncertainty does (Clark, 2005; Dietze, 2017a) (Dietze, 2017a; Gupta et al., 2012;

[revised manuscript text omitted]
, instead of emulating the model outputs, we emulated the likelihood surface. In that sense, our scheme is similar to the we focused on calibrating process-based mechanistic simulators (ecosystem models) using computationally cheaper emulators. Variations of emulator approach are many, and can be found in Jandarov et al. (2014), Aslanyan et al. (2015),

- 10 Huang et al. (2016), Oakley and Youngman (2017) and the references therein. Here we adopted the version which emulates the likelihood surface with a Gaussian process, similar to previous studies including applications with a cosmological likelihood function (Aslanyan et al., 2015), a stochastic natural history model (Oakley and Youngman, 2017), the Hartman function and a hydrologic model (Wang et al., 2014) and two land surface models (Li et al., 2018). Our scheme resembles the adaptive surrogate modelling-based optimization approach (ASMO; Wang et al. (2014)) which fits the emulator to the performance
- 15 metrics for the simulation model, and has been recently used in optimization of two land surface models (Li et al., 2018). In addition to (
[revised manuscript text omitted]

5

---

## Author Response (AR2)

We thank the Associate Editor Dr. Sönke Zaehle and the anonymous reviewer for their helpful comments. We addressed them all in a revised version, please find them listed below as point-by-point responses (line numbers refer to the marked up version).

**Referee's Comments (RC) – Ensure that all symbols used are described, for example I couldn't see where \tau in Equation 1 was described, also u* on page 7.**

Authors' Comments (AC) - We thank the reviewer for pointing these out. We now added descriptions "precision parameter" below Eqn 1 on p7.L17, and "friction velocity" on p6.L2 for \tau and u*, respectively.

**RC - Page 2, line 33 "By reducing parameter uncertainties, PDA helps us identify where we need further data collection and improved model representations" - is it by reducing parameter uncertainties that you do this? The sentence doesn't really tell me much.**

AC – We thank the reviewer for highlighting this sentence. We deleted this sentence as it does not change the flow and as this issue is examined in the discussion section.  p2.L32

**RC - Many of the symbols with words as subscripts need reformatting. I assume LaTeX was used, so for example $N_{eff}$ should be $N_{\mbox eff}$ that will bring the letters in 'eff' together, and not appear as if they are separate variables. Many other examples, e.g. RCI_FINAL, N_parameters.**

AC – We thank the reviewer for pointing this out, we went through the subscripts and added \mbox where necessary.

**RC - Page 9, line 27 (also line 30 and page 17 line 22) - "..., we conducted an uncertainty analysis, ..." - is this better described as a "sensitivity analysis"?**

AC – We thank the reviewer for this remark. We preferred to use "uncertainty analysis" because overall this analysis partitions model predictive uncertainty into the contributions from different model parameters. This can happen in three ways: i) a parameter can remain highly uncertain after the meta-analysis (i.e. not enough data has been found on this parameter from the previous studies or measurements), ii) model calculations can be highly sensitive to the parameter (even if it varies in a relatively narrow range), iii) or both. Sensitivity analysis is certainly a part of this uncertainty analysis workflow. In other words, the model predictive uncertainty analysis considers both parameter uncertainty and model sensitivity. The two cited papers, LeBauer et al. 2013 and Dietze et al. 2014 explains PEcAn uncertainty workflow in more detail. In order to increase the clarity, we changed the phrase to "predictive uncertainty analysis". p9.L19-22

**RC - Caption of Fig A3 - Explain the numbers in the upper right panels.**

AC - We now extended the caption of Fig A3 to explain the panels better.

**Editor's Comments (EC) – p 4 L30: add "of the joint distribution" to avoid confusion (each variable is sampled uniformly, so for each variable, the tails and the centre are equally likely, making the last part of the sentence illogical).**

AC – We thank the editor for pointing this out, indeed this sentence is not clear. We meant to state that the parameters were not sampled uniformly from a given range, but they are sampled as uniform quantiles of their prior distributions (i.e. Inverse transform sampling). If their prior is a uniform distribution, then this would result in uniform sampling from that range. If their their prior is not uniform, the uniform quantiles of this distribution will not result in uniform sampling but in greater sampling in the regions of higher probability and less sampling in the tails.

We now added the emphasis at the end of this sentence as: "In the current application, the sequences for each variable are constructed to be uniform quantiles of the prior distributions (see section, *Model information and priors*), which results in greater sampling in the regions of higher probability and less sampling in the tails **for non-uniform prior distributions.**" p5.L2

**EC - p9 L 21: How is uncertainty defined here, and based on which model output. Obviously, parameters will have different effect sizes on different variables of interest, so the choice of this criterion does matter for the validity of the approach.**

AC – We thank the editor for this remark. Here "uncertainty" refers to the "model predictive uncertainty" as now stated explicitly in the manuscript after reviewer's comment above.
It is correct that parameters will have different effects on different variables of interest. We based it on theNEE, LE and SoilResp outputs of the model, as can be seen in Figure 2, as these map to the data used to constrain the model. For example, if a parameter was contributing more than 5% to SIPNET's NEE output it was chosen for calibration even if it does not contribute to uncertainty in LE predictions, whereas some parameters such as optimum temperature for photosynthesis were contributing to uncertainty of more than one model output variable. In order to facilitate comparisons among the parameter effects on output variables despite differences in the units, predictive partial variances were used (by normalization: contribution of each parameter to variable of interest divided by overall predictive variance for that variable). We now revised this section accordingly. p9.L25-27

**EC - P9 L24. Having problems understanding the iteratively here, maybe "for each parameter set" is clearer?**

AC – We apologize for the confusion. Here "iteratively" refers to the emulator rounds. For example, if $9^3$ (= 729) knots were to be proposed in total, in the first round emulator was built on 243 (= 729 ÷ 3) knots (i.e. model was run 243 times with 243 parameter vectors), in the second round it was built on 486 (= 243 + 243) knots, and in the third and last round it was built on 729 (= 486 + 243) knots. We rephrased the sentence now. p9.L31

**EC - Table 2: explain the use of bold font.**

AC - We put this explanation as the table footnote as "Bold RMSE values for NEE and SoilResp were rescaled by $10^9$ for easier comparison." We can move it to the table caption if requested.

**EC - P15 L 24: remove reference to Cameron et al. in prep, unless this is now in press or open-discussion.**

AC – We removed the reference to Cameron et al. in prep as this paper is neither in press or under open-discussion currently.

**EC - P18 L 17: to my knowledge, M van Ojien is not an editor for Biogeosciences**

[revised manuscript text omitted]

---

## Author Response (AR3)

Dear Associate Editor Dr. Sönke Zaehle,

**there are units missing for Table 2, Table A4 (alternatively add a new table corresponding to Table A2), Figure A2 and Figure A6, which you will please add.**

Thank you very much for the catching these. We now added the missing units to Table 2, Figure A2 and Figure A6. We also added an extra table now (A4) that further explains calibrated ED2 model parameters with their corresponding units and revised the table references in the main text accordingly.